# From Muon to Gluon: Bridging Theory and Practice of LMO-based Optimizers for LLMs

Artem Riabinin [1]  Egor Shulgin [1]  Kaja Gruntkowska [1]  Peter Richtárik [1]

## Abstract

Recent developments in deep learning optimization have brought about radically new algorithms based on the Linear Minimization Oracle (LMO) framework, such as Muon (Jordan et al., 2024b) and Scion (Pethick et al., 2025b). After over a decade of Adam's dominance, these LMO-based methods are emerging as viable replacements, offering several practical advantages such as improved memory efficiency, better hyperparameter transferability, and most importantly, superior empirical performance on large-scale tasks, including LLM training. However, a significant gap remains between their practical use and our current theoretical understanding: prior analyses (1) overlook the layer-wise LMO application of these optimizers in practice, and (2) rely on an unrealistic smoothness assumption, leading to impractically small stepsizes. To address both, we propose a new LMO-based framework called Gluon, capturing prior theoretically analyzed methods as special cases, and introduce a new refined generalized smoothness model that captures the layer-wise geometry of neural networks, matches the layer-wise practical implementation of Muon and Scion, and leads to state-of-the-art convergence guarantees. Our experiments with `NanoGPT` and `CNN` confirm that our assumption holds along the optimization trajectory, ultimately narrowing the gap between theory and practice.

## 1. Introduction

The success of deep learning models across a wide range of challenging domains is inseparable from the optimization algorithms used to train them. As neural networks have grown deeper and datasets larger, optimization has quietly become one of the most consequential components of modern machine learning (ML). Nowhere is this more evident than in the training of large language models (LLMs), which routinely consume thousands of GPU-hours. Adam (Kingma & Ba, 2015) (and lately AdamW (Loshchilov & Hutter, 2019))—being effective, relatively reliable, and widely adopted—has for over a decade served as the default choice for this task. While this reliance has powered much of deep learning's progress, it has also exposed the shortcomings of adaptive moment estimation as a one-size-fits-all solution–namely, sensitivity to learning rate schedules, heavy tuning requirements (Wilson et al., 2017), and poor generalization when not carefully calibrated (Zou et al., 2023). However, a shift may now be underway. Recent optimizers, such as Muon (Jordan et al., 2024b) and Scion (Pethick et al., 2025b), represent a significant departure from Adam-type methods: they forgo the adaptive moment estimation in favor of a geometry-aware approach inspired by Frank-Wolfe algorithms (Frank & Wolfe, 1956; Pokutta, 2024). These optimizers are not only simpler to implement and easier to tune, but also appear empirically stronger, outperforming AdamW in LLM training (Liu et al., 2025a; Pethick et al., 2025b).

Yet, despite their potential, these new methods are still in their infancy, and our understanding of their theoretical foundations and practical utility in LLM training remains incomplete. Prior convergence guarantees in realistic nonconvex regimes are still far from satisfactory. Indeed, as we argue in Section 2, the (very few) existing theoretical analyses *fail to capture the true algorithms used in practice*, focusing instead on simplified variants that diverge from actual implementations. We identify two key mismatches—*neglect of layer-wise structure* (Section 2.1) and flawed stepsize choices stemming from an *inaccurate smoothness model* (Section 2.2)—and close this gap with a *solution to both*. We elaborate on these advances in the remainder of the paper.

Our goal is to solve the general optimization problem

$$\min_{X \in \mathcal{S}} \{f(X) := \mathbb{E}_{\xi \sim \mathcal{D}}\left[f_\xi(X)\right]\}, \tag{2}$$

where $\mathcal{S}$ is a finite-dimensional vector space and $f_\xi : \mathcal{S} \mapsto$

[1]King Abdullah University of Science and Technology (KAUST), Thuwal, Saudi Arabia. Correspondence to: Egor Shulgin <shulgin.yegor@gmail.com>.

*Proceedings of the 43rd International Conference on Machine Learning*, Seoul, South Korea. PMLR 306, 2026. Copyright 2026 by the author(s).

**Algorithm 1** Gluon: Stochastic Adaptive Layer-Wise LMO-based Optimizer with Momentum

---

1: **Input:** Initial model parameters $X^0 = [X_1^0, \ldots, X_p^0] \in \mathcal{S}$, momentum $M^0 = [M_1^0, \ldots, M_p^0] \in \mathcal{S}$, momentum decay factors $\beta^k \in [0, 1)$ for all iterations $k \geq 0$
2: **for** $k = 0, 1, 2, \ldots, K - 1$ **do**
3:     Sample $\xi^k \sim \mathcal{D}$
4:     **for** $i = 1, 2, \ldots, p$ **do**
5:         Compute stochastic gradient $\nabla_i f_{\xi^k}(X^k)$ for layer $i$
6:         Update momentum for layer $i$ via

$$M_i^k = \beta^k M_i^{k-1} + (1 - \beta^k) \nabla_i f_{\xi^k}(X^k)$$

7:         Choose adaptive stepsize/radius $t_i^k > 0$ for layer $i$
8:         Update parameters for layer $i$ via LMO over $\mathcal{B}_i^k := \{X_i \in \mathcal{S}_i : \|X_i - X_i^k\|_{(i)} \leq t_i^k\}$ via

$$\begin{aligned} X_i^{k+1} &= \mathrm{LMO}_{\mathcal{B}_i^k}\left(M_i^k\right) \\ &:= \underset{X_i \in \mathcal{B}_i^k}{\arg\min} \langle M_i^k, X_i \rangle_{(i)} \end{aligned} \quad (1)$$

9:     **end for**
10:   Update full parameter vector $X^{k+1} = [X_1^{k+1}, \ldots, X_p^{k+1}]$
11: **end for**

---

$\mathbb{R}$ are potentially non-convex and non-smooth but continuously differentiable functions. Here, $f_\xi(X)$ represents the loss of model parameterized by $X$ associated with training data point $\xi$ sampled from probability distribution $\mathcal{D}$. To make the problem meaningful, we assume that $f^{\inf} := \inf_{X \in \mathcal{S}} f(X) > -\infty$. In this work we are particularly interested in the scenario when the parameter vector $X \in \mathcal{S}$ is obtained by collecting the matrices $X_i \in \mathcal{S}_i := \mathbb{R}^{m_i \times n_i}$ of trainable parameters across all layers $i = 1, \ldots, p$ of a deep model. For simplicity, we therefore write $X = [X_1, \ldots, X_p]$. This means that, formally, $\mathcal{S}$ is the $d$-dimensional product space

$$\mathcal{S} := \bigotimes_{i=1}^{p} \mathcal{S}_i \equiv \mathcal{S}_1 \otimes \cdots \otimes \mathcal{S}_p,$$

where $d := \sum_{i=1}^{p} m_i n_i$. With each space $\mathcal{S}_i$ we associate the trace inner product $\langle X_i, Y_i \rangle_{(i)} := \mathrm{tr}(X_i^\top Y_i)$ for $X_i, Y_i \in \mathcal{S}_i$, and an arbitrary norm $\|\cdot\|_{(i)}$, not necessarily induced by the inner product.

## 2. Theory vs. Practice of Muon and Scion

In this work, we focus on an algorithm based on iteratively calling linear minimization oracles (LMOs) across all layers, formalized in Algorithm 1, for which we coin the name

Gluon. In particular, for each layer $i$, independently across all layers, Gluon iteratively updates the parameters via

$$X_i^{k+1} = \mathrm{LMO}_{\mathcal{B}_i^k}(M_i^k) := \underset{X_i \in \mathcal{B}_i^k}{\arg\min} \langle M_i^k, X_i \rangle_{(i)},$$

where $\mathcal{B}_i^k := \{X_i \in \mathcal{S}_i : \|X_i - X_i^k\|_{(i)} \leq t_i^k\}$ and $t_i^k > 0$ is an adaptively chosen stepsize/radius/learning rate.[1] Note that the momentum $M^k = [M_1^k, \ldots, M_p^k] \in \mathcal{S}$ accumulates the contributions from the stochastic gradients $\nabla f_{\xi^k}(X^k) = [\nabla_1 f_{\xi^k}(X^k), \ldots, \nabla_p f_{\xi^k}(X^k)] \in \mathcal{S}$ (see Step 1 of Algorithm 1).

The Gluon framework generalizes a range of methods, including Muon and Scion, which are recovered as special cases under specific norm choices (see Section 4.1 and Appendix D.1). Beyond their ability to outperform AdamW on large-scale benchmarks, these optimizers offer a number of attractive properties: improved memory efficiency, greater robustness to hyperparameter settings, and the ability to transfer those settings across model sizes (Pethick et al., 2025b; Shah et al., 2025). Moreover, in contrast to Adam, they were theoretically analyzed shortly after release and are guaranteed to converge under standard assumptions of Lipschitz smoothness[2] and bounded variance of stochastic gradients (Kovalev, 2025; Li & Hong, 2025; Pethick et al., 2025b).

Gluon presents the method that is deployed in practice (Jordan et al., 2024a; Pethick et al., 2025a) and has proven highly effective. That said, we argue that existing analyses (Kovalev, 2025; Li & Hong, 2025; Pethick et al., 2025b) do *not* accurately reflect this implementation, diverging from it in two key ways. As such, they *fail to explain why the algorithm performs so well*. Let us detail why.

### 2.1. Layer-Wise Structure

First, we briefly walk through the theoretical understanding offered by previous studies. Muon is an optimizer specifically designed for hidden layers, leaving the first and last layers to be handled by some other optimizer, e.g., Adam(W). Its original introduction by Jordan et al. (2024b) was purely empirical, with no attempt at theoretical analysis. The first convergence result came from Li & Hong (2025), who analyzed the smooth nonconvex setting but focused solely on problem (2) with $p = 1$, effectively limiting the scope to the

---

[1] In this context, the radii defining the norm balls in the LMOs effectively act as stepsizes–see Appendix C.1. Accordingly, we use the terms *radius*, *stepsize*, and *learning rate* interchangeably throughout.

[2] A function $f : \mathcal{S} \mapsto \mathbb{R}$ is $L$-smooth if $\|\nabla f(x) - \nabla f(y)\|_\star \leq L \|x - y\|$ for all $x, y \in \mathcal{S}$, where $\mathcal{S}$ is a finite-dimensional vector space equipped with a norm $\|\cdot\|$ and $\|\cdot\|_\star$ is the dual norm associated with $\|\cdot\|$.

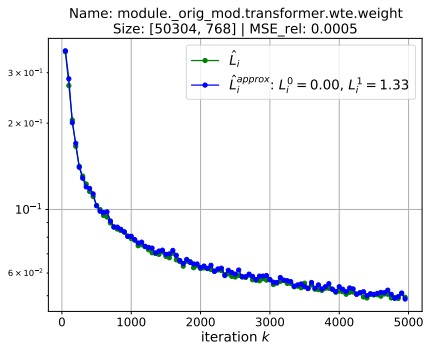
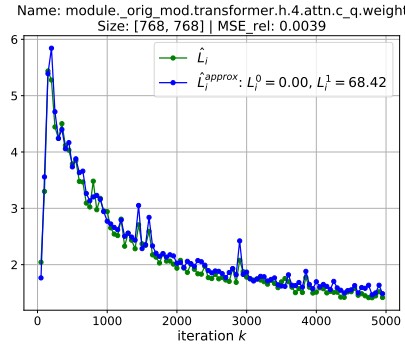
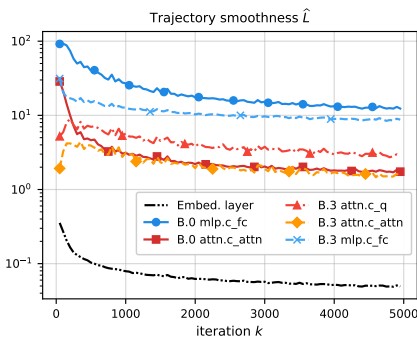

*(a)* Token embedding matrix from the first/last layer.

*(b)* Self-attention query matrix from the 4th transformer block.

*(c)* Trajectory smoothness across different blocks (B.$i$) and layers.

*Figure 1.* Training `NanoGPT` on `FineWeb` validates our layer-wise $(L^0, L^1)$-smoothness model.

single-layer case. The Scion[3] optimizer (a special case of Gluon) proposed by Pethick et al. (2025b) improves upon Muon by applying the LMO-based rule to all layers, ultimately achieving better empirical performance. Both this work and that of Kovalev (2025) analyze (a variant of) the general update rule

$$M^k = \beta^k M^{k-1} + (1 - \beta^k)\nabla f_{\xi^k}(X^k),$$
$$X^{k+1} = \mathrm{LMO}_{\mathcal{B}^k}(M^k), \tag{3}$$

where $\beta^k \in [0, 1)$ is momentum, $\nabla f_{\xi^k}(X^k)$ is the stochastic gradient sampled at iteration $k$, and $\mathcal{B}^k := \{X \in \mathcal{S} : \|X - X^k\| \leq t^k\}$ is a norm ball centered at $X^k$ with stepsize $t^k > 0$. This setup closely resembles the structure of Gluon, but is *not* exactly the same. Indeed, Gluon updates the parameters *layer-wise*, not jointly over the full vector $X$. This distinction is critical since for practical, extremely high-dimensional models, calculating a single global LMO for the entire parameter vector is prohibitively expensive, while breaking the problem into "smaller", per-layer LMOs restores computational feasibility.

Motivated by this disconnect, we formulate our analysis in the matrix product space $\mathcal{S}$, explicitly honoring the layer-wise structure. This enables us to study the actual per-layer updates (1), with assumptions and hyperparameters adapted to each layer.

## 2.2. Toward Bridging Theory and Practice

All prior works claiming to guarantee convergence of Algorithm 1 come with several serious analytical shortcomings–and these directly translate into practical deficiencies. Concretely, all existing analyses of Muon/Scion are built on the classical $L$-smoothness assumption, imposing a uniform smoothness constant across all layers. This is problematic, as *different layers have different geometries*, and thus *should be treated differently*.

But the issue runs much deeper. These algorithms are built for deep learning, where the objective functions are already well known *not* to be smooth (Crawshaw et al., 2022; Zhang et al., 2020). This mismatch has consequences: prior convergence analyses prescribe *tiny constant stepsizes* (see Table 1), uniform across all parameter groups, which bear little resemblance to the tuned learning rates that yield state-of-the-art empirical performance in practice. Consequently, they completely fail to explain why these methods perform so well empirically. In other words, the theory falls short at the one thing it should do best: guiding practical choices, leaving practitioners reliant on costly manual tuning.

Our result in Theorem 4.1 shows this mismatch is *not* inevitable. To better reflect the behavior of deep models, we introduce a more expressive regularity condition: the *layer-wise $(L^0, L^1)$-smoothness*[4]–an extension of the generalized smoothness model of Zhang et al. (2020), applied at the layer level.

**Assumption 2.1** (Layer-wise $(L^0, L^1)$-smoothness)**.** The function $f : \mathcal{S} \mapsto \mathbb{R}$ is layer-wise $(L^0, L^1)$-smooth with constants $L^0 := (L_1^0, \ldots, L_p^0) \in \mathbb{R}_+^p$ and $L^1 :=$

---

[3]Pethick et al. (2025b) introduce two variants of the Scion optimizer: one for constrained optimization, called simply "Scion", and another for unconstrained problems, referred to as "unconstrained Scion". In this work, "Scion" refers to either variant, and "unScion" is used when referring to the unconstrained version.

[4]While we state Assumption 2.1 in this general form, it is worth noting that the proofs do not rely on its full strength. In all cases, we only require the assumption to hold for pairs $X$, $Y$ such that $\|X - Y\| < c$ for some constant $c \geq 0$ (where $\|\cdot\|$ is any norm on $\mathcal{S}$). Specifically, the assumption is only invoked with $X = X^k$, $Y = X^{k+1}$, and since the stepsizes we use are bounded, the distances between consecutive iterates remain bounded as well. For clarity and consistency across results–since the relevant constants vary by theorem–we state the assumption in its stronger, global form, even though the local version suffices for all proofs.

$(L_1^1, \ldots, L_p^1) \in \mathbb{R}_+^p$. That is, the inequality

$$
\begin{aligned}
\|\nabla_i f(X) &- \nabla_i f(Y)\|_{(i)\star} \\
&\leq \left(L_i^0 + L_i^1 \|\nabla_i f(X)\|_{(i)\star}\right) \|X_i - Y_i\|_{(i)}
\end{aligned}
\tag{4}
$$

holds for all $i = 1, \ldots, p$ and all $X = [X_1, \ldots, X_p] \in \mathcal{S}$, $Y = [Y_1, \ldots, Y_p] \in \mathcal{S}$, where $\|\cdot\|_{(i)\star}$ is the dual norm associated with $\|\cdot\|_{(i)}$ (i.e., $\|X_i\|_{(i)\star} := \sup_{\|Z_i\|_{(i)} \leq 1} \langle X_i, Z_i \rangle_{(i)}$ for any $X_i \in \mathcal{S}_i$).

Assumption 2.1 can be viewed as a generalization of the anisotropic "vector" $(L^0, L^1)$–smoothness introduced by Liu et al. (2025b) (now framed in terms of arbitrary norms), which itself is a generalization of the $(L^0, L^1)$–smoothness model of Zhang et al. (2020). As such, our analysis of Gluon goes beyond all existing results, which have only considered the classical $L$-smooth setting. Crucially, however, this is *not* generalization for its own sake–we argue that this is in fact *the right* model for the problem setting at hand. Why? There are (at least) two reasons.

First, unlike classical $L$-smoothness, our formulation *aligns very closely with empirical observations*. In Figures 1a and 1b, we validate Assumption 2.1 in the context of training `NanoGPT` on the `FineWeb` dataset. We plot estimated *trajectory smoothness* $\hat{L}_i[k]$ (defined in (10)) alongside the approximation $\hat{L}_i^{\text{approx}}[k] := L_i^0 + L_i^1 \|\nabla_i f_{\xi^{k+1}}(X^{k+1})\|_{(i)\star}$, where $L_i^0, L_i^1$ are layer-specific parameters estimated from the training run. The figures show these quantities for parameters from the embedding layer and one of the transformer blocks. The close correspondence between $\hat{L}_i[k]$ and $\hat{L}_i^{\text{approx}}[k]$ provides strong evidence that Assumption 2.1 holds approximately along the training trajectory. In Section 5, we further corroborate this finding, showing that our assumption is satisfied *across the entire model architecture* for both the `NanoGPT` language modeling task and a `CNN` trained on `CIFAR-10`. In all cases, we find that $L_i^0 \approx 0$ for all $i$, again highlighting the limitations of classical smoothness. Moreover, as shown in Figure 1c, trajectory smoothness varies substantially across blocks and layers, underscoring the need for per-layer treatment. Complementary experiments using `AdamW` as the optimizer (Figure 10) confirm that this heterogeneity is an intrinsic property of the loss landscape. Together, these results suggest that layer-wise $(L^0, L^1)$-smoothness offers a *significantly more realistic model of the loss landscape in modern deep learning*.

Secondly, Assumption 2.1 not only better captures the geometry of the models, but also *directly informs the design of adaptive and practically effective stepsizes*. In Theorem 4.1, we derive learning rates that reflect the local geometry of each parameter group, guided by our layer-wise smoothness model. As demonstrated in Section 5, our theoretically grounded stepsizes turn out to accurately capture the relative magnitudes of the layer-wise learning rates obtained

by Pethick et al. (2025b) via hyperparameter tuning–a striking validation of our approach, which further highlights the need for layer-wise reasoning. This proves that *refined theoretical models can help effectively guide hyperparameter tuning*.

## 3. Contributions

We present a comprehensive theoretical and empirical study of a broad class of layer-wise LMO-based optimization algorithms. Our key contributions can be summarized as follows:

◇ **A new generalized smoothness framework for neural networks.** We introduce *layer-wise* $(L^0, L^1)$-*smoothness* (Assumption 2.1), a novel non-Euclidean generalized smoothness condition that reflects the anisotropic, layer-wise structure of modern deep networks. This framework extends standard $(L^0, L^1)$-smoothness assumption (Zhang et al., 2020) to arbitrary norms while capturing per-layer variation, offering a *realistic foundation for analyzing deep learning optimizers*.

◇ **First principled analysis of layer-wise methods.** Building on our new assumption, we develop the first faithful convergence analysis for a class of LMO-based algorithms we term Gluon (Algorithms 1 and 2). We recover known algorithms, including state-of-the-art Muon-type optimizers, as special cases (Section 4.1 and Appendix D.1), and pinpoint why earlier theoretical works *fail* to explain the empirical success of these methods (Section 2). In contrast to prior analyses that oversimplify the update rules used in practice, our framework directly aligns with real-world implementations, bridging a critical gap between theory and application.

◇ **Sharper and more general convergence theory.** We develop a convergence theory that extends prior work in both scope and sharpness. In the deterministic case (Algorithm 2), we establish convergence for general non-convex objectives under our Assumption 2.1 (Theorem 4.1), and under the block-wise PŁ condition (Theorem D.4). Unlike earlier analyses, our theory yields *adaptive, layer-wise stepsizes* that can guide real-world training decisions (Section 5). We next analyze the practical stochastic variant with time-varying stepsizes and momentum (Algorithm 1), proving convergence under bounded variance assumption (Theorem 4.1). In both deterministic and stochastic regimes, our guarantees offer *tighter convergence rates* under *more general assumptions* (Table 1), providing the first such results the in non-smooth setting. Moreover, we provide the first theoretical explanation of the benefits of layer-wise learning rates, clearly establishing the advantages of structured, anisotropic optimization in deep learning.

◇ **Empirical evidence.** We validate our theoretical insights

through extensive experiments (Section 5 and Appendix F) in both language modeling (NanoGPT on FineWeb) and image classification (CNN on CIFAR-10). The results confirm that our Assumption 2.1 holds approximately throughout training and demonstrate the practical utility of our theoretically prescribed stepsizes from Theorem 4.1.

## 4. Main Theory and Results

To gain a better intuition into the structure of the updates, we begin with a deterministic formulation of Gluon, formalized in Algorithm 2. At each iteration, the method independently minimizes a linear approximation of $f$ around each parameter group $X_i^k$ within a ball of radius $t_i^k > 0$, ultimately allowing for layer-specific algorithmic design choices.

### 4.1. Examples of Optimizers Within Our Framework

Deterministic Gluon describes a general class of methods, parameterized by the choice of norms $\|\cdot\|_{(i)}$ in the LMO. To illustrate the flexibility of this framework, we highlight several notable special cases (see Appendix D.1 for more details). First, observe that the update rule (12) can be written as

$$
\begin{aligned}
X_i^{k+1} &= X_i^k + t_i^k \mathrm{LMO}_{\{X_i \in \mathcal{S}_i : \|X_i\|_{(i)} \leq 1\}} \left(\nabla_i f(X^k)\right) \\
&= X_i^k + t_i^k \underset{\|X_i\|_{(i)} \leq 1}{\arg\min} \langle \nabla_i f(X^k), X_i \rangle_{(i)}.
\end{aligned}
$$
(5)

For any $X_i \in \mathcal{S}_i = \mathbb{R}^{m_i \times n_i}$, define $\|X_i\|_{\alpha \to \beta} := \sup_{\|z\|_\alpha = 1} \|X_i z\|_\beta$, where $\|\cdot\|_\alpha$ and $\|\cdot\|_\beta$ are some (possibly non-Euclidean) norms on $\mathbb{R}^{n_i}$ and $\mathbb{R}^{m_i}$, respectively. Note that (5) naturally recovers several known updates for specific choices of the layer norms, e.g., layer-wise normalized GD (Yu et al., 2018) for Euclidean norms $\|\cdot\|_{(i)} = \|\cdot\|_2$, and layer-wise signGD (Balles et al., 2020) for max-norms $\|\cdot\|_{(i)} = \|\cdot\|_\infty$. Two special cases are particularly relevant to our analysis:

$\diamond$ **Muon** (Jordan et al., 2024b) when $\|\cdot\|_{(i)} = \|\cdot\|_{2\to 2}$ for all hidden layers.

$\diamond$ **unScion for LLM training** (Pethick et al., 2025b) when $\|\cdot\|_{(i)} = \sqrt{n_i/m_i}\|\cdot\|_{2\to 2}$ for $i = 1, \ldots, p-1$, corresponding to weight matrices of transformer blocks, and $\|\cdot\|_{(p)} = n_p\|\cdot\|_{1\to\infty}$ for the last group $X_p$, representing the embedding and output layers (the two coincide under the weight sharing

regime[5] considered here). In this case, update (5) becomes

$$
X_i^{k+1} = X_i^k - t_i^k \sqrt{\frac{m_i}{n_i}} U_i^k \left(V_i^k\right)^\top, \quad i = 1, \ldots, p-1,
$$

$$
X_p^{k+1} = X_p^k - \frac{t_p^k}{n_p} \mathrm{sign}\left(\nabla_p f(X^k)\right),
$$
(6)

where the matrices $U_i^k, V_i^k$ are obtained from the (reduced) SVD of $\nabla_i f(X^k) = U_i^k \Sigma_i^k \left(V_i^k\right)^\top$.

### 4.2. Convergence Results

Having demonstrated the framework's flexibility through concrete examples, we now state a general convergence result for deterministic Gluon.

**Theorem 4.1.** *Let Assumption 2.1 hold and fix $\varepsilon > 0$. Let $X^0, \ldots, X^{K-1}$ be the iterates of deterministic Gluon (Algorithm 2) run with stepsizes $t_i^k = \frac{\|\nabla_i f(X^k)\|_{(i)\star}}{L_i^0 + L_i^1 \|\nabla_i f(X^k)\|_{(i)\star}}$. Then, to guarantee that*

$$
\min_{k=0,\ldots,K-1} \sum_{i=1}^p \left[ \frac{1/L_i^1}{\frac{1}{p}\sum_{j=1}^p 1/L_j^1} \left\|\nabla_i f(X^k)\right\|_{(i)\star} \right] \leq \varepsilon, \quad (7)
$$

*it suffices to run the algorithm for*

$$
K = \left\lceil \frac{2\Delta^0 \left(\sum_{i=1}^p L_i^0/(L_i^1)^2\right)}{\varepsilon^2 \left(\frac{1}{p}\sum_{j=1}^p 1/L_j^1\right)^2} + \frac{2\Delta^0}{\varepsilon \left(\frac{1}{p}\sum_{j=1}^p 1/L_j^1\right)} \right\rceil \quad (8)
$$

*iterations, where $\Delta^0 := f(X^0) - f^{\mathrm{inf}}$.*

Several important observations follow.

**Convergence rate.** In Appendix D.2, we prove an additional result (Theorem D.1) that modifies the first term in (8) to $2\Delta^0 \sum_{i=1}^p L_i^0/\epsilon^2$, potentially leading to improvements in certain settings (depending on the relationship between the sequences $\{L_i^0\}$ and $\{L_i^1\}$–see Remark D.2). However, this introduces a dependence on $L_{\max}^1 := \max_{i=1,\ldots,p} L_i^1$ in the second term. Empirically, we find that $L_i^0 \approx 0$ across all layers (see Section 5), making the first term vanish in both bounds. In this case, the rate (8) is clearly superior, replacing the worst-case constant $L_{\max}^1$ with the more favorable harmonic mean.

When $p = 1$, our rates match the best-known complexity for finding a stationary point of $(L^0, L^1)$-smooth functions, $\mathcal{O}\left(L^0\Delta^0/\epsilon^2 + L^1\Delta^0/\epsilon\right)$, as established by Vankov et al. (2025) for the Gradient Method. While no prior work has analyzed deterministic Gluon under general $(L^0, L^1)$-smoothness, there exist analyses under classical

---

[5]Weight sharing refers to the practice of using the same parameters (weights) for different parts of a model, rather than allowing each part to have its own unique parameters.

*Table 1.* Comparison of convergence guarantees for Gluon (Algorithms 1 and 2) to achieve $\min_{k=0,\ldots,K-1} \sum_{i=1}^{p} \mathbb{E}[\|\nabla_i f(X^k)\|_{(i)\star}] \leq \varepsilon$, where the $\mathcal{O}(\cdot)$ notation hides logarithmic factors. Notation: $K$ = total number of iterations, $(L^0, L^1)$ = the result holds under layer-wise $(L^0, L^1)$-smoothness, $t_i^k$ = radius/stepsize, $1 - \beta^k$ = momentum.

| Result | Stochastic? | $(L^0, L^1)$ | Rate | Stepsizes $t_i^k$ | $1 - \beta^k$ |
|---|---|---|---|---|---|
| (Kovalev, 2025, Theorem 1) | ✗ | ✗ | $\mathcal{O}\left(\frac{1}{K^{1/2}}\right)$ | const $\propto \frac{1}{K^{1/2}}$ (b) | — |
| (Kovalev, 2025, Theorem 2) | ✓ | ✗ | $\mathcal{O}\left(\frac{1}{K^{1/4}}\right)$ | const $\propto \frac{1}{K^{3/4}}$ (b) | const $\propto \frac{1}{K^{1/2}}$ |
| (Li & Hong, 2025, Theorem 2.1)(a) | ✓ | ✗ | $\mathcal{O}\left(\frac{1}{K^{1/4}}\right)$ | const $\propto \frac{1}{K^{3/4}}$ (b) | const $\propto \frac{1}{K^{1/2}}$ |
| (Pethick et al., 2025b, Lemma 5.4) | ✓ | ✗ | $\mathcal{O}\left(\frac{1}{K^{1/4}}\right)$ | const $\propto \frac{1}{K^{3/4}}$ (b) | $\propto \frac{1}{k^{1/2}}$ |
| **NEW: Theorem 4.1** | ✗ | ✓ | $\mathcal{O}\left(\frac{1}{K^{1/2}}\right)$ | Adaptive | — |
| **NEW: Theorem 4.3** | ✓ | ✓ | $\mathcal{O}\left(\frac{1}{K^{1/4}}\right)$ | $\propto \frac{1}{k^{3/4}}$ | $\propto \frac{1}{k^{1/2}}$ |

(a) Applies only to the Muon/Scion update in (13) with $p = 1$.
(b) These stepsizes are impractically tiny since they have an inverse dependence on the total number of iterations $K$.

$L$-smoothness, treating the parameters as a single vector. The analysis by Kovalev (2025) guarantees convergence in $K = \lceil 6L\Delta^0/\epsilon^2 \rceil$ iterations. The same bound appears in Li & Hong (2025) and Pethick et al. (2025b) (by setting $\sigma^2 = 0$). Since for $p = 1$, $L$-smoothness implies Assumption 2.1 with $L^1 = 0$ (Lemma B.2), our rates match these prior results up to a constant factor. Thus, even in the smooth setting, our bounds are as tight as those derived specifically for it.

However, the real strength of our guarantees lies in their broader applicability. Our analysis is much more general than prior studies, as it extends beyond standard smoothness–allowing $L_i^1 > 0$ introduces additional terms that drive the accelerated convergence enabled by $(L^0, L^1)$-smoothness. This richer model is *essential for explaining the empirical speedup* of methods like Muon, and much more accurately reflects the geometry of neural network loss surfaces. Indeed, as we demonstrate in Section 5, the assumption typically holds with $L_i^0 \approx 0$ and $L_i^1 > 0$.

**Practical radii $t_i^k$.** Unlike previous analyses (Kovalev, 2025; Li & Hong, 2025; Pethick et al., 2025b), which prescribe impractically small constant radii proportional to $\epsilon$, our framework allows $t_i^k$ to be *adaptive* to the loss landscape. Therefore, $t_i^k$ can be larger early in training when $\|\nabla_i f(X^k)\|_{(i)\star}$ is large and gradually shrink as the gradient norm decreases. In the special case when $L_i^0 \approx 0$ (as observed empirically), $t_i^k \approx 1/L_i^1$, which is substantially larger than the radii dictated by earlier analyses.

### 4.3. Stochastic Case

In practice, computing full gradients is often infeasible due to the scale of modern ML problems. We therefore turn to the practical Gluon (Algorithm 1), a stochastic variant of Algorithm 2 that operates with noisy gradient estimates available through a stochastic gradient oracle $\nabla f_\xi, \xi \sim \mathcal{D}$.

**Assumption 4.2.** The stochastic gradient estimator $\nabla f_\xi$ :

$\mathcal{S} \mapsto \mathcal{S}$ is unbiased and has bounded variance. That is, $\mathbb{E}_{\xi\sim\mathcal{D}}[\nabla f_\xi(X)] = \nabla f(X)$ for all $X \in \mathcal{S}$ and there exists $\sigma \geq 0$ such that $\mathbb{E}_{\xi\sim\mathcal{D}}\left[\|\nabla_i f_\xi(X) - \nabla_i f(X)\|_2^2\right] \leq \sigma^2$ for all $X \in \mathcal{S}, i = 1, \ldots, p$.

Note that the choice of norm in Assumption 4.2 is not restrictive: in finite-dimensional spaces, all norms are equivalent, so variance bounds remain valid up to a constant factor when compared to those based on any non-Euclidean norm. The following result establishes the convergence properties.

**Theorem 4.3.** *Let Assumptions 2.1 and 4.2 hold and fix $\varepsilon > 0$. Let $X^0, \ldots, X^{K-1}$ be the iterates of Gluon (Algorithm 1) run with $\beta^k = 1 - (k+1)^{-1/2}$, $t_i^k = t_i(k+1)^{-3/4}$ for some $t_i > 0$, and $M_i^0 = \nabla_i f_{\xi^0}(X^0)$. Then*

$$
\min_{k=0,\ldots,K-1} \sum_{i=1}^{p} \frac{1}{12L_i^1} \mathbb{E}\left[\|\nabla_i f(X^k)\|_{(i)\star}\right]
$$
$$
\lesssim \frac{\Delta^0}{K^{1/4}} + \frac{1}{K^{1/4}} \sum_{i=1}^{p} \left[\frac{\sigma}{L_i^1} + \frac{L_i^0}{(L_i^1)^2}\right], \tag{9}
$$

*where $\Delta^0 := f(X^0) - f^{\inf}$ and the notation $\lesssim$ omits numerical constants and logarithmic factors.*

For $p = 1$, our rate in (9) recovers the complexity for finding a stationary point of $(L^0, L^1)$-smooth functions established by Hübler et al. (2024) for normalized SGD with momentum. When $p \geq 1$, compared to existing guarantees for Gluon, our Theorem 4.3 operates under the significantly more general Assumption 2.1 and uniquely supports training with larger, non-constant stepsizes $t_i^k \propto k^{-3/4}$. In contrast, prior analyses prescribe constant, vanishingly small stepsizes $t_i^k \equiv t_i \propto K^{-3/4}$, tied to the *total* number of iterations $K$ (see Table 1).

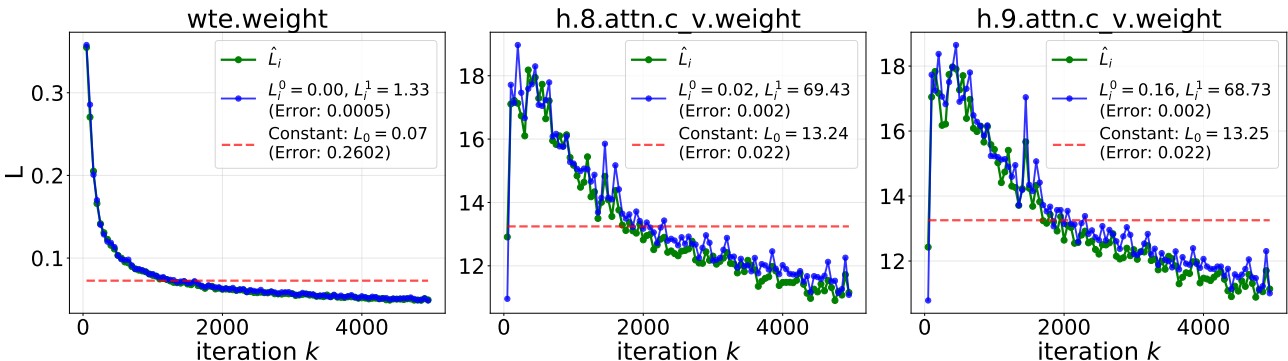

*Figure 2.* **Validation of Assumption 2.1** in `NanoGPT-124M` along training trajectories of unScion. The estimates $\hat{L}_i[k]$ are closely matched by $\hat{L}_i^{\mathrm{approx}}[k]$, while a constant $L_0$ (corresponding to the classical smoothness assumption), results in a substantially poorer fit.

## 5. Experiments

Below, we highlight selected experimental results for the un-Scion optimizer, a special case of Gluon (see Appendix D.1). The primary aim is to verify our layer-wise $(L^0, L^1)$-smoothness model (Assumption 2.1). Additional details and further experiments are provided in Appendix F.[6]

In the first set of experiments, we train the `NanoGPT` model with 124M parameters on the `FineWeb` dataset, leveraging two open-source GitHub repositories (Jordan et al., 2024a; Pethick et al., 2025a). We use the unScion optimizer, i.e., Gluon with the norm choices as in (6). We adopt the hyperparameters from Pethick et al. (2025b, Table 7), mapping their values $\gamma = 0.00036$, $\rho_2 = 50$, and $\rho_3 = 3000$ into our notation as follows: $t_i^k \equiv \gamma\rho_2 = 0.018$ for $i = 1, \ldots, p-1$ (corresponding to the transformer block layers), and $t_p^k \equiv \gamma\rho_3 = 1.08$ (token embeddings and output projections, due to weight sharing). We set the number of warmdown iterations to 0 to keep the learning rates constant throughout training. The model is trained for 5,000 iterations in accordance with the Chinchilla scaling laws. In Figures 1, 5, 6, we plot the estimated *trajectory smoothness* as a function of the iteration index $k$

$$\hat{L}_i[k] := \frac{\|\nabla_i f_{\xi^{k+1}}(X^{k+1}) - \nabla_i f_{\xi^k}(X^k)\|_{(i)\star}}{\|X_i^{k+1} - X_i^k\|_{(i)}} \quad (10)$$

for parameter groups from the embedding layer and 4th and 8th transformer blocks (with similar trends observed across all blocks). We compare this to the approximation

$$\hat{L}_i^{\mathrm{approx}}[k] := L_i^0 + L_i^1 \|\nabla_i f_{\xi^{k+1}}(X^{k+1})\|_{(i)\star},$$

where $L_i^0, L_i^1 \geq 0$ are fitted to minimize the Euclidean error between $\hat{L}_i[k]$ and $\hat{L}_i^{\mathrm{approx}}[k]$, with hinge-like penalty on underestimation (see Appendix F.2). The close alignment between these curves implies that *Assumption 2.1 is approximately satisfied along the training trajectories*.

---

[6]Code for our experiments is available here.

In Appendix F.3.4, we present a gallery of the best and worst per-layer smoothness fits, demonstrating that the layer-wise $(L_i^0, L_i^1)$ model achieves substantially smaller relative error than constant-$L_i$ fits (see Figure 12). This aligns with prior work indicating that classical smoothness does not adequately capture the geometry of deep learning loss landscapes (Zhang et al., 2020). We further summarize the fit quality across all blocks and matrix types in Figure 13.

**Effect of scaling factors.** We next evaluate the impact of the learning rate scaling factors $\rho_2$ and $\rho_3$ on the performance of the unScion optimizer. For consistency, all other hyperparameters are fixed as described earlier. As a baseline, we include results obtained with the AdamW optimizer, using the hyperparameter settings from Section F.3.3. Figure 3 presents (a) validation curves for both optimizers, with varying $\rho_3$ in unScion, and (b) the final validation loss for unScion across different combinations of $\rho_2$ and $\rho_3$. The best performance is achieved with $\rho_2 = 50$ and $\rho_3 = 3000$, i.e., $t_i^k = 0.018$ for $i = 1, \ldots, p-1$ and $t_p^k = 1.08$. This further supports the use of non-uniform scaling across layers, with larger step sizes for the embedding layer, and provides additional evidence in favor of our layer-wise treatment.

**Learning rate transfer from AdamW.** A careful reader may note that our estimates are computed from statistics collected along the training trajectory of Scion, which raises the question of whether the observed phenomena reflect intrinsic properties of the loss landscape or are instead artifacts of a particular optimization trajectory induced by Scion. To further substantiate our claims, we repeat the layer-wise $(L^0, L^1)$-smoothness verification using the same methodology as above, but employing the AdamW optimizer. In Figure 4, we present the results for the estimated trajectory smoothness $\hat{L}_i$ and its approximation $\hat{L}_i^{\mathrm{approx}}$ for several parameter groups along the training trajectory (additional plots are provided in Figure 10, along with an aggregate over all layers in Figure 11). Notably, for the embedding layer parameters $X_p$ (last panel of Figure 4), the

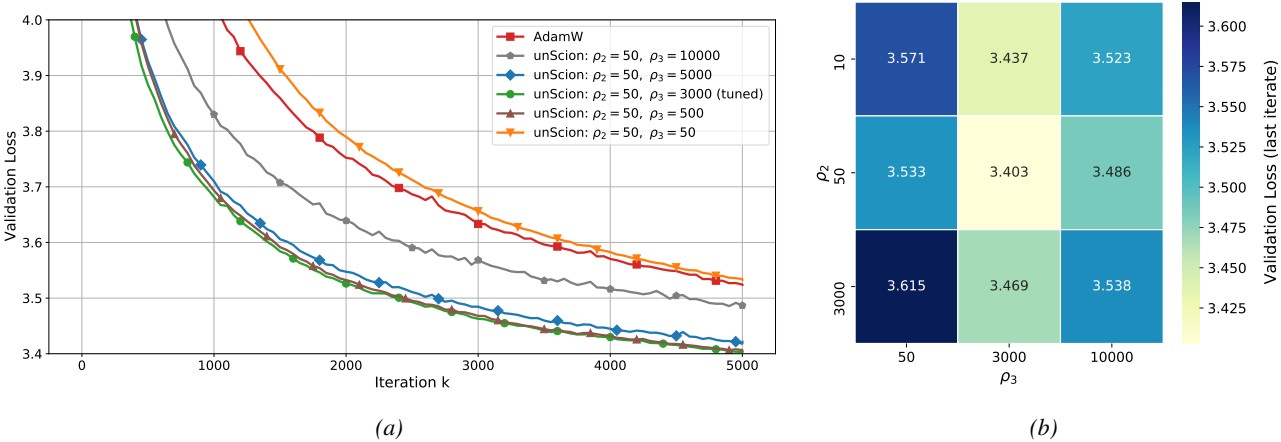

*(a)*                                                                                      *(b)*

*Figure 3.* (a) Validation curves for AdamW and unScion with varying $\rho_3$ values; (b) Heatmap of validation loss from the last iteration of unScion across different combinations of $\rho_2$ and $\rho_3$.

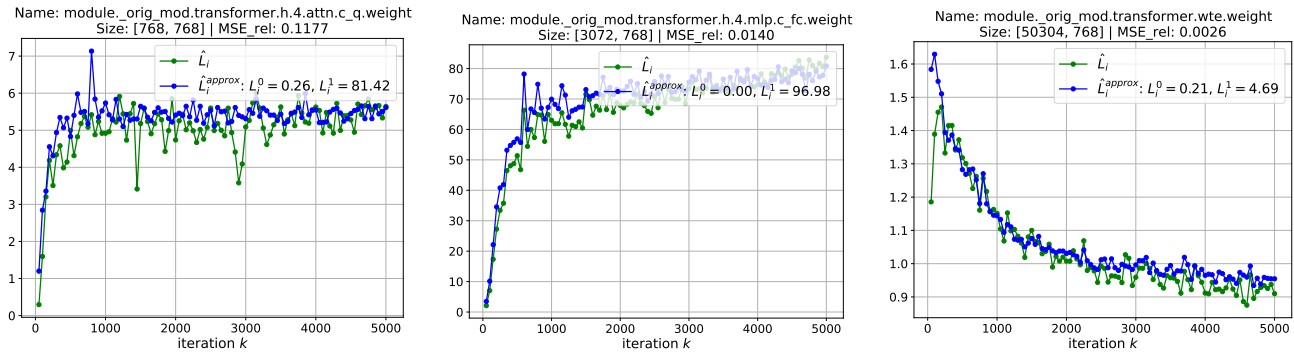

*Figure 4.* Validation of layer-wise $(L^0, L^1)$-smoothness for different groups of parameters in NanoGPT-124M along AdamW training trajectories.

fitted value of $L_p^1$ is approximately 20–30 times smaller than in other groups. Across all plots, we observe that $L_i^0 \ll L_i^1 \|\nabla_i f_{\xi^k}(X^k)\|_{(i)\star}$, implying via Theorem 4.1 that $t_i^k \approx 1/L_i^k$. Thus, $t_i^k$ should be 20–30 times larger than $t_p^k$ for $i = 1, \dots, p - 1$, consistent with the tuned parameters from Pethick et al. (2025b, Table 7).

The fact that our layer-wise $(L^0, L^1)$-smoothness model accurately approximates trajectory smoothness during AdamW training–and that a comparable separation between transformer and embedding layers emerges as in Scion training–provides further support for our claims. Moreover, it suggests that smoothness statistics from AdamW training can guide per-layer learning rate tuning in Scion.

Additional experimental details, conclusions, and practical recommendations are provided in Appendix F.3.3.

### 5.1. Additional Studies

In Appendix F.3.2, we present an ablation study demonstrating that specialized norms provide a better approximation of

trajectory smoothness compared to the standard Euclidean norm. Appendix F.3.5 examines whether the observations made for NanoGPT-124M extend to larger model scales. We report experiments with GPT-2 Medium (∼355M parameters) and GPT-2 Large (∼774M parameters), finding that the phenomena described above persist across all scales. Finally, in Appendix F.4 we validate our smoothness model by training a CNN model on the CIFAR-10 dataset.

## 6. Conclusion and Future Work

In this work, we propose Gluon, an LMO-based optimizer that recovers state-of-the-art optimizers such as Muon and Scion as special cases. We develop a principled analytical framework for layer-wise optimization based on a novel *layer-wise $(L^0, L^1)$-smoothness* assumption, which captures the anisotropic structure of modern deep networks. This assumption enables sharper and more general convergence guarantees and, unlike prior analyses, leads to stepsizes with both theoretical justification and practical relevance. Our framework thus provides *the first rigorous and*

*practically useful analysis of modern layer-wise optimizers.* Experiments confirm that our assumption holds approximately throughout training. Together, these results offer a foundation for structured optimization in deep learning.

While this work resolves two key theoretical gaps (Sections 2.1 and 2.2), it also highlights important directions for future research. Our analysis assumes exact LMO computations, whereas practical implementations use approximations (Appendix F.1). Additionally, our stochastic guarantees (Theorem 4.3) rely on the widely adopted bounded variance assumption, which may not hold in certain scenarios, e.g., under subsampling (Khaled & Richtárik, 2023). Finally, our support for adaptive stepsizes is restricted to the deterministic setting, and a complete theoretical justification in the stochastic regime remains an open challenge.

In summary, although we close the two most critical gaps–establishing a realistic generalized smoothness model and aligning analysis with actual implementations–no single work can exhaust the subject. The field remains open, with many fruitful directions left to pursue.

## Acknowledgements

The research reported in this publication was supported by funding from King Abdullah University of Science and Technology (KAUST): i) KAUST Baseline Research Scheme, ii) CRG Grant ORFS-CRG12-2024-6460, iii) Center of Excellence for Generative AI, under award number 5940, and iv) SDAIA-KAUST Center of Excellence in Artificial Intelligence and Data Science.

## Impact Statement

This paper presents work whose goal is to advance the field of Machine Learning. There are many potential societal consequences of our work, none which we feel must be specifically highlighted here.

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

# Appendix

## A. Related Works

**Generalized Smoothness.** The classical $L$-smoothness assumption, where the gradient is Lipschitz continuous with a global constant $L$, often fails to accurately capture the complex geometry of loss landscapes in deep learning (Crawshaw et al., 2022; Zhang et al., 2020). To address this, Zhang et al. (2020) introduced the $(L^0, L^1)$-smoothness condition, empirically observing in language model experiments that a bound of the form $\|\nabla^2 f(x)\| \leq L_0 + L_1 \|\nabla f(x)\|$ better described the Hessian norm behavior. Subsequent works have analyzed standard optimization algorithms under this generalized smoothness framework. For instance, Gorbunov et al. (2025) and Vankov et al. (2025) provided convergence analyses for the Gradient Method. Hübler et al. (2024) analyzed Normalized SGD with momentum in a parameter-agnostic setting under $(L^0, L^1)$-smoothness. (Yu et al., 2026) proposed non-Euclidean generalized smoothness and established convergence rates for mirror-descent-type methods. Our work extends this line by incorporating $(L^0, L^1)$-smoothness into a *layer-wise* context using arbitrary norms, an approach that is particularly well-suited for the LMO-based optimizers we study.

**Anisotropic Smoothness.** Recognizing the heterogeneous nature of parameters in large models, researchers have explored anisotropic smoothness conditions, where smoothness constants can vary across different dimensions or parameter blocks. Early work in this direction includes coordinate-wise Lipschitz continuity for coordinate descent methods (Nesterov, 2012; Richtárik & Takáč, 2014). More recently, Bernstein et al. (2018) analyzed signSGD under a weaker notion of coordinate-wise smoothness. Crawshaw et al. (2022) further developed this by analyzing Generalized signSGD under a generalized coordinate-wise smoothness assumption, highlighting that different parameter groups can exhibit vastly different geometries. Jiang et al. (2024) focused on Adagrad's analysis under coordinate-wise smoothness and established lower bounds for SGD, underscoring the benefits of adaptivity. Liu et al. (2025b) proposed "Anisotropic $(L^0, L^1)$-smoothness" (a vector version of $(L_0, L_1)$-smoothness applied coordinate-wise) and demonstrated Adagrad's provable advantages over SGD in this setting. Xie et al. (2025) also leveraged anisotropic smoothness concepts in their convergence analysis of Adam. Our work contributes by defining and analyzing *layer-wise $(L^0, L^1)$-smoothness*, which combines the benefits of the generalized smoothness model with a structured, anisotropic perspective tailored to the layer-block architecture of neural networks and compatible with arbitrary layer-specific norms. This framework is essential for understanding LMO-based methods like Muon and Scion.

**LMO-based Optimizers.** The optimizers Muon (Jordan et al., 2024b) and Scion (Pethick et al., 2025b) represent a recent class of methods that have shown strong empirical performance in deep learning. Muon was initially introduced as an effective empirical method, with its update rule for hidden layers inspired by ideas from Bernstein & Newhouse (2024). Subsequently, Pethick et al. (2025b) (authors of Scion) explicitly connected these types of updates to the Frank-Wolfe (FW) framework (Frank & Wolfe, 1956; Jaggi, 2013), proposing the use of layer-specific norms within an LMO-based update rule. These methods perform updates by solving, for each layer, a linear minimization problem over a norm ball centered at the current iterate. Prior theoretical analyses of these optimizers (Kovalev, 2025; Li & Hong, 2025; Pethick et al., 2025b) have relied on standard $L$-smoothness and analyzed a simplified global update. Our work provides the first convergence guarantees for these methods under the more realistic layer-wise $(L_0, L_1)$-smoothness, directly addressing their practical layer-wise nature and leveraging the geometric insights offered by LMOs over general norms.

## B. Auxiliary Lemmas

**Lemma B.1.** *Let $f : \mathcal{S} \mapsto \mathbb{R}$ satisfy Assumption 2.1. Then, for any $X, Y \in \mathcal{S}$, we have*

$$|f(Y) - f(X) - \langle \nabla f(X), Y - X \rangle| \leq \sum_{i=1}^{p} \frac{L_i^0 + L_i^1 \|\nabla_i f(X)\|_{(i)\star}}{2} \|Y_i - X_i\|_{(i)}^2.$$

*Proof.* For all $X, Y \in \mathcal{S}$ we have

$$f(Y) = f(X) + \int_0^1 \langle \nabla f(X + \tau(Y - X)), Y - X \rangle \, d\tau$$

$$= f(X) + \langle \nabla f(X), Y - X \rangle + \int_0^1 \langle \nabla f(X + \tau(Y - X)) - \nabla f(X), Y - X \rangle \, d\tau.$$

Therefore, using the Cauchy-Schwarz inequality and Assumption 2.1, we obtain

$$|f(Y) - f(X) - \langle \nabla f(X), Y - X \rangle|$$

$$\leq \left| \int_0^1 \sum_{i=1}^{p} \langle \nabla_i f(X + \tau(Y - X)) - \nabla_i f(X), Y_i - X_i \rangle_{(i)} \, d\tau \right|$$

$$\leq \int_0^1 \sum_{i=1}^{p} \left| \langle \nabla_i f(X + \tau(Y - X)) - \nabla_i f(X), Y_i - X_i \rangle_{(i)} \right| d\tau$$

$$\leq \int_0^1 \sum_{i=1}^{p} \|\nabla_i f(X + \tau(Y - X)) - \nabla_i f(X)\|_{(i)\star} \|Y_i - X_i\|_{(i)} d\tau$$

$$\leq \int_0^1 \sum_{i=1}^{p} \tau \left( L_i^0 + L_i^1 \|\nabla_i f(X)\|_{(i)\star} \right) \|Y_i - X_i\|_{(i)}^2 d\tau$$

$$= \sum_{i=1}^{p} \frac{L_i^0 + L_i^1 \|\nabla_i f(X)\|_{(i)\star}}{2} \|Y_i - X_i\|_{(i)}^2.$$

$\square$

**Lemma B.2.** *Suppose that $f$ is $L$-smooth with respect to the norm defined in (11), i.e.,*

$$\|\nabla f(X) - \nabla f(Y)\|_{\max \star} \leq L \|X - Y\|_{\max},$$

*where $X = [X_1, \ldots, X_p]$ and $Y = [Y_1, \ldots, Y_p]$ with $X_i, Y_i \in \mathcal{S}_i$. Then Assumption 2.1 holds with $L_i^0 \leq L$ and $L_i^1 = 0$ for all $i = 1, \ldots, p$.*

*Proof.* $L$-smoothness and the definition of the norm give

$$\sum_{i=1}^{p} \|\nabla_i f(X) - \nabla_i f(Y)\|_{(i)\star} \leq L \max \left\{ \|X_1 - Y_1\|_{(1)}, \ldots, \|X_p - Y_p\|_{(p)} \right\}$$

for all $X, Y \in \mathcal{S}$. In particular, choosing $X = [X_1, \ldots, X_p]$ and $Y = [X_1, \ldots, X_{j-1}, Y_j, X_{j+1}, \ldots X_p]$, we have

$$\|\nabla_j f(X) - \nabla_j f(Y)\|_{(j)\star} \leq \sum_{i=1}^{p} \|\nabla_i f(X) - \nabla_i f(Y)\|_{(i)\star} \leq L \|X_j - Y_j\|_{(j)}$$

for any $j \in \{1, \ldots, p\}$, proving the claim. $\square$

**Lemma B.3.** *Suppose that $x_1, \ldots, x_p, y_1, \ldots, y_p \in \mathbb{R}$, $\max_{i \in [p]} |x_i| > 0$ and $z_1, \ldots, z_p > 0$. Then*

$$\sum_{i=1}^{p} \frac{y_i^2}{z_i} \geq \frac{\left( \sum_{i=1}^{p} x_i y_i \right)^2}{\sum_{i=1}^{p} z_i x_i^2}.$$

*Proof.* Cauchy-Schwarz inequality gives

$$\left(\sum_{i=1}^{p} x_i y_i\right)^2 = \left(\sum_{i=1}^{p} \frac{y_i}{\sqrt{z_i}} \sqrt{z_i} x_i\right)^2 \le \left(\sum_{i=1}^{p} \frac{y_i^2}{z_i}\right)\left(\sum_{i=1}^{p} z_i x_i^2\right).$$

Rearranging, we obtain the result. $\square$

**Lemma B.4** (Technical Lemma 10 by Hübler et al. (2024)). *Let $q \in (0,1)$, $p \ge 0$, and $p \ge q$. Further, let $a, b \in \mathbb{N}_{\ge 2}$ with $a \le b$. Then*

$$\sum_{k=a-1}^{b-1} (1+k)^{-p} \prod_{\tau=a-1}^{k} \left(1 - (\tau+1)^{-q}\right) \le (a-1)^{q-p} \exp\left(\frac{a^{1-q} - (a-1)^{1-q}}{1-q}\right).$$

**Lemma B.5** (Technical Lemma 11 by Hübler et al. (2024)). *Let $t > 0$ and for $k \in \mathbb{N}_{\ge 0}$, set $\beta^k = 1 - (k+1)^{-1/2}$, $t^k = t(k+1)^{-3/4}$, $t > 0$. Then, for all $K \in \mathbb{N}_{\ge 1}$ the following inequalities hold:*

*(i)* $\sum_{k=0}^{K-1} t^k \sqrt{\sum_{\tau=0}^{k}(1-\beta^\tau)^2 \prod_{\kappa=\tau+1}^{k}(\beta^\kappa)^2} \le t\left(\frac{7}{2} + \sqrt{2e^2}\log(K)\right)$,

*(ii)* $\sum_{k=0}^{K-1} t^k \sum_{\tau=1}^{k} t^\tau \prod_{\kappa=\tau}^{k} \beta^\kappa \le 7t^2 (3 + \log(K))$.

*Proof.* This is a direct consequence of Lemma 11 by Hübler et al. (2024). To obtain *(ii)*, it suffices to take the limit as $L^1 \to 0$ in statement *(ii)* of part (b). $\square$

# C. Remarks on the Theoretical Results

## C.1. Note on Radii and Stepsizes

It is known (see, e.g., Gruntkowska et al. (2025, Theorem D.1), who establish this for $\mathcal{S} = \mathbb{R}^d$ under Euclidean norms; the extension to general normed vector spaces is entirely analogous) that if $g$ is a convex function, then the solution to the problem

$$\underset{X \in \mathcal{B}^k}{\arg\min} \, g(X)$$

lies on the boundary of the ball $\mathcal{B}^k := \{X \in \mathcal{S} : \|X - X^k\| \leq t^k\}$ (unless $\mathcal{B}^k \cap \arg\min_{X \in \mathcal{S}} g(X) \neq \emptyset$, that is, the ball intersects the set of minimizers of $g$).

This applies directly to the LMO subproblem solved at each iteration of Gluon in (1), since the objective $\langle M_i^k, X_i \rangle_{(i)}$ is linear in $X_i$, and hence convex. In other words, each LMO step moves the iterate from the center of the ball $X_i^k$ to a new point $X_i^{k+1}$ located on the boundary of $\mathcal{B}_i^k$, effectively traversing a distance of $t_i^k$ at each step. For this reason, we use the terms *radius*, *stepsize*, and *learning rate* interchangeably.

## C.2. Note on Prior Analyses

As presented, prior convergence results do not directly apply to the algorithms used in practice. However, there is a workaround. Specifically, some of the existing convergence guarantees (Kovalev, 2025; Pethick et al., 2025b) expressed in terms of the flat vector $x$ are transferable to the structured parameters $X = [X_1, \ldots, X_l] \in \mathcal{S}$ by employing the max-norm (Bernstein & Newhouse, 2025; Large et al., 2024), defined as

$$\|X\|_{\max} := \max\left\{\|X_1\|_{(1)}, \ldots, \|X_p\|_{(p)}\right\}, \tag{11}$$

with corresponding dual norm $\|Y\|_{\max \star} = \sup_{\|X\|_{\max} \leq 1} \langle X, Y \rangle = \sum_{i=1}^p \|Y_i\|_{(i)\star}$. Nevertheless, these works do not make this connection explicit, and an additional layer of analysis is required to ensure the guarantees meaningfully extend to the structured practical setting. Even if such a translation was attempted, the global treatment introduces serious practical limitations. For example, real-world training pipelines tune parameters on a per-layer basis, reflecting the heterogeneous structure of deep networks. Max-norm-based guarantees overlook this variability and offer no mechanism for per-layer control in hyperparameter selection.

---

**Algorithm 2** Deterministic Adaptive Layer-Wise LMO-based Optimizer

---

1: **Input:** Initial model parameters $X^0 = [X_1^0, \ldots, X_p^0] \in \mathcal{S}$
2: **for** $k = 0, 1, \ldots, K - 1$ **do**
3:     **for** $i = 1, 2, \ldots, p$ **do**
4:         Choose adaptive stepsize/radius $t_i^k > 0$ for layer $i$
5:         Update parameters for layer $i$ via LMO over $\mathcal{B}_i^k := \{X_i \in \mathcal{S}_i : \|X_i - X_i^k\|_{(i)} \leq t_i^k\}$:

$$X_i^{k+1} = \text{LMO}_{\mathcal{B}_i^k}\left(\nabla_i f(X^k)\right) := \arg\min_{X_i \in \mathcal{B}_i^k} \langle \nabla_i f(X^k), X_i \rangle_{(i)} \tag{12}$$

6:     **end for**
7:     Update full vector: $X^{k+1} = [X_1^{k+1}, \ldots, X_p^{k+1}]$
8: **end for**

---

# D. Deterministic Case

We begin by considering the deterministic counterpart of Gluon, as formalized in Algorithm 2. We first review several existing algorithms that fall within this framework (Appendix D.1), followed by a proof of Theorem 4.1 (Appendix D.2). Finally, we present an additional convergence guarantee under the layer-wise Polyak-Łojasiewicz (PŁ) condition (Appendix D.3).

## D.1. Special Cases of the LMO Framework

As outlined in Section 4.1, deterministic Gluon encompasses a general class of algorithms, parameterized by the choice of norms $\|\cdot\|_{(i)}$ in the LMO. We now provide a more detailed discussion of the most notable special cases.

**Layer-wise normalized GD (Yu et al., 2018).** Let $\|\cdot\|_{(i)} = \|\cdot\|_{2 \to 2}$ for each parameter group and assume that $n_i = 1$ for all $i = 1, \ldots, p$. In this case, the spectral norm reduces to the standard Euclidean norm $\|\cdot\|_2$, yielding the update rule

$$X_i^{k+1} = X_i^k - t_i^k \frac{\nabla_i f(X^k)}{\|\nabla_i f(X^k)\|_2}, \quad i = 1, \ldots, p,$$

which corresponds to layer-wise normalized GD. With a suitable choice of $t_i^k$ (see Theorem 4.1), the method can also recover the Gradient Method for $(L^0, L^1)$-smooth functions (Vankov et al., 2025).

**Layer-wise signGD (Balles et al., 2020).** Suppose that $\|\cdot\|_{(i)} = \|\cdot\|_{1 \to \infty}$ for each parameter group, with $n_i = 1$ for all $i = 1, \ldots, p$. Then, $\|\cdot\|_{1 \to \infty}$ reduces to $\|\cdot\|_\infty$, and the update becomes

$$X_i^{k+1} = X_i^k - t_i^k \text{sign}\left(\nabla_i f(X^k)\right), \quad i = 1, \ldots, p,$$

where the sign function is applied element-wise. This is equivalent to layer-wise signGD.

**Muon (Jordan et al., 2024b).** Here, the spectral norm $\|\cdot\|_{2 \to 2}$ is used for all parameter groups, without restrictions on $n_i$. In this case, it can be shown that (12) is equivalent to

$$X_i^{k+1} = X_i^k - t_i^k U_i^k \left(V_i^k\right)^\top, \quad i = 1, \ldots, p, \tag{13}$$

where $\nabla_i f(X^k) = U_i^k \Sigma_i^k \left(V_i^k\right)^\top$ is the singular value decomposition (Bernstein & Newhouse, 2024). This is exactly the per-layer deterministic version of the Muon optimizer. In practical LLM training, a more general variant of (13) incorporating stochasticity and momentum is applied to the intermediate layers, while the input and output layers are optimized using other methods.

**Unconstrained Scion (Pethick et al., 2025b).** We can also recover two variants of unScion introduced by Pethick et al. (2025b): one for training LLMs on next-token prediction, and another for training CNNs for image classification.

- **Training LLMs.** Define the norms $\| \cdot \|_{(i)}$ as follows: for $i = 1, \ldots, p - 1$, corresponding to weight matrices of transformer blocks, set $\| \cdot \|_{(i)} = \sqrt{n_i/m_i} \| \cdot \|_{2 \to 2}$, and for the last group $X_p$, representing the embedding and output layers (which coincide under the weight sharing regime considered here), let $\| \cdot \|_{(p)} = n_p \| \cdot \|_{1 \to \infty}$. In this case, (12) becomes

$$
X_i^{k+1} = X_i^k - t_i^k \sqrt{\frac{m_i}{n_i}} U_i^k \left( V_i^k \right)^\top, \quad i = 1, \ldots, p - 1,
$$

$$
X_p^{k+1} = X_p^k - \frac{t_p^k}{n_p} \mathrm{sign} \left( \nabla_p f(X^k) \right),
$$

(14)

where $\nabla_i f(X^k) = U_i^k \Sigma_i^k \left( V_i^k \right)^\top$ is the singular value decomposition. This is equivalent to deterministic layer-wise unScion optimizer without momentum. A more general variant, incorporating stochasticity and momentum and applied to all layers, was shown by Pethick et al. (2025b) to outperform Muon on LLM training tasks.

- **Training CNNs.** The main difference in the CNN setting is the presence of not only 2D weight matrices, but also 1D bias vectors and 4D convolutional kernels parameters. Biases are 1D tensors of shape $\mathbb{R}^{C_i^{out}}$, for which we use scaled Euclidean norms. Convolutional parameters (conv) are 4D tensors with shapes $\mathbb{R}^{C_i^{out} \times C_i^{in} \times k \times k}$, where $C_i^{out}$ and $C_i^{in}$ denote the number of output and input channels, and $k$ is the kernel size. To compute norms, we reshape each 4D tensor to a 2D matrix of shape $\mathbb{R}^{C_i^{out} \times C_i^{in} k^2}$, and then apply a scaled $\| \cdot \|_{2 \to 2}$ norm. This yields the norm choices $\| \cdot \|_{(i)} = \sqrt{1/C_i^{out}} \| \cdot \|_2$ for biases, $\| \cdot \|_{(i)} = k^2 \sqrt{C_i^{in}/C_i^{out}} \| \cdot \|_{2 \to 2}$ for conv, and $\| \cdot \|_{(p)} = n_p \| \cdot \|_{1 \to \infty}$ for the last group $X_p$, associated with classification head weights. Then, it can be shown that (12) is equivalent to

$$
X_i^{k+1} = X_i^k - t_i^k \sqrt{C_i^{out}} \frac{\nabla_i f(X^k)}{\| \nabla_i f(X^k) \|_2}, \qquad \text{(for biases)},
$$

$$
X_i^{k+1} = X_i^k - t_i^k \frac{1}{k^2} \sqrt{\frac{C_i^{out}}{C_i^{in}}} U_i^k \left( V_i^k \right)^\top, \quad \text{(for conv)},
$$

(15)

$$
X_p^{k+1} = X_p^k - \frac{t_p^k}{n_p} \mathrm{sign} \left( \nabla_p f(X^k) \right), \qquad \text{(for head)}
$$

where $\nabla_i f(X^k) = U_i^k \Sigma_i^k \left( V_i^k \right)^\top$ is the singular value decomposition. This corresponds to the deterministic layer-wise unScion optimizer without momentum.

## D.2. Proof of Theorem 4.1

We now state and prove a generalization of Theorem 4.1.

**Theorem D.1.** *Let Assumption 2.1 hold and fix $\varepsilon > 0$. Let $X^0, \ldots, X^{K-1}$ be the iterates of deterministic* Gluon *(Algorithm 2) run with stepsizes $t_i^k = \frac{\| \nabla_i f(X^k) \|_{(i)\star}}{L_i^0 + L_i^1 \| \nabla_i f(X^k) \|_{(i)\star}}$. Then,*

*1. In order to reach the precision*

$$
\min_{k=0,\ldots,K-1} \sum_{i=1}^{p} \left\| \nabla_i f(X^k) \right\|_{(i)\star} \le \epsilon,
$$

*it suffices to run the algorithm for*

$$
K = \left\lceil \frac{2\Delta^0 \sum_{i=1}^{p} L_i^0}{\epsilon^2} + \frac{2\Delta^0 L_{\max}^1}{\epsilon} \right\rceil
$$

(16)

*iterations;*

*2. In order to reach the precision*

$$
\min_{k=0,\ldots,K-1} \sum_{i=1}^{p} \left[ \frac{\frac{1}{L_i^1}}{\frac{1}{p} \sum_{j=1}^{p} \frac{1}{L_j^1}} \left\| \nabla_i f(X^k) \right\|_{(i)\star} \right] \le \varepsilon,
$$

(17)

*it suffices to run the algorithm for*

$$K = \left\lceil \frac{2\Delta^0 \left( \sum_{i=1}^p \frac{L_i^0}{(L_i^1)^2} \right)}{\varepsilon^2 \left( \frac{1}{p} \sum_{j=1}^p \frac{1}{L_j^1} \right)^2} + \frac{2\Delta^0}{\varepsilon \left( \frac{1}{p} \sum_{j=1}^p \frac{1}{L_j^1} \right)} \right\rceil \tag{18}$$

*iterations,*

*where* $\Delta^0 := f(X^0) - \inf_{X \in \mathcal{S}} f(X)$ *and* $L_{\max}^1 := \max_{i=1,\ldots,p} L_i^1$.

*Remark* D.2. Let us compare bounds (16) and (18). Due to the reweighting of the gradient component norms in (17), the rates are not exactly equivalent. Nevertheless, both use weights that sum to $p$, ensuring a fair comparison. Obviously, $\left(1/p \sum_{j=1}^p 1/L_j^1\right)^{-1} \leq L_{\max}^1$, so the second term in (18) is always no worse than its counterpart in (16). The comparison of the first terms, however, depends on how the sequences $\{L_i^0\}$ and $\{L_i^1\}$ relate: if larger values of $L_i^0$'s tend to be attached to smaller values of $L_i^1$, then the first term in (16) improves over that in (18), while for a positive correlation the opposite is true. Indeed, in the extreme case when $L_1^0 \geq \ldots \geq L_p^0$ and $L_1^1 \leq \ldots \leq L_p^1$ (or the reverse ordering), Chebyshev's sum inequality implies that

$$\frac{\sum_{i=1}^p \frac{L_i^0}{(L_i^1)^2}}{\left( \frac{1}{p} \sum_{j=1}^p \frac{1}{L_j^1} \right)^2} \geq \frac{\left( \frac{1}{p} \sum_{i=1}^p \frac{L_i^0}{L_i^1} \right) \left( \frac{1}{p} \sum_{i=1}^p \frac{1}{L_i^1} \right)}{\frac{1}{p} \left( \frac{1}{p} \sum_{j=1}^p \frac{1}{L_j^1} \right)^2} \geq \frac{\left( \frac{1}{p} \sum_{i=1}^p L_i^0 \right) \left( \frac{1}{p} \sum_{i=1}^p \frac{1}{L_i^1} \right)}{\frac{1}{p} \left( \frac{1}{p} \sum_{j=1}^p \frac{1}{L_j^1} \right)} = \sum_{i=1}^p L_i^0.$$

Conversely, if both sequences $\{L_i^0\}$ and $\{L_i^1\}$ are sorted in the same order (either increasing or decreasing), the inequality reverses, and the first term of (18) may be tighter. That said, empirical evidence we provide in Section 5 indicates that in practice $L_i^0 \approx 0$ across all layers, in which case the first terms in (16) and (18) effectively vanish. Then, (18) is clearly superior, replacing the worst-case constant $L_{\max}^1$ by the harmonic mean.

*Proof.* We start with the result obtained in Lemma B.1 with $X = X^k$ and $Y = X^{k+1}$

$$f(X^{k+1}) \leq f(X^k) + \left\langle \nabla f(X^k), X^{k+1} - X^k \right\rangle + \sum_{i=1}^p \frac{L_i^0 + L_i^1 \|\nabla_i f(X^k)\|_{(i)\star}}{2} \|X_i^k - X_i^{k+1}\|_{(i)}^2$$

$$= f(X^k) + \sum_{i=1}^p \left[ \left\langle \nabla_i f(X^k), X_i^{k+1} - X_i^k \right\rangle_{(i)} + \frac{L_i^0 + L_i^1 \|\nabla_i f(X^k)\|_{(i)\star}}{2} \|X_i^k - X_i^{k+1}\|_{(i)}^2 \right].$$

The update rule (12) and the definition of the dual norm $\|\cdot\|_{(i)\star}$ give

$$\|X_i^k - X_i^{k+1}\|_{(i)}^2 \leq \left( t_i^k \right)^2$$

and

$$\begin{aligned}
\left\langle \nabla_i f(X^k), X_i^{k+1} - X_i^k \right\rangle_{(i)} &= \left\langle \nabla_i f(X^k), \mathrm{LMO}_{\mathcal{B}_i^k} \left( \nabla_i f(X^k) \right) - X_i^k \right\rangle_{(i)} \\
&= -t_i^k \max_{\|X_i\|_{(i)} \leq 1} \left\langle \nabla_i f(X^k), X_i \right\rangle_{(i)} \\
&= -t_i^k \|\nabla_i f(X^k)\|_{(i)\star}.
\end{aligned}$$

Consequently,

$$f(X^{k+1}) \leq f(X^k) + \sum_{i=1}^p \left[ -t_i^k \|\nabla_i f(X^k)\|_{(i)\star} + \frac{L_i^0 + L_i^1 \|\nabla_i f(X^k)\|_{(i)\star}}{2} \left( t_i^k \right)^2 \right].$$

Now, choosing

$$t_i^k = \frac{\|\nabla_i f(X^k)\|_{(i)\star}}{L_i^0 + L_i^1 \|\nabla_i f(X^k)\|_{(i)\star}},$$

which minimizes the right-hand side of the last inequality, yields the descent inequality

$$f(X^{k+1}) \leq f(X^k) - \sum_{i=1}^{p} \frac{\|\nabla_i f(X^k)\|_{(i)\star}^2}{2\left(L_i^0 + L_i^1 \|\nabla_i f(X^k)\|_{(i)\star}\right)}. \tag{19}$$

Summing the terms, we obtain

$$\begin{aligned}
\sum_{k=0}^{K-1} \sum_{i=1}^{p} \frac{\|\nabla_i f(X^k)\|_{(i)\star}^2}{2\left(L_i^0 + L_i^1 \|\nabla_i f(X^k)\|_{(i)\star}\right)} &\leq \sum_{k=0}^{K-1} \left(f(X^k) - f(X^{k+1})\right) \\
&= f(X^0) - f(X^K) \\
&\leq f(X^0) - \inf_{X \in \mathcal{S}} f(X) =: \Delta^0.
\end{aligned} \tag{20}$$

Now, the analysis can proceed in two ways:

1. Upper-bounding $L_i^1$ by $L_{\max}^1 := \max_{i=1,\dots,p} L_i^1$ in (20), we obtain

$$\sum_{k=0}^{K-1} \sum_{i=1}^{p} \frac{\|\nabla_i f(X^k)\|_{(i)\star}^2}{2\left(L_i^0 + L_{\max}^1 \|\nabla_i f(X^k)\|_{(i)\star}\right)} \leq \Delta^0. \tag{21}$$

Now, applying Lemma B.3 with $x_i = 1$, $y_i = \|\nabla_i f(X^k)\|_{(i)\star}$ and $z_i = 2\left(L_i^0 + L_{\max}^1 \|\nabla_i f(X^k)\|_{(i)\star}\right)$ gives

$$\begin{aligned}
\phi\left(\sum_{i=1}^{p} \|\nabla_i f(X^k)\|_{(i)\star}\right) &= \frac{\left(\sum_{i=1}^{p} \|\nabla_i f(X^k)\|_{(i)\star}\right)^2}{2\left(\sum_{i=1}^{p} L_i^0 + L_{\max}^1 \sum_{i=1}^{p} \|\nabla_i f(X^k)\|_{(i)\star}\right)} \\
&\leq \sum_{i=1}^{p} \frac{\|\nabla_i f(X^k)\|_{(i)\star}^2}{2\left(L_i^0 + L_{\max}^1 \|\nabla_i f(X^k)\|_{(i)\star}\right)},
\end{aligned}$$

where $\phi(t) := \frac{t^2}{2\left(\sum_{i=1}^{p} L_i^0 + L_{\max}^1 t\right)}$. Combining the last inequality with (21) and using the fact that $\phi$ is increasing, we obtain

$$K\phi\left(\min_{k=0,\dots,K-1} \sum_{i=1}^{p} \|\nabla_i f(X^k)\|_{(i)\star}\right) \leq \sum_{k=0}^{K-1} \phi\left(\sum_{i=1}^{p} \|\nabla_i f(X^k)\|_{(i)\star}\right) \leq \Delta^0, \tag{22}$$

and hence

$$\min_{k=0,\dots,K-1} \sum_{i=1}^{p} \|\nabla_i f(X^k)\|_{(i)\star} \leq \phi^{-1}\left(\frac{\Delta^0}{K}\right),$$

where $\phi^{-1}$ is the inverse function (which exists since $\phi$ is increasing). Therefore, to reach the precision $\min_{k=0,\dots,K-1} \sum_{i=1}^{p} \|\nabla_i f(X^k)\|_{(i)\star} \leq \epsilon$, it is sufficient to choose the number of iterations to be

$$K = \left\lceil \frac{\Delta^0}{\phi(\epsilon)} \right\rceil = \left\lceil \frac{2\sum_{i=1}^{p} L_i^0 \Delta^0}{\epsilon^2} + \frac{2L_{\max}^1 \Delta^0}{\epsilon} \right\rceil.$$

2. Alternatively, we can start from the inequality (20) and apply Lemma B.3 with $x_i = 1/L_i^1$, $y_i = \left\|\nabla_i f(X^k)\right\|_{(i)\star}$ and $z_i = 2(L_i^0 + L_i^1 \left\|\nabla_i f(X^k)\right\|_{(i)\star})$ to obtain

$$
\begin{aligned}
\Delta^0 &\geq \sum_{k=0}^{K-1} \sum_{i=1}^{p} \frac{\|\nabla_i f(X^k)\|_{(i)\star}^2}{2\left(L_i^0 + L_i^1 \|\nabla_i f(X^k)\|_{(i)\star}\right)} \\
&\geq \sum_{k=0}^{K-1} \frac{\left(\sum_{i=1}^{p} \frac{1}{L_i^1} \left\|\nabla_i f(X^k)\right\|_{(i)\star}\right)^2}{2\left(\sum_{i=1}^{p} \frac{1}{(L_i^1)^2}\left(L_i^0 + L_i^1 \left\|\nabla_i f(X^k)\right\|_{(i)\star}\right)\right)} \\
&= \sum_{k=0}^{K-1} \frac{\left(\sum_{i=1}^{p} \frac{1}{L_i^1} \left\|\nabla_i f(X^k)\right\|_{(i)\star}\right)^2}{2\left(\sum_{i=1}^{p} \frac{L_i^0}{(L_i^1)^2} + \sum_{i=1}^{p} \frac{1}{L_i^1} \left\|\nabla_i f(X^k)\right\|_{(i)\star}\right)} \\
&= \sum_{t=0}^{K-1} \psi\left(\sum_{i=1}^{p} \frac{1}{L_i^1} \left\|\nabla_i f(X^k)\right\|_{(i)\star}\right),
\end{aligned}
$$

where $\psi(t) := \frac{t^2}{2\left(\sum_{i=1}^{p} \frac{L_i^0}{(L_i^1)^2}+t\right)}$. Since the function $\psi$ is increasing for $t > 0$, $\psi^{-1}$ exists. It follows that

$$
\begin{aligned}
\Delta^0 &\geq \sum_{k=0}^{K-1} \psi\left(\sum_{i=1}^{p} \frac{1}{L_i^1} \left\|\nabla_i f(X^k)\right\|_{(i)\star}\right) \\
&\geq K\psi\left(\min_{k=0,\ldots,K-1} \sum_{i=1}^{p} \frac{1}{L_i^1} \left\|\nabla_i f(X^k)\right\|_{(i)\star}\right),
\end{aligned}
$$

and hence

$$
\min_{k=0,\ldots,K-1} \sum_{i=1}^{p} \frac{1}{L_i^1} \left\|\nabla_i f(X^k)\right\|_{(i)\star} \leq \psi^{-1}\left(\frac{\Delta^0}{K}\right).
$$

This in turn means that to reach the precision

$$
\min_{k=0,\ldots,K-1} \sum_{i=1}^{p} \left[\frac{\frac{1}{L_i^1}}{\frac{1}{p}\sum_{j=1}^{p}\frac{1}{L_j^1}} \left\|\nabla_i f(X^k)\right\|_{(i)\star}\right] \leq \varepsilon,
$$

it suffices to run the algorithm for

$$
K = \left\lceil \frac{\Delta^0}{\psi\left(\varepsilon\left(\frac{1}{p}\sum_{j=1}^{p}\frac{1}{L_j^1}\right)\right)} \right\rceil = \left\lceil \frac{2\Delta^0\left(\sum_{i=1}^{p}\frac{L_i^0}{(L_i^1)^2}\right)}{\varepsilon^2\left(\frac{1}{p}\sum_{j=1}^{p}\frac{1}{L_j^1}\right)^2} + \frac{2\Delta^0}{\varepsilon\left(\frac{1}{p}\sum_{j=1}^{p}\frac{1}{L_j^1}\right)} \right\rceil
$$

iterations.

$\square$

## D.3. Convergence under the PŁ Condition

We now establish convergence rates under the layer-wise Polyak–Łojasiewicz (PŁ) condition, introduced in Assumption D.3. This property is especially relevant for heavily over-parameterized neural networks, as it has been shown to capture the properties of their loss landscapes (Liu et al., 2022).

**Assumption D.3** (Layer-wise Polyak-Łojasiewicz condition)**.** The function $f : \mathcal{S} \mapsto \mathbb{R}$ satisfies the layer-wise Polyak-Łojasiewicz (PŁ) condition with a constant $\mu > 0$, i.e., for any $X \in \mathcal{S}$

$$
\sum_{i=1}^{p} \|\nabla_i f(X)\|_{(i)\star}^2 \geq 2\mu\left(f(X) - f^\star\right),
$$

where $f^\star := \inf_{X \in \mathcal{S}} f(X) > -\infty$.

Assumption D.3 reduces to the standard PŁ condition (Karimi et al., 2016) by vectorizing the parameters and adopting the Euclidean norm $\|\cdot\|_2$.

**Theorem D.4.** *Let Assumptions 2.1 and D.3 hold and fix $\varepsilon > 0$. Let $X^0, \dots, X^{K-1}$ be the iterates of deterministic* Gluon *(Algorithm 2) run with* $t_i^k = \frac{\|\nabla_i f(X^k)\|_{(i)\star}}{L_i^0 + L_i^1 \|\nabla_i f(X^k)\|_{(i)\star}}$.

  1. *If $L_i^1 \geq 0$, then to reach the precision $\min_{k=0,\dots,K-1} f(X^k) - f^\star \leq \epsilon$, it suffices to run the algorithm for*

$$K = \left\lceil \frac{\sum_{i=1}^p L_i^0 \Delta^0}{\mu\epsilon} + \frac{\sqrt{2} L_{\max}^1 \Delta^0}{\sqrt{\mu\epsilon}} \right\rceil$$

  *iterations,*

  2. *If $L_i^1 = 0$ for all $i = 1, \dots, p$, then to reach the precision $f(X^K) - f^\star \leq \epsilon$, it suffices to run the algorithm for*

$$K = \left\lceil \frac{L_{\max}^0}{\mu} \log \frac{\Delta^0}{\epsilon} \right\rceil,$$

*where $L_{\max}^0 := \max_{i=1,\dots,p} L_i^0$, $L_{\max}^1 := \max_{i=1,\dots,p} L_i^1$, $\Delta^0 := f(X^0) - f^\star$ and $f^\star := \inf_{X \in \mathcal{S}} f(X)$.*

*Proof.* We consider two scenarios: (1) the general case with arbitrary $L_i^1 \geq 0$ and (2) $L_i^1 = 0$ for all $i = 1, \dots, p$.

**Case 1: $L_i^1 \geq 0$.** We start by following the same steps as in the proof of Theorem 4.1. From (22), we have

$$\sum_{k=0}^{K-1} \phi \left( \sum_{i=1}^p \|\nabla_i f(X^k)\|_{(i)\star} \right) \leq \Delta^0,$$

where $\phi(t) := \frac{t^2}{2(\sum_{i=1}^p L_i^0 + L_{\max}^1 t)}$. Now, using Assumption D.3, we get

$$\left( \sum_{i=1}^p \|\nabla_i f(X^k)\|_{(i)\star} \right)^2 \geq \sum_{i=1}^p \|\nabla_i f(X^k)\|_{(i)\star}^2 \geq 2\mu \left( f(X^k) - f^\star \right).$$

Consequently, since $\phi$ is an increasing function,

$$K\phi \left( \sqrt{2\mu} \sqrt{f(X^{k^\star}) - f^\star} \right) \leq \sum_{k=0}^{K-1} \phi \left( \sqrt{2\mu} \sqrt{f(X^k) - f^\star} \right)$$

$$\leq \sum_{k=0}^{K-1} \phi \left( \sum_{i=1}^p \|\nabla_i f(X^k)\|_{(i)\star} \right) \leq \Delta^0,$$

where $k^\star := \operatorname{argmin}_{k=0,\dots,K-1} f(X^k) - f^\star$. Denoting the corresponding inverse function (which exists since $\phi$ is increasing) by $\phi^{-1}$, it follows that

$$\sqrt{2\mu} \sqrt{f(X^{k^\star}) - f^\star} \leq \phi^{-1} \left( \frac{\Delta^0}{K} \right) \leq \sqrt{2\mu\epsilon}.$$

Therefore, to reach the precision $f(X^{k^\star}) - f^\star \leq \epsilon$, it is sufficient to choose the number of iterations

$$K = \left\lceil \frac{\Delta^0}{\phi \left( \sqrt{2\mu\epsilon} \right)} \right\rceil = \left\lceil \frac{\sum_{i=1}^p L_i^0 \Delta^0}{\mu\epsilon} + \frac{\sqrt{2} L_{\max}^1 \Delta^0}{\sqrt{\mu\epsilon}} \right\rceil.$$

**Case 2: $L_i^1 = 0$.** Inequality (19) from the proof of Theorem 4.1 with $L_i^1 = 0$ gives

$$f(X^{k+1}) \le f(X^k) - \sum_{i=1}^p \frac{\|\nabla_i f(X^k)\|_{(i)\star}^2}{2L_i^0}.$$

Using the fact that

$$\sum_{i=1}^p \frac{\|\nabla_i f(X^k)\|_{(i)\star}^2}{2L_i^0} \ge \min_{j=1,\dots,p} \frac{1}{2L_j^0} \sum_{i=1}^p \|\nabla_i f(X^k)\|_{(i)\star}^2 = \frac{1}{2\max_{j=1,\dots,p} L_j^0} \sum_{i=1}^p \|\nabla f(X^k)\|_{(i)\star}^2$$

along with Assumption D.3, we obtain

$$f(X^{k+1}) \le f(X^k) - \frac{\mu}{L_{\max}^0} \left(f(X^k) - f^\star\right).$$

The remaining part of the proof follows from the simple observation

$$\log\left(\frac{\Delta_0}{\epsilon}\right) \le k \frac{\mu}{L_{\max}^0} \le k \log\left(\frac{1}{1 - \frac{\mu}{L_{\max}^0}}\right).$$

$\square$

# E. Stochastic Case

## E.1. Adaptive Stepsizes

Before proving the main result from Section 4.3, we first present an attempt to formulate an adaptive stepsize strategy for the stochastic setting. This requires the following assumption:

**Assumption E.1.** The stochastic gradient estimator $\nabla f_\xi : \mathcal{S} \mapsto \mathcal{S}$ is unbiased and has bounded relative variance. That is, $\mathbb{E}[\nabla f_\xi(X)] = \nabla f(X)$ for all $X \in \mathcal{S}$ and there exists $0 \le \zeta < 1$ such that

$$\|\nabla_i f_\xi(X) - \nabla_i f(X)\|_{(i)\star} \le \zeta \|\nabla_i f_\xi(X)\|_{(i)\star}, \quad i = 1, \dots, p$$

holds almost surely for all $X \in \mathcal{S}$.

This assumption is somewhat unconventional due to the presence of the stochastic gradients on the right-hand side of the inequality. It does not follow from standard conditions and does not fall within known frameworks for modeling stochasticity, such as the ABC inequality of Khaled & Richtárik (2023). Instead, it introduces a novel structure with parallels to the literature on contractive compression (Beznosikov et al., 2023; Demidovich et al., 2023).

To elaborate, recall the definition of a contractive compressor:

**Definition E.2** (Contractive compressor). A stochastic mapping $\mathcal{C} : \mathcal{S} \to \mathcal{S}$ is called a *contractive compressor* if there exists $\alpha \in [0, 1)$ such that

$$\mathbb{E}\left[\|\mathcal{C}(X) - X\|^2\right] \le (1 - \alpha)\|X\|^2 \tag{23}$$

for any $X \in \mathcal{S}$.

There is a conceptual similarity between Assumption E.1 and the contractive property in (23). Assumption E.1 can be interpreted as asserting that the true gradient $\nabla f(X)$ is effectively a contraction of the stochastic gradient $\nabla f_\xi(X)$, with contraction factor $1 - \zeta$. Unlike contractive compressors, there is no explicit mapping from $\nabla f_\xi(X)$ to $\nabla f(X)$, and the uniform bound implies the same contraction-like behavior across all stochastic gradients.

Although Assumption E.1 is admittedly strong, it allows us to establish a convergence theorem using an adaptive stepsize strategy similar to the one employed in the deterministic case in Theorem D.1.

**Theorem E.3.** *Let Assumptions 2.1 and E.1 hold and fix $\varepsilon > 0$. Let $X^0, \dots, X^{K-1}$ be the iterates of* Gluon *(Algorithm 1) run with $\beta^k = 0$ and $t_i^k = \frac{(1-\zeta)\|\nabla_i f_{\xi^k}(X^k)\|_{(i)\star}}{L_i^0 + (1+\zeta)L_i^1\|\nabla_i f_{\xi^k}(X^k)\|_{(i)\star}}$. Then,*

1. *In order to reach the precision*

$$\min_{k=0,\dots,K-1} \sum_{i=1}^p \mathbb{E}\left[\|\nabla_i f(X^k)\|_{(i)\star}\right] \le \epsilon,$$

   *it suffices to run the algorithm for*

$$K = \left\lceil \frac{2\sum_{i=1}^p L_i^0 \Delta^0}{(1-\zeta)^2 \epsilon^2} + \frac{2(1+\zeta)L_{\max}^1 \Delta^0}{(1-\zeta)^2 \epsilon} \right\rceil$$

   *iterations.*

2. *In order to reach the precision*

$$\min_{k=0,\dots,K-1} \sum_{i=1}^p \left[\frac{\frac{1}{L_i^1}}{\frac{1}{p}\sum_{j=1}^p \frac{1}{L_j^1}} \|\nabla_i f(X^k)\|_{(i)\star}\right] \le \varepsilon,$$

   *it suffices to run the algorithm for*

$$K = \left\lceil \frac{2\Delta^0 \sum_{i=1}^p \frac{L_i^0}{(L_i^1)^2}}{\varepsilon^2(1-\zeta)^2\left(\frac{1}{p}\sum_{j=1}^p \frac{1}{L_j^1}\right)^2} + \frac{2\Delta^0(1+\zeta)}{\varepsilon(1-\zeta)^2\left(\frac{1}{p}\sum_{j=1}^p \frac{1}{L_j^1}\right)} \right\rceil$$

   *iterations,*

*where* $\Delta^0 := f(X^0) - \inf_{X \in \mathcal{S}} f(X)$ *and* $L^1_{\max} := \max_{i=1,\ldots,p} L^1_i$.

*Proof.* Lemma B.1 with $X = X^k$ and $Y = X^{k+1}$ gives

$$
\begin{aligned}
&f(X^{k+1}) \\
&\leq f(X^k) + \left\langle \nabla f(X^k), X^{k+1} - X^k \right\rangle + \sum_{i=1}^{p} \frac{L^0_i + L^1_i \|\nabla_i f(X^k)\|_{(i)\star}}{2} \|X^k_i - X^{k+1}_i\|^2_{(i)} \\
&= f(X^k) + \sum_{i=1}^{p} \left[ \left\langle \nabla_i f(X^k), X^{k+1}_i - X^k_i \right\rangle_{(i)} + \frac{L^0_i + L^1_i \|\nabla_i f(X^k)\|_{(i)\star}}{2} \|X^k_i - X^{k+1}_i\|^2_{(i)} \right] \\
&= f(X^k) + \sum_{i=1}^{p} \left[ \left\langle \nabla_i f_{\xi^k}(X^k), X^{k+1}_i - X^k_i \right\rangle_{(i)} + \left\langle \nabla_i f(X^k) - \nabla_i f_{\xi^k}(X^k), X^{k+1}_i - X^k_i \right\rangle_{(i)} \right] \\
&\quad + \sum_{i=1}^{p} \frac{L^0_i + L^1_i \|\nabla_i f(X^k)\|_{(i)\star}}{2} \|X^k_i - X^{k+1}_i\|^2_{(i)},
\end{aligned}
$$

and applying the Cauchy-Schwarz inequality, we get

$$
\begin{aligned}
f(X^{k+1}) \leq f(X^k) + \sum_{i=1}^{p} \Bigg[ &\left\langle \nabla_i f_{\xi^k}(X^k), X^{k+1}_i - X^k_i \right\rangle_{(i)} \\
&+ \|\nabla_i f(X^k) - \nabla_i f_{\xi^k}(X^k)\|_{(i)\star} \|X^{k+1}_i - X^k_i\|_{(i)} \\
&+ \frac{L^0_i + L^1_i \|\nabla_i f(X^k)\|_{(i)\star}}{2} \|X^k_i - X^{k+1}_i\|^2_{(i)} \Bigg].
\end{aligned}
$$

The update rule (1) and the definition of the dual norm $\|\cdot\|_{(i)\star}$ give

$$
\|X^k_i - X^{k+1}_i\|^2_{(i)} \leq \left(t^k_i\right)^2
$$

and

$$
\begin{aligned}
\left\langle \nabla_i f_{\xi^k}(X^k), X^{k+1}_i - X^k_i \right\rangle_{(i)} &= \left\langle \nabla_i f_{\xi^k}(X^k), \mathrm{LMO}_{\mathcal{B}^k_i}\left(\nabla_i f_{\xi^k}(X^k)\right) - X^k_i \right\rangle_{(i)} \\
&= -t^k_i \max_{\|X_i\|_{(i)} \leq 1} \left\langle \nabla_i f_{\xi^k}(X^k), X_i \right\rangle_{(i)} \\
&= -t^k_i \|\nabla_i f_{\xi^k}(X^k)\|_{(i)\star}.
\end{aligned}
$$

Consequently, using Assumption E.1, we obtain

$$
\begin{aligned}
f(X^{k+1}) &\leq f(X^k) + \sum_{i=1}^{p} \Bigg[ -t^k_i \|\nabla_i f_{\xi^k}(X^k)\|_{(i)\star} + t^k_i \|\nabla_i f(X^k) - \nabla_i f_{\xi^k}(X^k)\|_{(i)\star} \\
&\qquad\qquad\qquad + \frac{L^0_i + L^1_i \|\nabla_i f(X^k)\|_{(i)\star}}{2} \left(t^k_i\right)^2 \Bigg] \\
&\leq f(X^k) + \sum_{i=1}^{p} \Bigg[ -(1-\zeta)t^k_i \|\nabla_i f_{\xi^k}(X^k)\|_{(i)\star} \\
&\qquad\qquad\qquad + \frac{L^0_i + (1+\zeta)L^1_i \|\nabla_i f_{\xi^k}(X^k)\|_{(i)\star}}{2} \left(t^k_i\right)^2 \Bigg].
\end{aligned}
$$

Minimizing the right-hand side of the last inequality with respect to $t^k_i$ yields

$$
t^k_i = \frac{(1-\zeta)\|\nabla_i f_{\xi^k}(X^k)\|_{(i)\star}}{L^0_i + (1+\zeta)L^1_i \|\nabla_i f_{\xi^k}(X^k)\|_{(i)\star}}.
$$

This greedy approach for choosing $t_i^k$ gives the descent inequality

$$f(X^{k+1}) \le f(X^k) - \sum_{i=1}^{p} \frac{(1-\zeta)^2 \|\nabla_i f_{\xi^k}(X^k)\|_{(i)\star}^2}{2\left(L_i^0 + (1+\zeta)L_i^1 \|\nabla_i f_{\xi^k}(X^k)\|_{(i)\star}\right)}.$$

Taking expectations, we have

$$\mathbb{E}[f(X^{k+1})] \le \mathbb{E}[f(X^k)] - \sum_{i=1}^{p} \mathbb{E}\left[\frac{(1-\zeta)^2 \|\nabla_i f_{\xi^k}(X^k)\|_{(i)\star}^2}{2\left(L_i^0 + (1+\zeta)L_i^1 \|\nabla_i f_{\xi^k}(X^k)\|_{(i)\star}\right)}\right]. \tag{24}$$

Now, let us define the function $\phi_i(t) := \frac{(1-\zeta)^2 t^2}{2(L_i^0 + (1+\zeta)L_i^1 t)}$. Since $\phi_i(t)$ is convex, Jensen's inequality gives

$$\mathbb{E}[f(X^k)] - \mathbb{E}[f(X^{k+1})] \ge \sum_{i=1}^{p} \mathbb{E}\left[\frac{(1-\zeta)^2 \|\nabla_i f_{\xi^k}(X^k)\|_{(i)\star}^2}{2\left(L_i^0 + (1+\zeta)L_i^1 \|\nabla_i f_{\xi^k}(X^k)\|_{(i)\star}\right)}\right]$$

$$\ge \sum_{i=1}^{p} \frac{(1-\zeta)^2 \left(\mathbb{E}\left[\|\nabla_i f_{\xi^k}(X^k)\|_{(i)\star}\right]\right)^2}{2\left(L_i^0 + (1+\zeta)L_i^1 \mathbb{E}\left[\|\nabla_i f_{\xi^k}(X^k)\|_{(i)\star}\right]\right)}.$$

By Jensen's inequality and Assumption E.1

$$\mathbb{E}\left[\|\nabla_i f(X^k)\|_{(i)\star}\right] = \mathbb{E}\left[\left\|\mathbb{E}\left[\nabla_i f_{\xi_k}(X^k)\,\middle|\, X^k\right]\right\|_{(i)\star}\right]$$

$$\le \mathbb{E}\left[\mathbb{E}\left[\|\nabla_i f_{\xi_k}(X^k)\|_{(i)\star}\,\middle|\, X^k\right]\right]$$

$$= \mathbb{E}\left[\|\nabla_i f_{\xi_k}(X^k)\|_{(i)\star}\right],$$

and hence, using the fact that $\phi_i$ is increasing, we get

$$\mathbb{E}[f(X^k)] - \mathbb{E}[f(X^{k+1})] \ge \sum_{i=1}^{p} \frac{(1-\zeta)^2 \left(\mathbb{E}\left[\|\nabla_i f(X^k)\|_{(i)\star}\right]\right)^2}{2\left(L_i^0 + (1+\zeta)L_i^1 \mathbb{E}\left[\|\nabla_i f(X^k)\|_{(i)\star}\right]\right)}.$$

Summing the terms gives

$$\sum_{k=0}^{K-1} \sum_{i=1}^{p} \frac{(1-\zeta)^2 \left(\mathbb{E}\left[\|\nabla_i f(X^k)\|_{(i)\star}\right]\right)^2}{2\left(L_i^0 + (1+\zeta)L_i^1 \mathbb{E}\left[\|\nabla_i f(X^k)\|_{(i)\star}\right]\right)} \le \sum_{k=0}^{K-1} \left(\mathbb{E}[f(X^k)] - \mathbb{E}[f(X^{k+1})]\right)$$

$$= \mathbb{E}[f(X^0)] - \mathbb{E}[f(X^K)] \tag{25}$$

$$\le f(X^0) - \inf_{X \in \mathcal{S}} f(X) =: \Delta^0,$$

The remaining part of the proof closely follows the proof of Theorem D.1. We can proceed in two ways:

1. Upper-bounding $L_i^1$ by $L_{\max}^1 := \max_{i=1,\dots,p} L_i^1$ in (25), we obtain

$$\sum_{k=0}^{K-1} \sum_{i=1}^{p} \frac{(1-\zeta)^2 \left(\mathbb{E}\left[\|\nabla_i f(X^k)\|_{(i)\star}\right]\right)^2}{2\left(L_i^0 + (1+\zeta)L_{\max}^1 \mathbb{E}\left[\|\nabla_i f(X^k)\|_{(i)\star}\right]\right)} \le \Delta^0. \tag{26}$$

Now, Lemma B.3 with $x_i = 1$, $y_i = (1-\zeta)\mathbb{E}\left[\|\nabla_i f(X^k)\|_{(i)\star}\right]$ and $z_i = 2\left(L_i^0 + (1+\zeta)L_{\max}^1 \mathbb{E}\left[\|\nabla_i f(X^k)\|_{(i)\star}\right]\right)$ gives

$$\phi\left(\sum_{i=1}^{p} \mathbb{E}\left[\|\nabla_i f(X^k)\|_{(i)\star}\right]\right) = \frac{\left((1-\zeta)\sum_{i=1}^{p} \mathbb{E}\left[\|\nabla_i f(X^k)\|_{(i)\star}\right]\right)^2}{2\sum_{i=1}^{p}\left(L_i^0 + (1+\zeta)L_{\max}^1 \mathbb{E}\left[\|\nabla_i f(X^k)\|_{(i)\star}\right]\right)}$$

$$\le \sum_{i=1}^{p} \frac{(1-\zeta)^2 \mathbb{E}\left[\|\nabla_i f(X^k)\|_{(i)\star}\right]^2}{2\left(L_i^0 + (1+\zeta)L_{\max}^1 \mathbb{E}\left[\|\nabla_i f(X^k)\|_{(i)\star}\right]\right)}$$

where $\phi(t) := \frac{(1-\zeta)^2 t^2}{2(\sum_{i=1}^p L_i^0 + (1+\zeta)L_{\max}^1 t)}$. Combining the last inequality with (26) and using the fact that $\phi$ is increasing, we get

$$K\phi\left(\min_{k=0,\ldots,K-1}\sum_{i=1}^p \mathbb{E}\left[\|\nabla_i f(X^k)\|_{(i)\star}\right]\right) \le \sum_{k=0}^{K-1}\phi\left(\sum_{i=1}^p \mathbb{E}\left[\|\nabla_i f(X^k)\|_{(i)\star}\right]\right) \le \Delta^0.$$

and hence

$$\min_{k=0,\ldots,K-1}\sum_{i=1}^p \mathbb{E}\left[\|\nabla_i f(X^k)\|_{(i)\star}\right] \le \phi^{-1}\left(\frac{\Delta^0}{K}\right),$$

where $\phi^{-1}$ denotes the inverse function (which exists since $\phi$ is increasing). Therefore, to reach the precision $\min_{k=0,\ldots,K-1}\sum_{i=1}^p \mathbb{E}\left[\|\nabla_i f(X^k)\|_{(i)\star}\right] \le \epsilon$, it suffices to run the algorithm for

$$K = \left\lceil\frac{\Delta^0}{\phi(\epsilon)}\right\rceil = \left\lceil\frac{2\Delta^0 \sum_{i=1}^p L_i^0}{(1-\zeta)^2\epsilon^2} + \frac{2\Delta^0(1+\zeta)L_{\max}^1}{(1-\zeta)^2\epsilon}\right\rceil$$

iterations.

2. Alternatively, we can start from inequality (25) and apply Lemma B.3 with $x_i = 1/L_i^1$, $y_i = (1-\zeta)\mathbb{E}\left[\|\nabla_i f(X^k)\|_{(i)\star}\right]$ and $z_i = 2\left(L_i^0 + (1+\zeta)L_i^1\mathbb{E}\left[\|\nabla_i f(X^k)\|_{(i)\star}\right]\right)$ to obtain

$$\begin{aligned}
\Delta^0 &\ge \sum_{k=0}^{K-1}\sum_{i=1}^p \frac{(1-\zeta)^2\mathbb{E}\left[\|\nabla_i f(X^k)\|_{(i)\star}\right]^2}{2\left(L_i^0 + (1+\zeta)L_i^1\mathbb{E}\left[\|\nabla_i f(X^k)\|_{(i)\star}\right]\right)} \\
&\ge \sum_{k=0}^{K-1}\frac{\left(\sum_{i=1}^p \frac{1}{L_i^1}(1-\zeta)\mathbb{E}\left[\|\nabla_i f(X^k)\|_{(i)\star}\right]\right)^2}{2\sum_{i=1}^p\left(\frac{L_i^0}{(L_i^1)^2} + (1+\zeta)\frac{1}{L_i^1}\mathbb{E}\left[\|\nabla_i f(X^k)\|_{(i)\star}\right]\right)} \\
&= \sum_{t=0}^{K-1}\psi\left(\sum_{i=1}^p \frac{1}{L_i^1}\mathbb{E}\left[\|\nabla_i f(X^k)\|_{(i)\star}\right]\right),
\end{aligned}$$

where $\psi(t) := \frac{(1-\zeta)^2 t^2}{2\left(\sum_{i=1}^p \frac{L_i^0}{(L_i^1)^2} + (1+\zeta)t\right)}$. Since the function $\psi$ is increasing for $t > 0$, $\psi^{-1}$ exists. It follows that

$$\begin{aligned}
\Delta^0 &\ge \sum_{k=0}^{K-1}\psi\left(\sum_{i=1}^p \frac{1}{L_i^1}\mathbb{E}\left[\|\nabla_i f(X^k)\|_{(i)\star}\right]\right) \\
&\ge K\psi\left(\min_{k=0,\ldots,K-1}\sum_{i=1}^p \frac{1}{L_i^1}\mathbb{E}\left[\|\nabla_i f(X^k)\|_{(i)\star}\right]\right),
\end{aligned}$$

and hence

$$\min_{k=0,\ldots,K-1}\sum_{i=1}^p \frac{1}{L_i^1}\mathbb{E}\left[\|\nabla_i f(X^k)\|_{(i)\star}\right] \le \psi^{-1}\left(\frac{\Delta^0}{K}\right).$$

This in turn means that to reach the precision

$$\min_{k=0,\ldots,K-1}\sum_{i=1}^p \left[\frac{\frac{1}{L_i^1}}{\frac{1}{p}\sum_{j=1}^p \frac{1}{L_j^1}}\|\nabla_i f(X^k)\|_{(i)\star}\right] \le \varepsilon,$$

it suffices to run the algorithm for

$$
K = \left\lceil \frac{\Delta^0}{\psi \left( \varepsilon \left( \frac{1}{p} \sum_{j=1}^{p} \frac{1}{L_j^1} \right) \right)} \right\rceil
$$

$$
= \left\lceil \frac{2\Delta^0 \sum_{i=1}^{p} \frac{L_i^0}{(L_i^1)^2}}{(1-\zeta)^2 \varepsilon^2 \left( \frac{1}{p} \sum_{j=1}^{p} \frac{1}{L_j^1} \right)^2} + \frac{2\Delta^0(1+\zeta)}{(1-\zeta)^2 \varepsilon \left( \frac{1}{p} \sum_{j=1}^{p} \frac{1}{L_j^1} \right)} \right\rceil
$$

iterations.

$\square$

### E.2. Proof of Theorem 4.3

We now establish the main result of Section 4.3. The guarantees in Theorem 4.3 follow from the more general result below. Here, $\rho_i > 0$ for $i \in [p]$ denote the norm equivalence constants, i.e., $\|X_i\|_{(i)\star} \leq \rho_i \|X_i\|_2$ for all $X_i \in \mathcal{S}_i$.

**Theorem E.4.** *Let Assumptions 2.1 and 4.2 hold and fix $\varepsilon > 0$. Let $X^0, \ldots, X^{K-1}$ be the iterates of* Gluon *(Algorithm 1) run with $\beta^k = 1 - (k+1)^{-1/2}$, $t_i^k = t_i(k+1)^{-3/4}$ for some $t_i > 0$, and $M_i^0 = \nabla_i f_{\xi^0}(X^0)$.*

*1. If $L_i^1 = 0$, then*

$$
\min_{k=0,\ldots,K-1} \sum_{i=1}^{p} t_i \mathbb{E}\left[ \|\nabla_i f(X^k)\|_{(i)\star} \right]
$$

$$
\leq \frac{\Delta^0}{K^{1/4}} + \frac{1}{K^{1/4}} \sum_{i=1}^{p} \left[ \sigma \rho_i t_i \left( 7 + 2\sqrt{2e^2} \log(K) \right) + L_i^0 t_i^2 \left( \frac{87}{2} + 14 \log(K) \right) \right],
$$

*2. If $L_i^1 \neq 0$, then for $t_i = \frac{1}{12 L_i^1}$, we have*

$$
\min_{k=0,\ldots,K-1} \sum_{i=1}^{p} \frac{1}{12 L_i^1} \mathbb{E}\left[ \|\nabla_i f(X^k)\|_{(i)\star} \right]
$$

$$
\leq \frac{2\Delta^0}{K^{1/4}} + \frac{1}{K^{1/4}} \sum_{i=1}^{p} \left[ \frac{\sigma \rho_i}{6 L_i^1} \left( 7 + 2\sqrt{2e^2} \log(K) \right) + \frac{L_i^0}{144 (L_i^1)^2} \left( 87 + 28 \log(K) \right) \right],
$$

*where $\Delta^0 := f(X^0) - \inf_{X \in \mathcal{S}} f(X)$.*

*Proof.* We again start with the result in Lemma B.1 with $X = X^k$ and $Y = X^{k+1}$, obtaining

$$
\begin{aligned}
f(X^{k+1}) &\leq f(X^k) + \langle \nabla f(X^k), X^{k+1} - X^k \rangle + \sum_{i=1}^{p} \frac{L_i^0 + L_i^1 \|\nabla_i f(X^k)\|_{(i)\star}}{2} \|X_i^k - X_i^{k+1}\|_{(i)}^2 \\
&= f(X^k) + \sum_{i=1}^{p} \left[ \langle \nabla_i f(X^k), X_i^{k+1} - X_i^k \rangle_{(i)} + \frac{L_i^0 + L_i^1 \|\nabla_i f(X^k)\|_{(i)\star}}{2} \|X_i^k - X_i^{k+1}\|_{(i)}^2 \right] \\
&= f(X^k) + \sum_{i=1}^{p} \left[ \langle M_i^k, X_i^{k+1} - X_i^k \rangle_{(i)} + \langle \nabla_i f(X^k) - M_i^k, X_i^{k+1} - X_i^k \rangle_{(i)} \right] \\
&\quad + \sum_{i=1}^{p} \frac{L_i^0 + L_i^1 \|\nabla_i f(X^k)\|_{(i)\star}}{2} \|X_i^k - X_i^{k+1}\|_{(i)}^2.
\end{aligned}
$$

Applying the Cauchy-Schwarz inequality, we have

$$
\begin{aligned}
f(X^{k+1}) \;\leq\; & f(X^k) + \sum_{i=1}^{p}\left[\left\langle M_i^k, X_i^{k+1}-X_i^k\right\rangle_{(i)} + \|\nabla_i f(X^k)-M_i^k\|_{(i)\star}\|X_i^{k+1}-X_i^k\|_{(i)}\right] \\
& + \sum_{i=1}^{p}\frac{L_i^0+L_i^1\|\nabla_i f(X^k)\|_{(i)\star}}{2}\|X_i^k-X_i^{k+1}\|_{(i)}^2.
\end{aligned}
$$

Now, the update rule (1) and the definition of the dual norm $\|\cdot\|_{(i)\star}$ give

$$
\|X_i^k-X_i^{k+1}\|_{(i)}^2 \leq \left(t_i^k\right)^2
$$

and

$$
\left\langle M_i^k, X_i^{k+1}-X_i^k\right\rangle = \left\langle M_i^k, \mathrm{LMO}_{\mathcal{B}_i^k}\left(M_i^k\right)-X_i^k\right\rangle = -t_i^k\max_{\|X_i\|_{(i)}\leq 1}\left\langle M_i^k, X_i\right\rangle = -t_i^k\|M_i^k\|_{(i)\star}.
$$

Consequently,

$$
\begin{aligned}
& f(X^{k+1}) \\
\leq\; & f(X^k) + \sum_{i=1}^{p}\left[-t_i^k\|M_i^k\|_{(i)\star} + t_i^k\|\nabla_i f(X^k)-M_i^k\|_{(i)\star} + \frac{L_i^0+L_i^1\|\nabla_i f(X^k)\|_{(i)\star}}{2}\left(t_i^k\right)^2\right] \\
=\; & f(X^k) + \sum_{i=1}^{p}\left[-t_i^k\|M_i^k-\nabla_i f(X^k)+\nabla_i f(X^k)\|_{(i)\star} + t_i^k\|M_i^k-\nabla_i f(X^k)\|_{(i)\star}\right] \\
& + \sum_{i=1}^{p}\frac{L_i^0+L_i^1\|\nabla_i f(X^k)\|_{(i)\star}}{2}\left(t_i^k\right)^2 \\
\leq\; & f(X^k) + \sum_{i=1}^{p}\left[-t_i^k\|\nabla_i f(X^k)\|_{(i)\star} + 2t_i^k\|M_i^k-\nabla_i f(X^k)\|_{(i)\star}\right] \\
& + \sum_{i=1}^{p}\frac{L_i^0+L_i^1\|\nabla_i f(X^k)\|_{(i)\star}}{2}\left(t_i^k\right)^2.
\end{aligned}
$$

Taking expectations, we obtain

$$
\begin{aligned}
\mathbb{E}[f(X^{k+1})] \leq \mathbb{E}[f(X^k)] + \sum_{i=1}^{p}\Bigg[ & -t_i^k\mathbb{E}[\|\nabla_i f(X^k)\|_{(i)\star}] + 2t_i^k\mathbb{E}\left[\|M_i^k-\nabla_i f(X^k)\|_{(i)\star}\right] \\
& + \frac{L_i^0+L_i^1\mathbb{E}[\|\nabla_i f(X^k)\|_{(i)\star}]}{2}\left(t_i^k\right)^2\Bigg].
\end{aligned}
$$

Telescoping the last inequality gives

$$
\begin{aligned}
\sum_{i=1}^{p}\sum_{k=0}^{K-1}t_i^k\mathbb{E}[\|\nabla_i f(X^k)\|_{(i)\star}] \leq \Delta^0 + \sum_{i=1}^{p}\Bigg[ & 2\sum_{k=0}^{K-1}t_i^k\mathbb{E}\left[\|M_i^k-\nabla_i f(X^k)\|_{(i)\star}\right] \\
& + \sum_{k=0}^{K-1}\frac{L_i^0}{2}\left(t_i^k\right)^2 + \sum_{k=0}^{K-1}\frac{L_i^1}{2}\mathbb{E}[\|\nabla_i f(X^k)\|_{(i)\star}]\left(t_i^k\right)^2\Bigg],
\end{aligned}
\tag{27}
$$

where $\Delta^0 := f(X^0) - \inf_{X\in\mathcal{S}}f(X)$.

Now, inspired by the analysis in Hübler et al. (2024), we introduce the following notation: $\mu_i^k := M_i^k-\nabla_i f(X^k)$, $\gamma_i^k := \nabla_i f_{\xi^k}(X^k)-\nabla_i f(X^k)$, $\alpha^k = 1-\beta^k$, $\beta^{a:b} := \prod_{k=a}^{b}\beta^k$ and $S_i^k := \nabla_i f(X^{k-1})-\nabla_i f(X^k)$. Then, we can rewrite

the algorithm's momentum update rule as

$$
\begin{aligned}
M_i^k &= \beta^k M_i^{k-1} + (1 - \beta^k)\nabla_i f_{\xi^k}(X^k) \\
&= \beta^k \left( \mu_i^{k-1} + \nabla_i f(X^{k-1}) \right) + (1 - \beta^k)\left( \gamma_i^k + \nabla_i f(X^k) \right) \\
&= \nabla_i f\left( X^k \right) + \alpha^k \gamma_i^k + \beta^k S_i^k + \beta^k \mu_i^{k-1}.
\end{aligned}
$$

This yields

$$
\begin{aligned}
\mu_i^k &= M_i^k - \nabla_i f\left( X^k \right) \\
&= \alpha^k \gamma_i^k + \beta^k S_i^k + \beta^k \mu_i^{k-1} \\
&= \sum_{\tau=1}^k \beta^{(\tau+1):k} \alpha^\tau \gamma_i^\tau + \sum_{\tau=1}^k \beta^{\tau:k} S_i^\tau + \beta^{1:k} \mu_i^0 \\
&= \sum_{\tau=0}^k \beta^{(\tau+1):k} \alpha^\tau \gamma_i^\tau + \sum_{\tau=1}^k \beta^{\tau:k} S_i^\tau,
\end{aligned}
$$

where the last line follows from the fact that $M_i^0 = \nabla_i f_{\xi^0}(X^0)$ and $\beta^0 = 0$. Thus,

$$
\begin{aligned}
\mathbb{E}\left[ \left\| M_i^k - \nabla_i f(X^k) \right\|_{(i)\star} \right] &= \mathbb{E}\left[ \left\| \mu_i^k \right\|_{(i)\star} \right] \\
&\leq \mathbb{E}\left[ \left\| \sum_{\tau=0}^k \beta^{(\tau+1):k} \alpha^\tau \gamma_i^\tau \right\|_{(i)\star} \right] + \sum_{\tau=1}^k \beta^{\tau:k} \mathbb{E}\left[ \left\| S_i^\tau \right\|_{(i)\star} \right] \\
&\leq \rho_i \mathbb{E}\left[ \left\| \sum_{\tau=0}^k \beta^{(\tau+1):k} \alpha^\tau \gamma_i^\tau \right\|_2 \right] + \sum_{\tau=1}^k \beta^{\tau:k} \mathbb{E}\left[ \left\| S_i^\tau \right\|_{(i)\star} \right] \\
&\leq \rho_i \sqrt{ \sum_{\tau=0}^k \left( \beta^{(\tau+1):k} \alpha^\tau \right)^2 \mathbb{E}\left[ \left\| \gamma_i^\tau \right\|_2^2 \right] } + \sum_{\tau=1}^k \beta^{\tau:k} \mathbb{E}\left[ \left\| S_i^\tau \right\|_{(i)\star} \right],
\end{aligned}
$$

where in the last line we used Jensen's inequality and the fact that for all $q < l$

$$
\begin{aligned}
\mathbb{E}\left[ (\gamma_i^l)^\top \gamma_i^q \right] &= \mathbb{E}\left[ \mathbb{E}\left[ (\gamma_i^l)^\top \gamma_i^q \mid X_i^l \right] \right] = \mathbb{E}\left[ \mathbb{E}\left[ \gamma_i^l \mid X_i^l \right]^\top \gamma_i^q \right] \\
&= \mathbb{E}\left[ \left( \mathbb{E}\left[ \nabla_i f_{\xi^l}(X^l) - \nabla_i f(X^l) \mid X_i^l \right] \right)^\top \gamma_i^q \right] = 0.
\end{aligned}
$$

Using Assumptions 2.1 and 4.2, we get

$$
\mathbb{E}\left[ \left\| \gamma_i^\tau \right\|_2^2 \right] = \mathbb{E}\left[ \underbrace{\mathbb{E}\left[ \left\| \gamma_i^\tau \right\|_2^2 \mid X_i^\tau \right]}_{\leq \sigma^2} \right] \leq \sigma^2
$$

and

$$
\left\| S_i^\tau \right\|_{(i)\star} \leq \left( L_i^0 + L_i^1 \left\| \nabla_i f(X^\tau) \right\|_{(i)\star} \right) \left\| X_i^{\tau+1} - X_i^\tau \right\|_{(i)} \leq \left( L_i^0 + L_i^1 \left\| \nabla_i f(X^\tau) \right\|_{(i)\star} \right) t_i^\tau.
$$

Therefore,

$$
\begin{aligned}
\mathbb{E}\left[ \left\| M_i^k - \nabla_i f(X^k) \right\|_{(i)\star} \right] \leq\ & \sigma \rho_i \sqrt{ \sum_{\tau=0}^k \left( \beta^{(\tau+1):k} \alpha^\tau \right)^2 } + L_i^0 \sum_{\tau=1}^k \beta^{\tau:k} t_i^\tau \\
& + L_i^1 \sum_{\tau=1}^k \beta^{\tau:k} t_i^\tau \mathbb{E}\left[ \left\| \nabla_i f(X^\tau) \right\|_{(i)\star} \right].
\end{aligned}
$$

Combining the last inequality with (27) gives

$$
\sum_{i=1}^{p} \sum_{k=0}^{K-1} t_i^k \mathbb{E}[\|\nabla_i f(X^k)\|_{(i)\star}]
$$

$$
\leq \Delta^0 + \sum_{i=1}^{p} \Bigg[ \underbrace{2\sigma\rho_i \sum_{k=0}^{K-1} t_i^k \sqrt{\sum_{\tau=0}^{k} \left(\beta^{(\tau+1):k}\alpha^\tau\right)^2}}_{=:I_1} + \underbrace{2L_i^0 \sum_{k=0}^{K-1} t_i^k \sum_{\tau=1}^{k} \beta^{\tau:k} t_i^\tau}_{=:I_2}
$$

$$
+ \underbrace{2L_i^1 \sum_{k=0}^{K-1} t_i^k \sum_{\tau=1}^{k} \beta^{\tau:k} t_i^\tau \mathbb{E}\left[\|\nabla_i f(X^\tau)\|_{(i)\star}\right]}_{=:I_3}
$$

$$
+ \underbrace{\frac{L_i^0}{2} \sum_{k=0}^{K-1} \left(t_i^k\right)^2}_{=:I_4} + \frac{L_i^1}{2} \sum_{k=0}^{K-1} \left(t_i^k\right)^2 \mathbb{E}\left[\|\nabla_i f(X^k)\|_{(i)\star}\right] \Bigg]. \tag{28}
$$

Let us now upper-bound each term $I_i$, $i = 1, 2, 3, 4$.

$I_1$: using Lemma B.5, we obtain

$$
I_1 \leq \sigma\rho_i t_i \left(7 + 2\sqrt{2e^2}\log(K)\right).
$$

$I_2$: using Lemma B.5, we obtain

$$
I_2 \leq 14L_i^0 t_i^2 \left(3 + \log(K)\right).
$$

$I_3$: rearranging the sums and using Lemma B.4 with $a = \tau + 1$, $b = K$, $p = 3/4$ and $q = 1/2$, we have

$$
I_3 = 2L_i^1 \sum_{k=0}^{K-1} t_i^k \sum_{\tau=1}^{k} \beta^{\tau:k} t_i^\tau \mathbb{E}\left[\|\nabla_i f(X^\tau)\|_{(i)\star}\right]
$$

$$
= 2L_i^1 \sum_{\tau=1}^{K-1} t_i^\tau \left(\sum_{k=\tau}^{K-1} t_i^k \beta^{\tau:k}\right) \mathbb{E}\left[\|\nabla_i f(X^\tau)\|_{(i)\star}\right]
$$

$$
= 2L_i^1 \sum_{\tau=1}^{K-1} t_i^\tau t_i \left(\sum_{k=\tau}^{K-1} (k+1)^{-3/4} \beta^{\tau:k}\right) \mathbb{E}\left[\|\nabla_i f(X^\tau)\|_{(i)\star}\right]
$$

$$
\leq 2L_i^1 \sum_{\tau=1}^{K-1} t_i^\tau t_i \tau^{-1/4} \underbrace{e^{2\left((\tau+1)^{1/2} - \tau^{1/2}\right)}}_{\leq e^{2(\sqrt{2}-1)} \text{ for } \tau \geq 1} \mathbb{E}\left[\|\nabla_i f(X^\tau)\|_{(i)\star}\right]
$$

$$
\leq 2e^{2(\sqrt{2}-1)} L_i^1 \sum_{\tau=1}^{K-1} t_i^\tau t_i \tau^{-1/4} \mathbb{E}\left[\|\nabla_i f(X^\tau)\|_{(i)\star}\right]
$$

$$
\leq 2e^{2(\sqrt{2}-1)} L_i^1 \sum_{k=0}^{K-1} t_i^k t_i \mathbb{E}\left[\|\nabla_i f(X^k)\|_{(i)\star}\right].
$$

$I_4$:

$$
I_4 = \frac{L_i^0}{2} \sum_{k=0}^{K-1} \left(t_i^k\right)^2 \leq \frac{L_i^0}{2} \sum_{k=0}^{\infty} \left(t_i^k\right)^2 = \frac{L_i^0}{2} t_i^2 \sum_{k=0}^{\infty} (1+k)^{-3/2}
$$

$$
\leq \frac{L_i^0}{2} t_i^2 \left(1 + \int_1^\infty \frac{1}{z^{3/2}}\, dz\right) = \frac{3L_i^0}{2} t_i^2.
$$

Combining the upper-bounds for $I_i$, $i = 1, 2, 3, 4$ with (28) gives

$$\sum_{i=1}^{p} \sum_{k=0}^{K-1} t_i^k \mathbb{E}[\|\nabla_i f(X^k)\|_{(i)\star}] \leq \Delta^0 + \sum_{i=1}^{p} \left[ \sigma \rho_i t_i \left( 7 + 2\sqrt{2e^2} \log(K) \right) + 14 L_i^0 t_i^2 \left( 3 + \log(K) \right) \right.$$
$$+ 2e^{2(\sqrt{2}-1)} L_i^1 \sum_{k=0}^{K-1} t_i^k t_i \mathbb{E} \left[ \|\nabla_i f(X^k)\|_{(i)\star} \right]$$
$$\left. + \frac{3 L_i^0}{2} t_i^2 + \frac{L_i^1}{2} \sum_{k=0}^{K-1} \left( t_i^k \right)^2 \mathbb{E}[\|\nabla_i f(X^k)\|_{(i)\star}] \right].$$

Using the fact that $t_i^k = t_i (1+k)^{-3/4} \leq t_i$, and denoting $C := 2e^{2(\sqrt{2}-1)} + \frac{1}{2} \leq 5.1$, we get

$$\sum_{i=1}^{p} \sum_{k=0}^{K-1} t_i^k \mathbb{E}[\|\nabla_i f(X^k)\|_{(i)\star}] \leq \Delta^0 + \sum_{i=1}^{p} \left[ \sigma \rho_i t_i \left( 7 + 2\sqrt{2e^2} \log(K) \right) + 14 L_i^0 t_i^2 \left( \frac{87}{28} + \log(K) \right) \right.$$
$$\left. + C L_i^1 t_i \sum_{k=0}^{K-1} t_i^k \mathbb{E} \left[ \|\nabla_i f(X^k)\|_{(i)\star} \right] \right].$$

Now, let us consider two options: (1) $L_i^1 = 0$ for all $i \in \{1, \ldots, p\}$ and (2) $L_i^1 \neq 0$, for all $i \in \{1, \ldots, p\}$.

**Case 1:** $L_i^1 = 0$, $i = 1, \ldots, p$.   In this case,

$$\sum_{i=1}^{p} \sum_{k=0}^{K-1} t_i^k \mathbb{E}[\|\nabla_i f(X^k)\|_{(i)\star}] \leq \Delta^0 + \sum_{i=1}^{p} \left[ \sigma \rho_i t_i \left( 7 + 2\sqrt{2e^2} \log(K) \right) + 14 L_i^0 t_i^2 \left( \frac{87}{28} + \log(K) \right) \right],$$

and therefore,

$$\min_{k=0,\ldots,K-1} \sum_{i=1}^{p} t_i \mathbb{E}[\|\nabla_i f(X^k)\|_{(i)\star}]$$
$$\leq \quad \frac{1}{K} \sum_{k=0}^{K-1} \sum_{i=1}^{p} t_i \mathbb{E}[\|\nabla_i f(X^k)\|_{(i)\star}]$$
$$\leq \quad \frac{1}{K^{1/4}} \sum_{k=0}^{K-1} \sum_{i=1}^{p} t_i (1+k)^{-3/4} \mathbb{E}[\|\nabla_i f(X^k)\|_{(i)\star}]$$
$$= \quad \frac{1}{K^{1/4}} \sum_{k=0}^{K-1} \sum_{i=1}^{p} t_i^k \mathbb{E}[\|\nabla_i f(X^k)\|_{(i)\star}]$$
$$\leq \quad \frac{\Delta^0}{K^{1/4}} + \frac{1}{K^{1/4}} \sum_{i=1}^{p} \left[ \sigma \rho_i t_i \left( 7 + 2\sqrt{2e^2} \log(K) \right) + L_i^0 t_i^2 \left( \frac{87}{2} + 14 \log(K) \right) \right].$$

**Case 2:** $L_i^1 \neq 0$, $i = 1, \ldots, p$.   Let us choose $t_i = \frac{1}{12 L_i^1}$. Then

$$\sum_{i=1}^{p} \sum_{k=0}^{K-1} t_i^k \mathbb{E}[\|\nabla_i f(X^k)\|_{(i)\star}] \leq 2\Delta^0 + \sum_{i=1}^{p} \left[ 2\sigma \rho_i t_i \left( 7 + 2\sqrt{2e^2} \log(K) \right) + L_i^0 t_i^2 \left( 87 + 28 \log(K) \right) \right],$$

and hence

$$
\min_{k=0,\ldots,K-1} \sum_{i=1}^{p} \frac{1}{12 L_i^1} \mathbb{E}[\|\nabla_i f(X^k)\|_{(i)\star}]
$$

$$
\leq \quad \frac{1}{K} \sum_{k=0}^{K-1} \sum_{i=1}^{p} t_i \mathbb{E}[\|\nabla_i f(X^k)\|_{(i)\star}]
$$

$$
\leq \quad \frac{1}{K^{1/4}} \sum_{k=0}^{K-1} \sum_{i=1}^{p} t_i (1+k)^{-3/4} \mathbb{E}[\|\nabla_i f(X^k)\|_{(i)\star}]
$$

$$
= \quad \frac{1}{K^{1/4}} \sum_{i=1}^{p} \sum_{k=0}^{K-1} t_i^k \mathbb{E}[\|\nabla_i f(X^k)\|_{(i)\star}]
$$

$$
\leq \quad \frac{2\Delta^0}{K^{1/4}} + \frac{1}{K^{1/4}} \sum_{i=1}^{p} \left[ \frac{\sigma \rho_i}{6 L_i^1} \left( 7 + 2\sqrt{2e^2} \log(K) \right) + \frac{L_i^0}{144 (L_i^1)^2} \left( 87 + 28 \log(K) \right) \right].
$$

$\square$

# F. Additional Experimental Results and Details

## F.1. Experimental Details

All experiments for the `NanoGPT` model are conducted using PyTorch[7] with Distributed Data Parallel (DDP)[8] across 4 NVIDIA A100 GPUs (40GB each). For the `CNN` experiments, training is performed on a single NVIDIA A100 GPU (40GB). The training and evaluation pipelines are implemented using open-source codebases (Jordan, 2024; Jordan et al., 2024a; Pethick et al., 2025a), with all modifications clearly documented and properly referenced where applicable.

For LMO-based methods, we compute inexact LMOs using the Newton–Schulz iteration when an analytical solution is unavailable (e.g., for SVD-type updates), following the approach proposed by Jordan et al. (2024b). This method provides a computationally efficient approximation of the required orthogonalization while preserving the convergence behavior of the overall algorithm.

## F.2. Fitting $L_i^0$ and $L_i^1$

To minimize the Euclidean error between the true value $\hat{L}_i[k]$ and its approximation $\hat{L}_i^{\text{approx}}[k]$, while penalizing underestimation, we incorporate a hinge-like penalty term. Specifically, we fit $L_i^0$ and $L_i^1$ by minimizing the loss function

$$\mathcal{L}_i\left(L_i^0, L_i^1\right) := \sum_{k=0}^{K-1} \left(\hat{L}_i[k] - \hat{L}_i^{\text{approx}}[k]\right)^2 + \lambda \sum_{k=0}^{K-1} \max\left(0, \hat{L}_i[k] - \hat{L}_i^{\text{approx}}[k]\right)^2. \tag{29}$$

The first term of $\mathcal{L}_i$ captures the standard Euclidean (squared) error, while the second term introduces an additional penalty proportional to the amount of underestimation (i.e., when $\hat{L}_i[k] > \hat{L}_i^{\text{approx}}[k]$). The hyperparameter $\lambda \geq 0$ controls the strength of this penalty.

## F.3. Training NanoGPT on FineWeb

In this section, we present additional results and experimental details for the experiment described in the main text, which involves training a `NanoGPT` model on the `FineWeb` dataset using the unScion optimizer.

### F.3.1. EMPIRICAL VALIDATION OF ASSUMPTION 2.1

We begin by presenting additional results for the experiments described in Section 5, aimed at empirically validating Assumption 2.1. We plot the estimated *trajectory smoothness*

$$\hat{L}_i[k] := \frac{\|\nabla_i f_{\xi^{k+1}}(X^{k+1}) - \nabla_i f_{\xi^k}(X^k)\|_{(i)\star}}{\|X_i^{k+1} - X_i^k\|_{(i)}}$$

and its approximation

$$\hat{L}_i^{\text{approx}}[k] := L_i^0 + L_i^1 \|\nabla_i f_{\xi^{k+1}}(X^{k+1})\|_{(i)\star}$$

as functions of the iteration index $k$, where $L_i^0, L_i^1 \geq 0$ are fitted using the procedure described in Appendix F.2.

Figures 5, 6, and 7 show results for parameter groups from the embedding layer and from the 4th and 8th transformer blocks. Similar patterns are observed across all layers. In each case, we see a strong agreement between $\hat{L}_i[k]$ and $\hat{L}_i^{\text{approx}}[k]$, suggesting that Assumption 2.1 holds approximately along the optimization trajectory.

---

[7]PyTorch Documentation. Available at: https://pytorch.org/docs/stable/index.html
[8]Distributed Data Parallel (DDP) in PyTorch. Available at: https://pytorch.org/docs/stable/notes/ddp.html

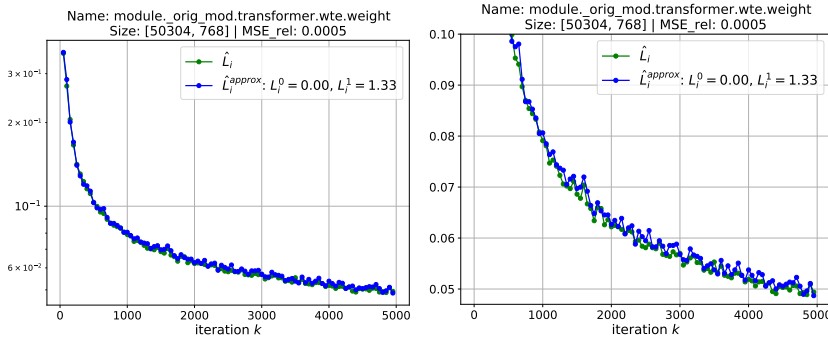

*Figure 5.* **Validation of layer-wise** $(L^0, L^1)$-**smoothness** for the group of parameters from the embedding layer of `NanoGPT-124M` along unScion training trajectories. The group norm is $\|\cdot\|_{(p)} = n_p\|\cdot\|_{1\to\infty}$, with fitted values $L_p^0 \approx 0$, $L_p^1 \approx 1.3$. The same plot is shown twice with different $y$-axis limits.

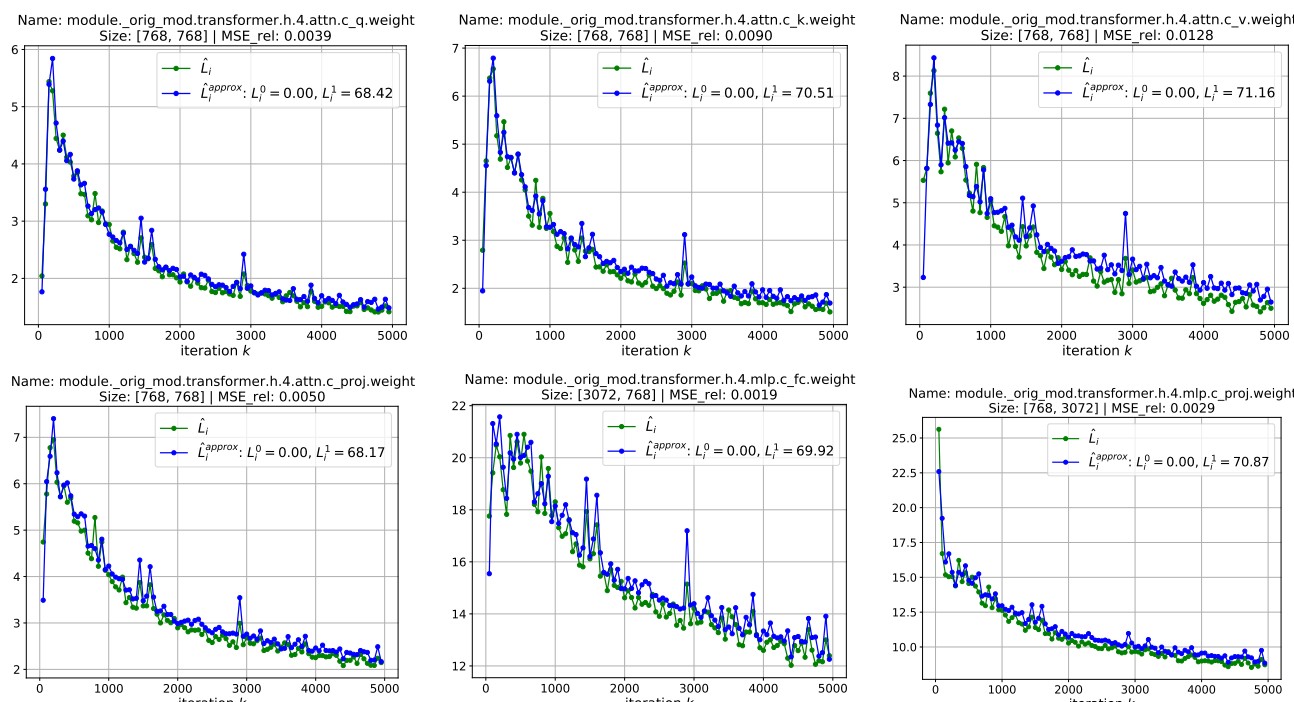

*Figure 6.* **Validation of layer-wise** $(L^0, L^1)$-**smoothness** for the group of parameters from the 4th transformer block of `NanoGPT-124M` along unScion training trajectories. The group norms are $\|\cdot\|_{(i)} = \sqrt{n_i/m_i}\|\cdot\|_{2\to 2}$, with fitted values $L_i^0 \approx 0$, $L_i^1 \approx 70$.

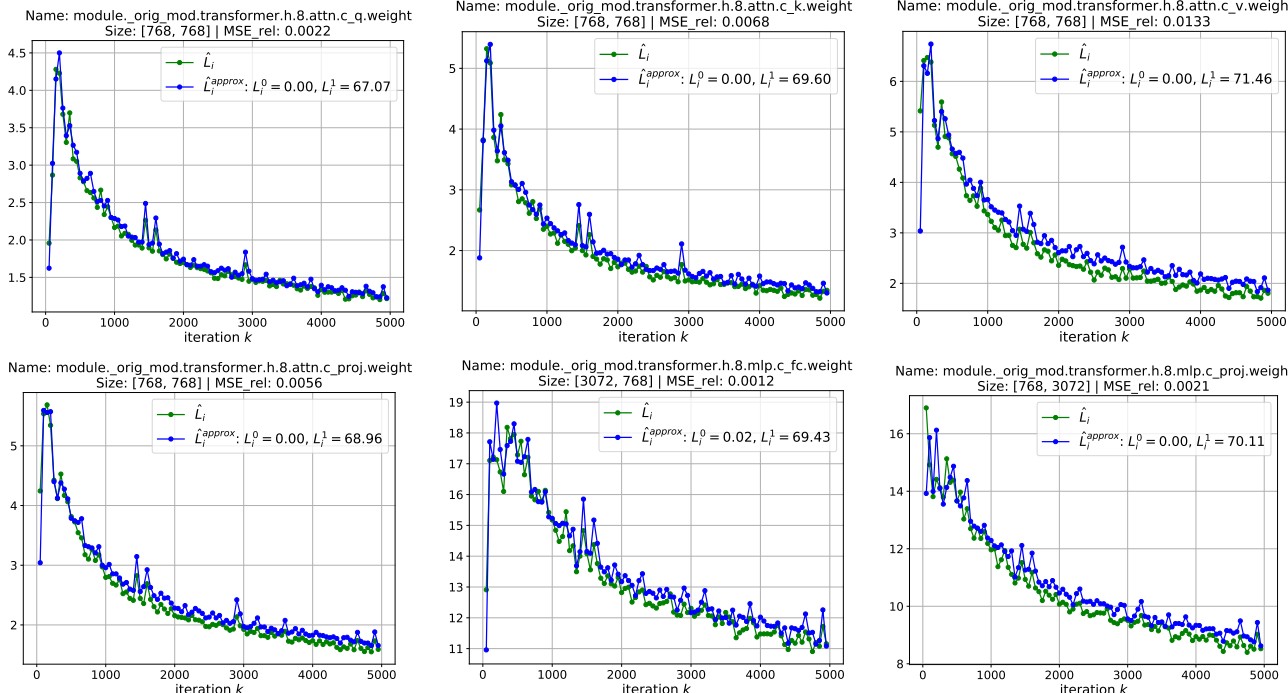

*Figure 7.* **Validation of layer-wise** $(L^0, L^1)$**-smoothness** for the group of parameters from the 8th transformer block of `NanoGPT-124M` along `unScion` training trajectories. The group norms are $\| \cdot \|_{(i)} = \sqrt{n_i/m_i} \| \cdot \|_{2\to 2}$, with fitted values $L_i^0 \approx 0$, $L_i^1 \approx 70$.

### F.3.2. GENERALIZED SMOOTHNESS UNDER EUCLIDEAN VS. SPECIALIZED NORMS

In this experiment, we compare how well the layer-wise $(L^0, L^1)$-smoothness assumption is satisfied under the standard Euclidean norms $\| \cdot \|_2$ for each parameter block, as opposed to the specialized norms described in (14). We adopt the same training setup as in Section 5, plotting the estimated trajectory smoothness $\hat{L}_i$ and its approximation $\hat{L}_i^{\text{approx}}$ along the training trajectories across several parameter groups. Unlike previous sections, here we do not penalize instances where $\hat{L}_i > \hat{L}_i^{\text{approx}}$ in order to find the best approximation (i.e., $\lambda = 0$ in (29)). Additionally, when using the standard Euclidean norm $\| \cdot \|_2$ for approximation, we exclude the first point, as it could distort the result.

We evaluate the quality of each approximation using the relative mean squared error ($\text{MSE}_i^{\text{rel}}$, denoted MSE_rel in the figures), defined as

$$\text{MSE}_i^{\text{rel}} := \frac{1}{K} \sum_{i=1}^{K} \left( \frac{\hat{L}_i[k] - \hat{L}_i^{\text{approx}}[k]}{\hat{L}_i[k]} \right)^2,$$

where a lower value indicates a better fit.

As shown in Figures 8 and 9, both visually and in terms of $\text{MSE}_i^{\text{rel}}$, using specialized norms for each group of parameters provides a better approximation than the standard Euclidean norm $\| \cdot \|_2$. Notably, the relative mean squared error $\text{MSE}_i^{\text{rel}}$ is consistently an order of magnitude lower under specialized norms.

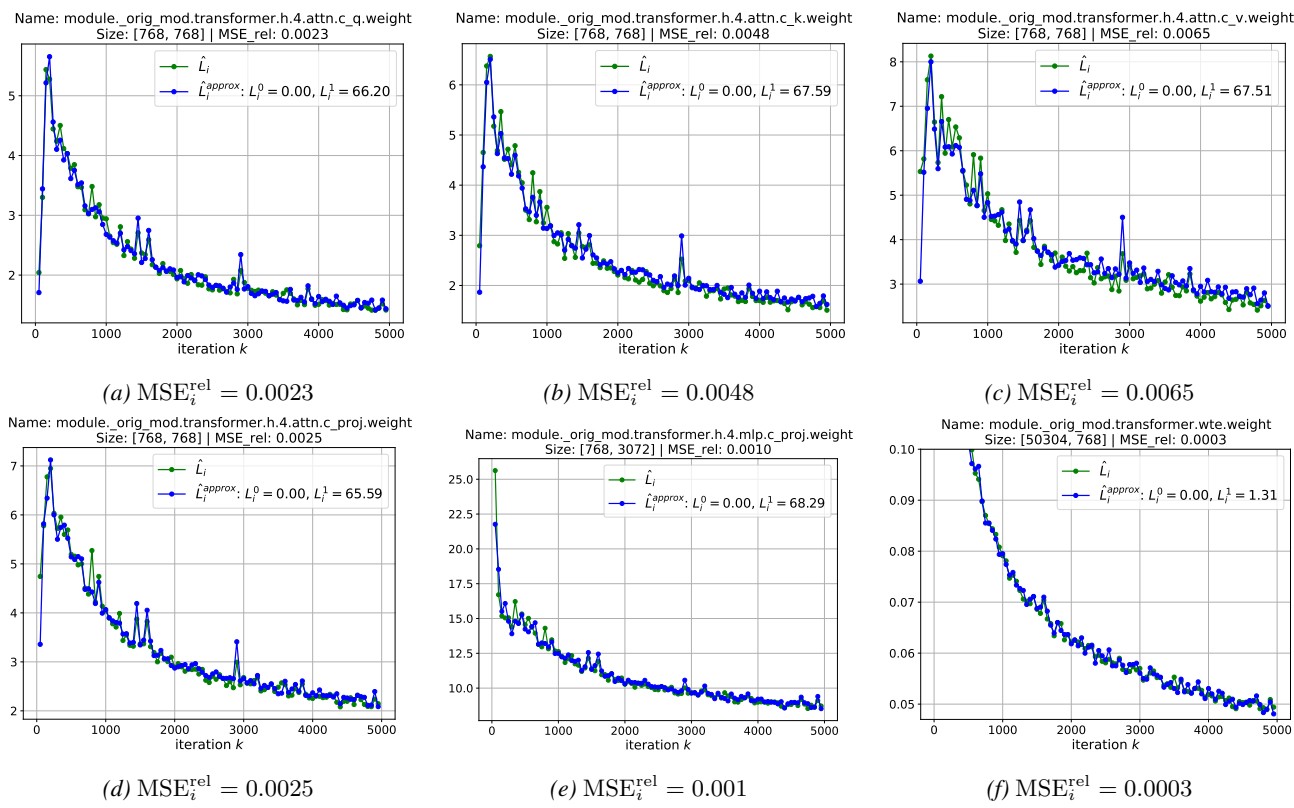

*Figure 8.* **Validation of layer-wise** $(L^0, L^1)$**-smoothness** for different groups of parameters in `NanoGPT-124M` along training trajectories of `unScion` using the specialized norm choices defined in (14).

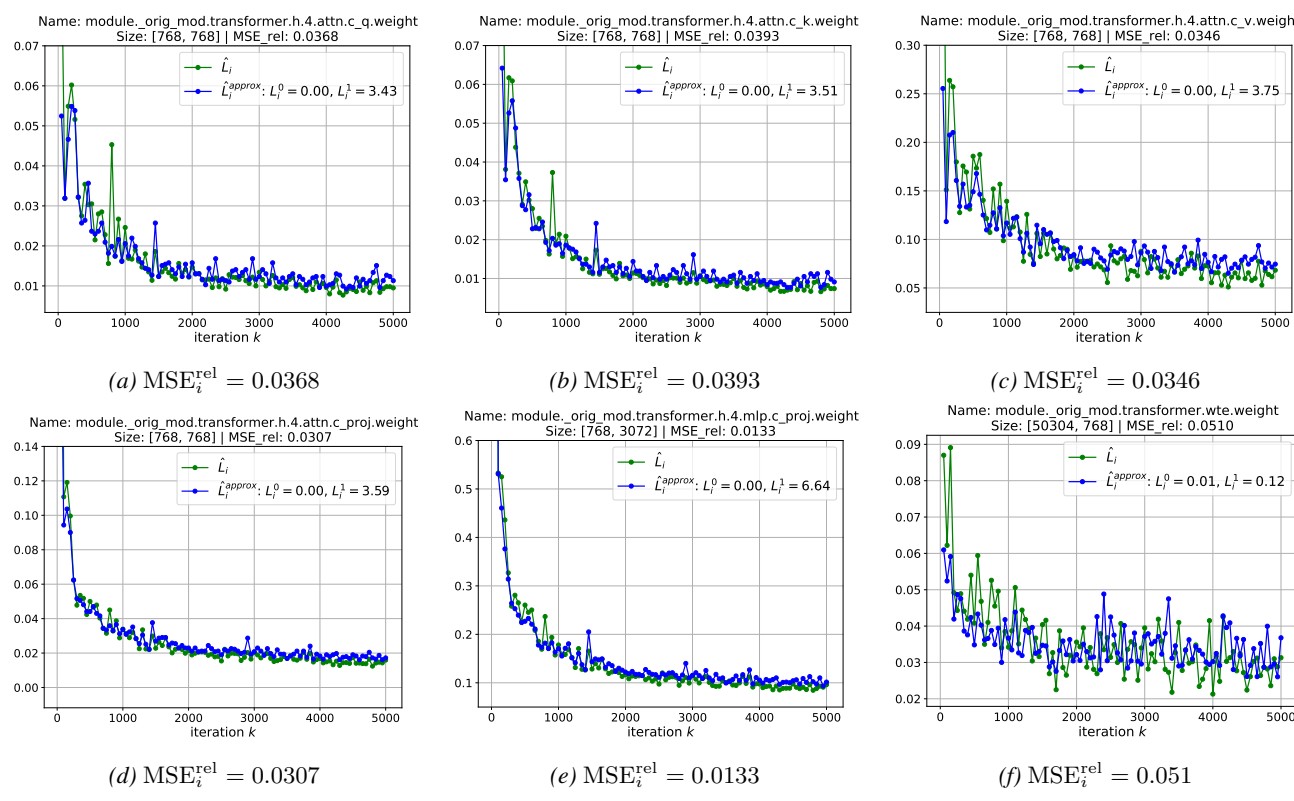

*Figure 9.* **Validation of layer-wise** $(L^0, L^1)$**-smoothness** for different groups of parameters in `NanoGPT-124M` along training trajectories of `unScion` using the standard Euclidean norm $\|\cdot\|_2$.

### F.3.3. LEARNING RATE TRANSFER FROM ADAMW

In this set of experiments, we aim to verify layer-wise $(L^0, L^1)$-smoothness following the approach used in Section 5, but employing the AdamW optimizer. We use hyperparameters specified in Pethick et al. (2025b, Table 7).

In Figure 10, we present the results for the estimated trajectory smoothness $\hat{L}_i$ and its approximation $\hat{L}_i^{\text{approx}}$ across several parameter groups along the training trajectories. Notably, for the group of parameters from the embedding layer $X_p$ (the last plot in Figure 10), the fitted value of $L_p^1$ is approximately 20–30 times smaller than in other groups. Since in all plots we observe that $L_i^0 \ll L_i^1 \|\nabla_i f_{\xi^k}(X^k)\|_{(i)\star}$, Theorem 4.1 implies that $t_i^k \approx 1/L_i^k$. Thus, $t_p^k$ should be 20–30 times larger than $t_i^k$ for $i = 1, \ldots, p - 1$, which is consistent with the tuned parameters from Pethick et al. (2025b, Table 7).

Figure 11 shows the estimated trajectory smoothness across all layers and blocks. We again observe substantial variation across layers within each block, reinforcing the need for the per-layer treatment introduced in our framework.

This insight provides an efficient and principled method for initializing learning rates in Scion. Smoothness statistics collected during standard AdamW training (which is commonly used for training LLMs) can serve as a strong prior, allowing practitioners to directly incorporate structure-aware choices, such as larger stepsizes for embedding layers, into their tuning process. Importantly, computing these statistics is computationally inexpensive, introducing minimal additional cost.

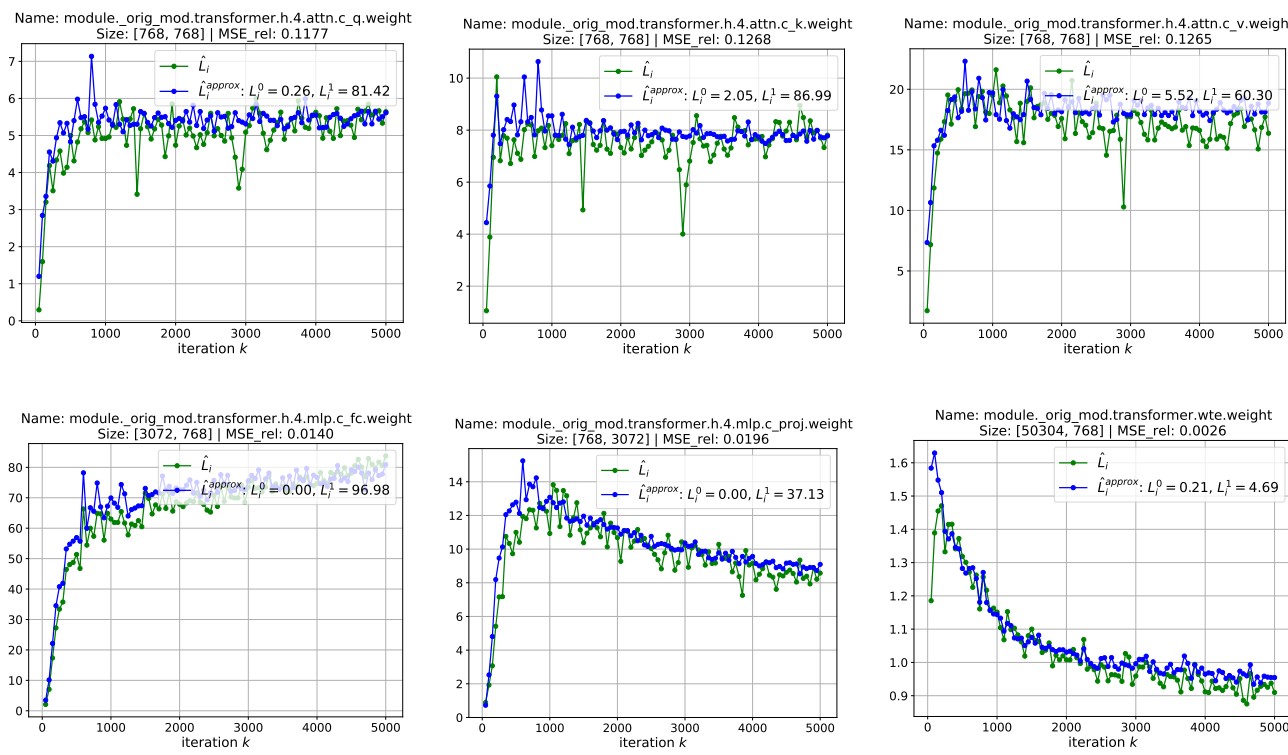

*Figure 10.* **Validation of layer-wise** $(L^0, L^1)$**-smoothness** for different groups of parameters in NanoGPT-124M along AdamW training trajectories.

### F.3.4. BEST VS. WORST FITS AND AGGREGATE FIT QUALITY

Figure 12 illustrates the best and worst per-layer fits of Assumption 2.1, comparing the measured trajectory smoothness $L_i[k]$ with its approximation $L_i^{\text{approx}}[k]$ along the unScion training trajectory on NanoGPT-124M. For comparison, we also report fits obtained under the classical (though layer-wise) smoothness model, corresponding to Assumption 2.1 with $L_i^1 = 0$ for all layers.

We observe very tight fits for many layers (e.g., embeddings and several attention V matrices–see also Figure 13), while a few layers show looser, yet still bounded, fits. Quantitatively, the error of our model, measured by the relative mean squared error, is more than an order of magnitude smaller than that of classical smoothness (i.e., constant-$L_i$ fits) for most layers, and in some cases smaller by several orders of magnitude. These results further demonstrate that Assumption 2.1 provides

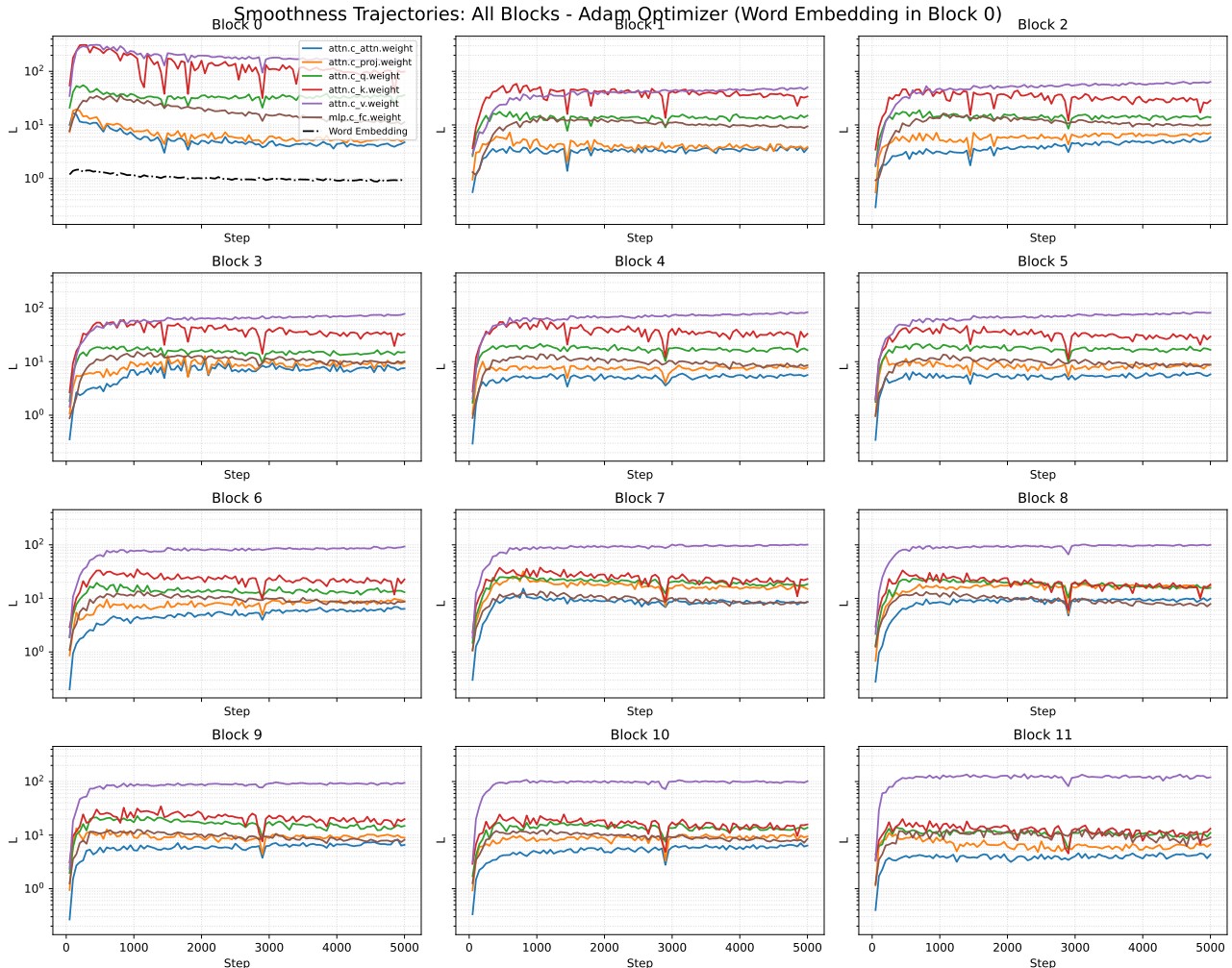

*Figure 11.* **Estimated trajectory smoothness for `NanoGPT-124M` trained with** AdamW. The same cross-layer heterogeneity pattern persists, indicating that layer-wise $(L_i^0, L_i^1)$-smoothness is not specific to unScion.

an accurate description of the loss landscape, whereas classical smoothness, even when applied in a layer-wise manner, yields a substantially poorer fit.

### F.3.5. LAYER-WISE SMOOTHNESS ACROSS LAYERS AND MODEL SCALES

The final set of language modeling experiments examines whether the observations made for `NanoGPT-124M` extend to larger model scales. To this end, we additionally estimate trajectory smoothness for models trained with the unScion optimizer at two larger scales: `GPT-2 Medium` ($\sim$355M parameters) and `GPT-2 Large` ($\sim$774M parameters). The estimation procedure follows the same methodology used in the preceding `NanoGPT-124M` experiments.

Figures 14, 15, and 16 report the aggregate layer-wise trajectory smoothness across all layers and blocks for the three model scales: `NanoGPT` ($\sim$124M), `GPT-2 Medium` ($\sim$355M), and `GPT-2 Large` ($\sim$774M). Across all scales, we consistently observe pronounced cross-layer heterogeneity and the empirical trend $L_i^0 \approx 0$, indicating that these phenomena persist from 124M up to 774M parameters.

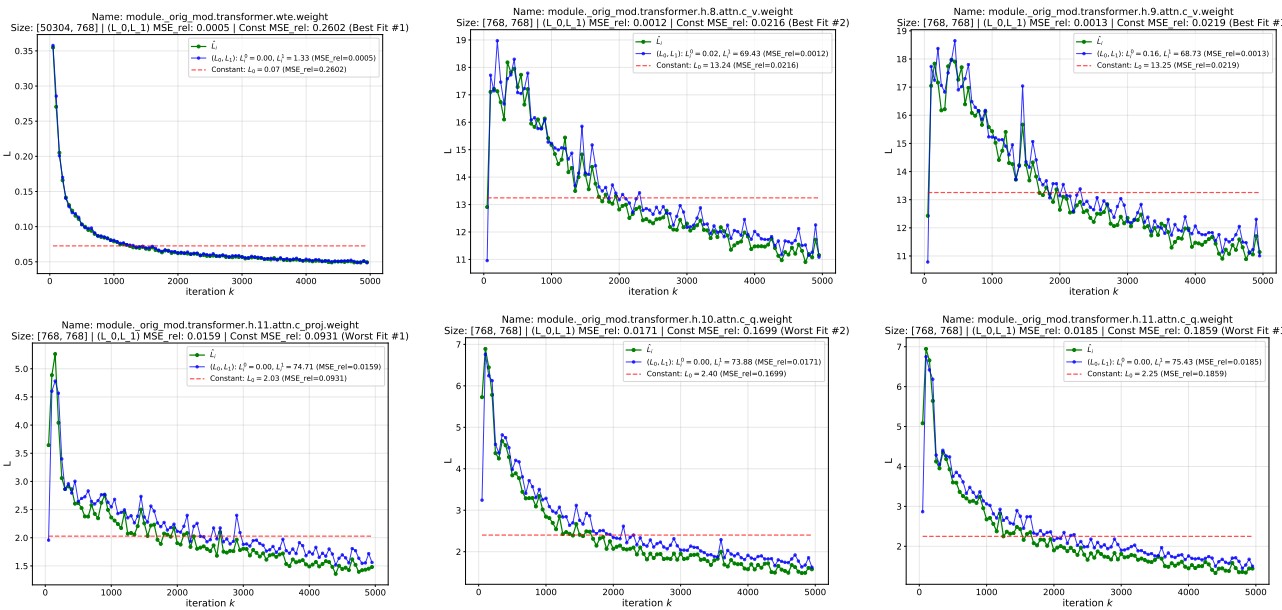

*Figure 12.* Illustrative *best* (top row) and *worst* (bottom row) per-layer fits of Assumption 1 (measured $L_i[k]$ vs. $L_i^{\text{approx}}[k]$) along `NanoGPT-124M` training.

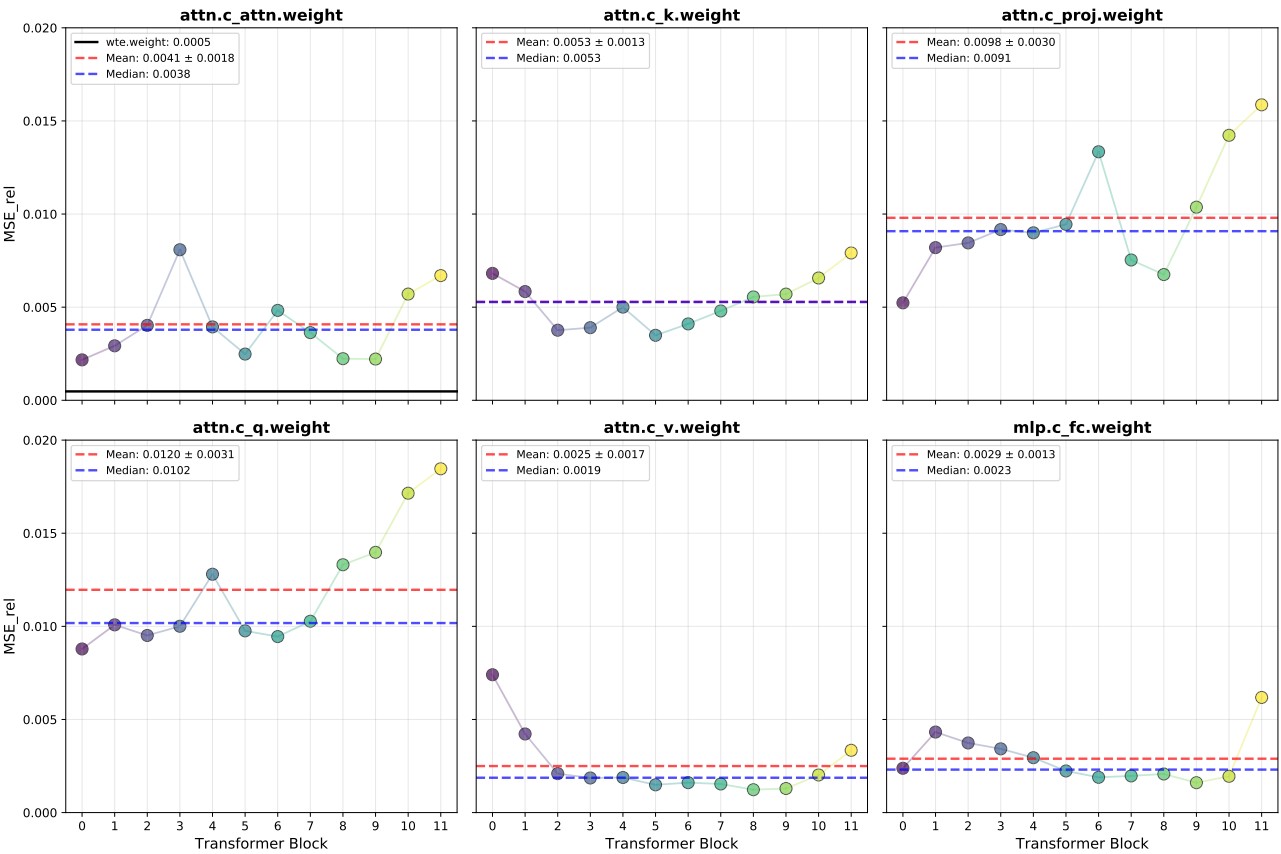

*Figure 13.* **Relative fit** $\text{MSE}_{\text{rel}}$ **error of the layer-wise** $(L_i^0, L_i^1)$ **model** across transformer blocks in `NanoGPT-124M` for each matrix type. Dashed lines indicate the mean and median over blocks (embedding matrix: $\text{MSE}_{\text{rel}} = 0.0005$).

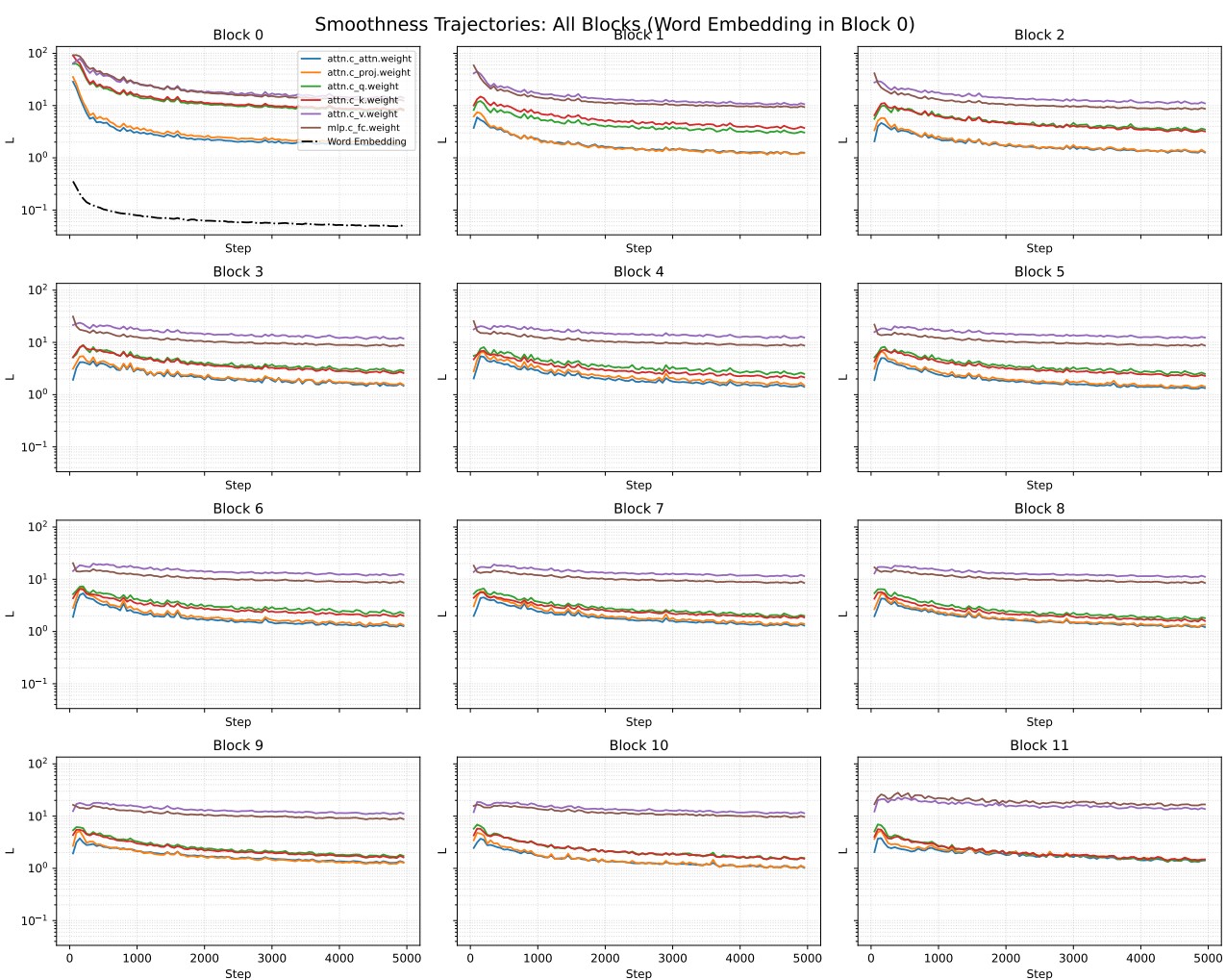

*Figure 14.* Estimated trajectory smoothness for `NanoGPT-124M` trained with unScion.

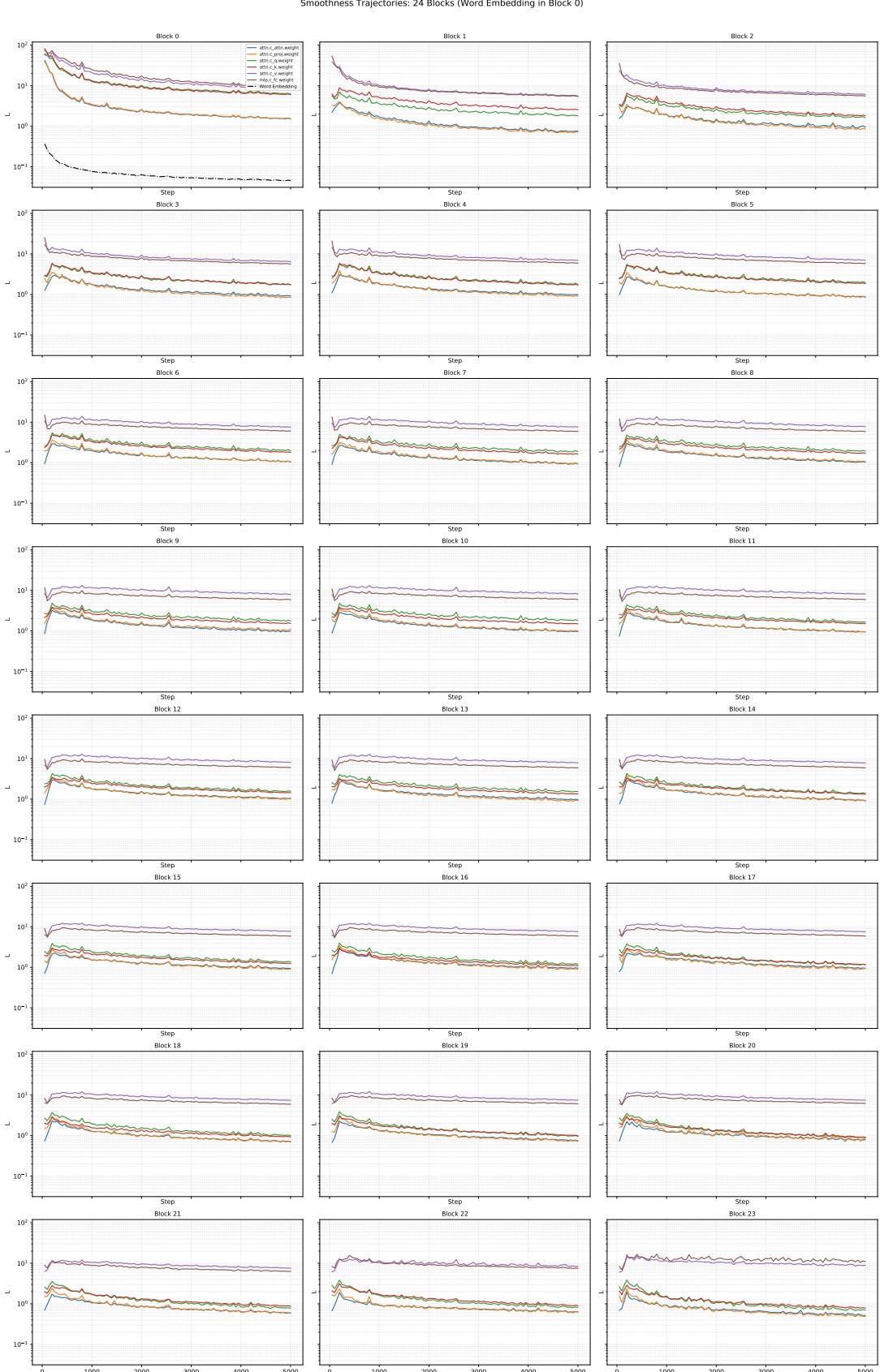

*Figure 15.* Estimated trajectory smoothness for `GPT-2 Medium` (∼355M parameters) trained with unScion.

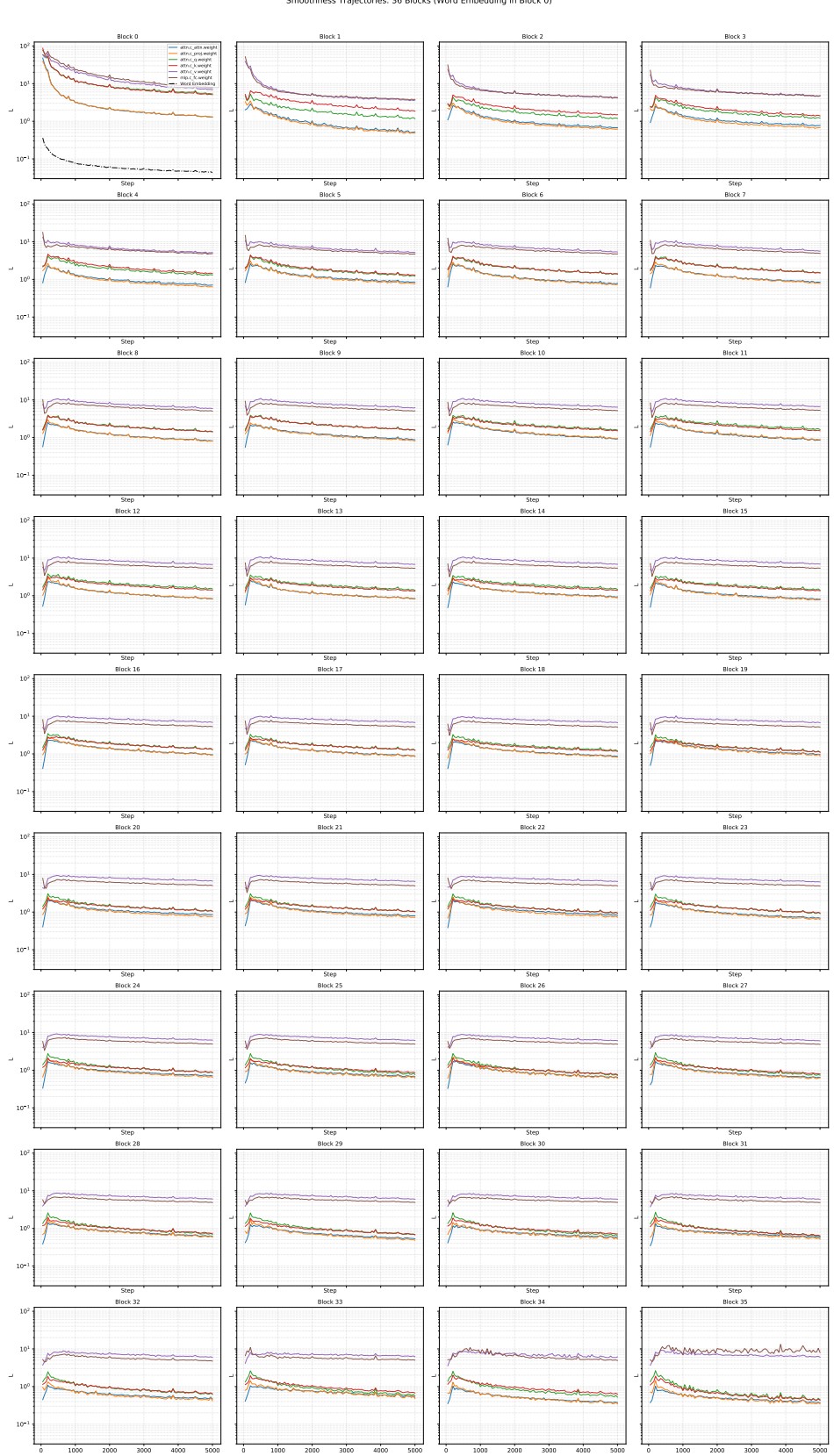

*Figure 16.* Estimated trajectory smoothness for `GPT-2 Large` (∼774M parameters) trained with unScion.

## F.4. Training CNN on CIFAR-10

In this experiment, we further validate layer-wise $(L^0, L^1)$-smoothness by training a `CNN` model on the `CIFAR-10` dataset, following implementations from two open-source GitHub repositories (Jordan, 2024; Pethick et al., 2025a).

**Full-batch (deterministic) gradients.** We begin by presenting results in the deterministic setting, where the model is trained using the unScion optimizer (15) with full-batch gradients $\nabla_i f$, no momentum and no learning rate decay. Other hyperparameters are as in Pethick et al. (2025b, Table 10), except that we train for more epochs. Similar to the `NanoGPT` experiments discussed in Section 5, we plot the estimated (non-stochastic) trajectory smoothness

$$\hat{L}_i[k] := \frac{\|\nabla_i f(X^{k+1}) - \nabla_i f(X^k)\|_{(i)\star}}{\|X_i^{k+1} - X_i^k\|_{(i)}}$$

alongside its approximation

$$\hat{L}_i^{\text{approx}}[k] := L_i^1 \|\nabla_i f(X^{k+1})\|_{(i)\star}$$

for selected parameter groups. In this experiment, we consider a simplified variant of Assumption 2.1, setting $L_i^0 = 0$, and estimate $L_i^1 \geq 0$ using the same procedure as in Section 5. Figure 17 presents the results, demonstrating that Assumption 2.1 is approximately satisfied along the training trajectory. When this condition holds with $L_i^0 = 0$, Theorem 4.1 guarantees convergence under the stepsize choice $t_i^k \equiv t_i = 1/L_i^1$. In this setting, the estimated $L_i^1$ values (shown in Figure 17) are $L_i^1 \approx 3$-4 for all parameter groups except for the classification head weights $X_p$, where $L_p^1 \approx 0.04$. This roughly two-orders-of-magnitude difference justifies the much larger radius $t_p^k$ used for the head weights in the tuned configuration reported in Pethick et al. (2025b, Table 10).

**Stochastic gradients.** Here, we report results for analogous experiments in the stochastic setting, using noisy gradients $\nabla_i f_{\xi^k}$. We use momentum as in Pethick et al. (2025b, Table 10), but do not apply a linear decay schedule. In Figure 18, we

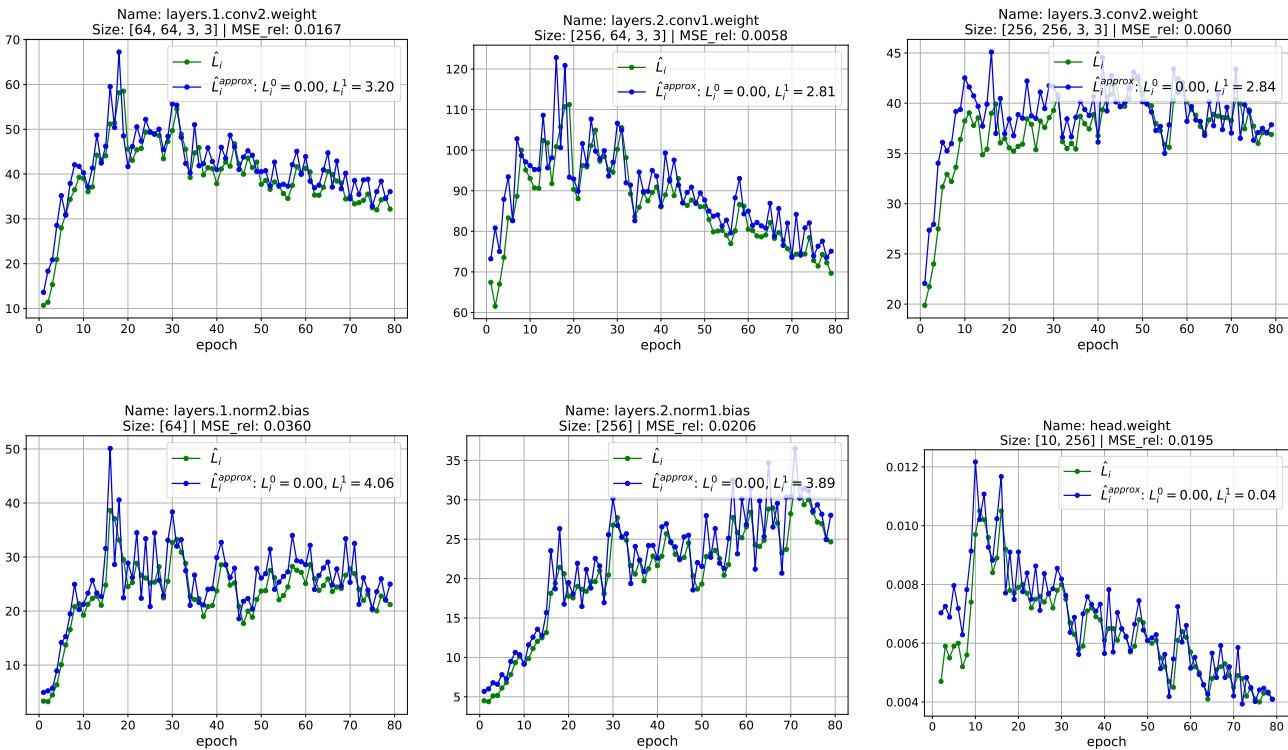

*Figure 17.* **Validation of layer-wise $(L^0, L^1)$-smoothness** for different groups of parameters of a `CNN` model along the training trajectories of unScion with **full-batch gradients**. The norms used for each group are as follows: $\|\cdot\|_{(i)} = \sqrt{1/C_i^{out}}\|\cdot\|_2$ for biases, $\|\cdot\|_{(i)} = k^2 \sqrt{C_i^{in}/C_i^{out}}\|\cdot\|_{2\to 2}$ for conv, and $\|\cdot\|_{(p)} = n_p\|\cdot\|_{1\to\infty}$ for the last group $X_p$, associated with classification head weights.

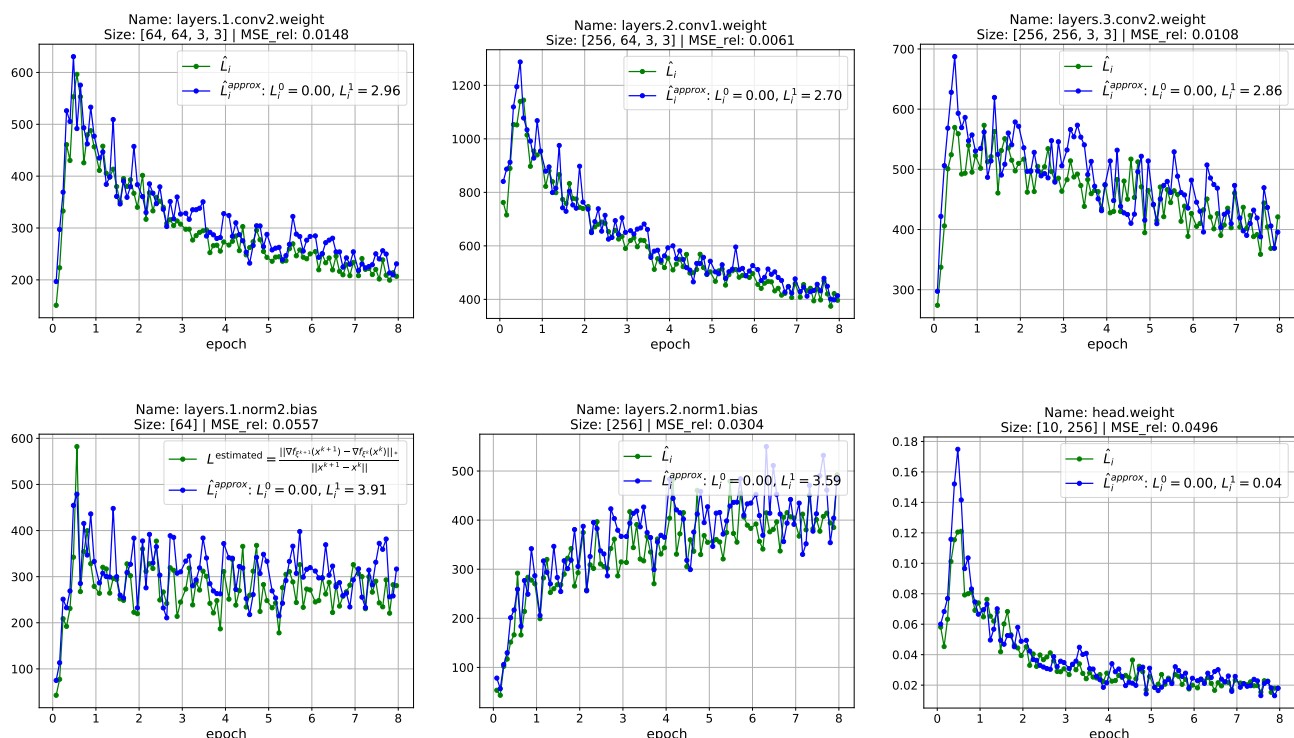

*Figure 18.* **Validation of layer-wise** $(L^0, L^1)$**-smoothness** for different groups of parameters of a CNN model along the training trajectories of unScion with **stochastic gradients**. The norms used for each group are as follows: $\|\cdot\|_{(i)} = \sqrt{1/C_i^{out}}\|\cdot\|_2$ for biases, $\|\cdot\|_{(i)} = k^2\sqrt{C_i^{in}/C_i^{out}}\|\cdot\|_{2\to 2}$ for conv, and $\|\cdot\|_{(p)} = n_p\|\cdot\|_{1\to\infty}$ for the last group $X_p$, associated with classification head weights.

plot

$$\hat{L}_i[k] = \frac{\|\nabla_i f_{\xi^{k+1}}(X^{k+1}) - \nabla_i f_{\xi^k}(X^k)\|_{(i)\star}}{\|X_i^{k+1} - X_i^k\|_{(i)}}, \qquad \hat{L}_i^{\text{approx}}[k] = L_i^1\|\nabla_i f_{\xi^{k+1}}(X^{k+1})\|_\star,$$

again setting $L_i^0 = 0$. Despite the added variance, we still observe that the stochastic trajectory roughly adheres to Assumption 2.1.

Altogether, the results further support the validity of Assumption 2.1 with $L_i^0 = 0$.

