# OpenReview forum: "From Muon to Gluon: Bridging Theory and Practice of LMO-based Optimizers for LLMs"
_ICML.cc/2026/Conference — ICML 2026 regular_

### Official Review · Reviewer_USDT · 2026-03-12

**Soundness:** 3
**Presentation:** 4
**Significance:** 3
**Originality:** 3
**Overall Recommendation:** 5
**Confidence:** 4

**Summary:**

This work introduces Gluon, an optimization framework based on Linear Minimization Oracles (LMO), which encompasses Muon and Scion as special cases under specific norm choices. To theoretically analyze Gluon, the authors introduce a layer-wise (L_0, L_1)-smoothness assumption, which generalizes the (L_0, L_1)-smoothness condition of Zhang et al. (2022) by accounting for the layer-wise structure of deep neural networks. Under this assumption, the authors establish convergence guarantees for Gluon in both deterministic and stochastic settings. Finally, they conduct experiments on NanoGPT and CNN models and observe that the layer-wise (L_0, L_1)-smoothness assumption can effectively capture properties of the loss landscape in practice.

**Compliance With Llm Reviewing Policy:**

Affirmed.

**Final Justification:**

I am satisfied with the clarification on the gap between the theoretical assumption and the empirical measurement from the author rebuttal. I am maintaining my positive score and recommend acceptance of this paper.

**Key Questions For Authors:**

Could Assumption 2.1 be replaced with a trajectory-dependent directional smoothness assumption, similar to the one introduced in Mishkin et al. (2024)? In the experiments, the smoothness constants are estimated by measuring directional smoothness along the update direction. Given this, it seems plausible that the theoretical analysis could be strengthened by replacing the layer-wise (L_0, L_1)-smoothness assumption with a corresponding directional smoothness assumption. If this is not feasible, it would be helpful for the authors to explicitly discuss the distinction between the theoretical assumption and the empirical measurement used in the experiments.

Mishkin et al. Directional Smoothness and Gradient Methods: Convergence and Adaptivity. NeurIPS 2024.

**Limitations:**

yes

**Strengths And Weaknesses:**

The paper is very well written and clearly presents its core ideas. The main strengths are twofold. First, the theory studies the Gluon framework, which generalizes recently proposed matrix-based optimizers such as Muon and Scion that have shown strong empirical performance in LLM training. Second, the main theoretical assumption—layer-wise (L_0, L_1)-smoothness—appears to better capture the behavior of practical deep network training compared to the standard L-smoothness condition, which is often unrealistic in modern deep learning settings. In this sense, the paper demonstrates good awareness of practical considerations by analyzing state-of-the-art optimization methods under assumptions that seem more aligned with real training dynamics. Overall, this is a solid piece of optimization theory that attempts to bridge theory and practice.

At the same time, one potential weakness concerns how the layer-wise (L_0, L_1)-smoothness constants are estimated in the experiments. In experiments, the authors measure directional smoothness along the update direction, as defined in Eq. (10). However, this directional quantity may differ substantially from the true smoothness constant unless the update direction happens to align with the worst-case direction. As a result, the empirical measurements may not fully reflect the theoretical smoothness parameters used in the analysis.

---

> ### Author Rebuttal · Authors · 2026-03-31
>
> We thank the reviewer for the positive evaluation and for the very insightful question about the gap between the theoretical assumption and the empirical measurement.
>
> > In the experiments, the authors measure directional smoothness along the update direction. This may differ substantially from the true smoothness constant.
>
> We agree that Eq. (10) does not directly estimate the global worst-case constants from Assumption 2.1. What it measures is a trajectory-level directional proxy along the updates actually taken during training. Our experimental goal was therefore not to certify the global assumption, but to test whether the proposed $(L_0,L_1)$ model meaningfully describes the parts of the loss landscape that practical training trajectories actually visit.
>
> We will revise the paper to make this distinction explicit. At the same time, this trajectory-level validation is still relevant to the theory. As noted in the footnote on p.4, the proofs only invoke Assumption 2.1 locally between consecutive iterates $X^k$ and $X^{k+1}$, rather than for arbitrary pairs of points in the domain. In that sense, the experiments are aligned with the part of the landscape that the analysis actually uses.
>
> Another reason we still find this proxy informative is comparative: even if it does not recover the worst-case constant, it allows us to test whether a constant-$L$ model or a layer-wise $(L_0,L_1)$ model better tracks the geometry encountered in practice. Empirically, the latter is substantially more accurate, which is exactly the intended empirical message of Section 5.
>
> We would also like to emphasize that the same coarse cross-layer heterogeneity appears under AdamW and persists at larger scales (GPT-2 Medium and GPT-2 Large). This suggests that the observed layer separation is not specific to a single optimizer trajectory.
>
> > Could Assumption 2.1 be replaced with a trajectory-dependent directional smoothness assumption, similar to Mishkin et al. (2024)?
>
> We agree that this is a very natural direction. In fact, our current proof already operates in a related spirit: although Assumption 2.1 is stated globally for simplicity, it is only used along the optimization trajectory, and only locally between consecutive iterates. A fully pathwise / directional formulation could therefore likely serve as an alternative starting point for analysis.
>
> We did not pursue this route in the present paper because our main goal was to develop a clean layer-wise non-Euclidean analogue of $(L_0,L_1)$-smoothness that both recovers classical smoothness as a special case and directly yields interpretable per-layer stepsize formulas. A purely directional assumption may well be possible, but it would answer a somewhat different question: it would be more tightly matched to the trajectory-level proxy used in the experiments, while giving up some of the generality, comparability to prior generalized-smoothness work, and interpretability of the current assumption.
>
> That said, we agree that the distinction between the stronger globally stated assumption and the weaker directional proxy used in the experiments should be discussed more explicitly. We will add this discussion and cite the connection to Mishkin et al. more clearly in the final version.
>
> Our claim is therefore not that we have globally certified Assumption 2.1 for deep networks, but that the proposed layer-wise $(L_0,L_1)$ model appears to describe the geometry encountered by modern training trajectories substantially better than classical smoothness. Thank you again for this very helpful suggestion.

---

> > ### Author Rebuttal · Reviewer_USDT · 2026-04-02
> >
> > Thank you very much for addressing my questions. I am very happy to maintain my positive score and recommend acceptance of this paper.

---

> > > ### Author Response · Authors · 2026-04-07
> > >
> > > Thank you for reading our rebuttal and for the positive feedback. We appreciate your support and your helpful comments during the review process.

---

### Official Review · Reviewer_izv4 · 2026-03-12

**Soundness:** 2
**Presentation:** 3
**Significance:** 2
**Originality:** 3
**Overall Recommendation:** 4
**Confidence:** 3

**Summary:**

This paper proposes Gluon, a unified optimization framework that recovers state-of-the-art LMO-based optimizers like Muon and Scion as special cases. The authors identify two critical gaps between theory and practice in prior analyses: the neglect of layer-wise structure and reliance on unrealistic classical smoothness assumptions that yield impractically small stepsizes. To address this, they introduce a novel layer-wise $(L^0, L^1)$-smoothness assumption that captures the anisotropic geometry of deep networks, where smoothness varies significantly across layers and depends on gradient magnitude. Under this assumption, their convergence theory produces adaptive, per-layer stepsizes that closely match the manually tuned values reported in prior empirical work—demonstrating, for the first time, that optimization theory can have genuine predictive power for these methods. Experiments on NanoGPT and CNN training confirm that the assumption holds along training trajectories, validating the practical relevance of the framework.

**Compliance With Llm Reviewing Policy:**

Affirmed.

**Final Justification:**

The author clarified some technical issues that I considered to be crucial, so I adjusted my score to 4 points.

**Key Questions For Authors:**

Questions：

Q1: The current empirical validation is limited to NanoGPT-124M and a small CNN on CIFAR-10. Could the authors provide additional experiments at larger scales (e.g., 1B+ parameters or longer training runs) to verify whether the key observations, such as $L_i^0 \approx 0$ and the significant variation of $L_i^1$ across layer types, continue to hold? This would substantially strengthen the empirical foundation of the proposed framework.

Q2: The smoothness constants $L_i^0$ and $L_i^1$ used to derive the theoretical stepsizes in Section 5.1 are fitted from the same training trajectory that was generated using manually tuned hyperparameters from Pethick et al. (2025). Have the authors considered running an end-to-end experiment where the smoothness constants are estimated from an independent source (e.g., a short AdamW pilot run, as suggested in Appendix F.3.3), and then used to set the stepsizes for a fresh Gluon training run without further tuning? Such an experiment would help clarify whether the theory has genuine predictive power beyond post-hoc consistency.

Q3: The convergence analysis assumes exact LMO computation, but practical implementations rely on Newton-Schulz iterations to approximate the SVD. Could the authors comment on the magnitude of this approximation error in their experiments, and whether incorporating inexact LMO analysis into the framework is feasible? Even an empirical measurement of the gap between the exact and approximate LMO solutions along the training trajectory would be informative.

If the authors can provide satisfactory responses to these questions, I would be willing to consider raising my score.

**Limitations:**

Yes

**Strengths And Weaknesses:**

Strengths

1. Proposed a Layer-wise Analysis Framework and Provided Empirical Evidence for Its Reasonableness:

The paper proposes a layer-wise $(L^0, L^1)$-smoothness assumption (Assumption 2.1), extending the generalized smoothness model of Zhang et al. (2020) to a per-layer, non-Euclidean setting. Rather than claiming this as a purely theoretical result, its main value lies in the accompanying empirical analysis that validates the framework's reasonableness. Specifically, by fitting $L_i^0$ and $L_i^1$ along training trajectories, the authors observe two concrete findings: first, $L_i^0 \approx 0$ holds consistently across all layers in both NanoGPT and CNN experiments (Figures 2, 4, 5 to 9), indicating that gradient-dependent smoothness dominates over the constant term; second, the fitted $L_i^1$ values differ dramatically across layer types. In NanoGPT, embedding layers have $L_p^1 \approx 1.3$ versus $L_i^1 \approx 70$ for transformer layers (a ~50x gap), and in the CNN, classification head weights have $L_p^1 \approx 0.03$ versus $L_i^1 \approx 3$ for convolutional layers (a ~100x gap). These observations provide concrete empirical evidence that the loss landscape's smoothness structure is highly heterogeneous across layers, lending support to the idea that per-layer treatment is not just convenient but reflects genuine geometric differences in the optimization problem.

2. Stochastic Stepsizes Depend on Current Iteration, Not Total Iteration Count:

A practically meaningful improvement is visible in Table 1: all prior stochastic convergence results for these optimizers use stepsizes $t_i^k \propto K^{-3/4}$, constant throughout training and requiring the total number of iterations $K$ to be fixed in advance. Theorem 4.3 instead uses $t_i^k \propto k^{-3/4}$, which depends only on the current iteration index $k$. This makes the algorithm "anytime": it can be stopped at any point with valid convergence guarantees, without needing to commit to a training budget upfront. Similarly, the momentum parameter changes from $\beta^k \propto K^{-1/2}$ (prior work) to $\beta^k = 1 - (k+1)^{-1/2}$ (Theorem 4.3), removing the dependence on $K$.


Weaknesses

1. The "Predictive Power" Claim is Circular:

The paper's headline result, that theoretical stepsizes (~0.014 and ~0.77) closely match manually tuned values (0.018 and 1.08), is based on circular reasoning. The smoothness constants $L_i^0$ and $L_i^1$ are fitted post-hoc from the training trajectory produced by unScion using exactly those tuned hyperparameters from Pethick et al. (2025). These fitted constants are then plugged into Theorem 4.1's formula, and the resulting stepsizes unsurprisingly approximate the values that generated the trajectory in the first place. A genuinely predictive test would estimate the smoothness constants from an independent source (e.g., an AdamW run or a short pilot), set stepsizes via Theorem 4.1, and show competitive performance on a new training run without additional tuning. The AdamW transfer idea in Appendix F.3.3 hints at this direction but is not carried through to an actual end-to-end experiment.

2. Gap Between Exact and Approximate LMO:

The convergence theory assumes exact LMO computation (exact SVD at each step), but practical implementations use Newton-Schulz iterations to approximate the SVD (Appendix F.1). This approximation error is entirely absent from the analysis. For a paper whose central claim is bridging theory and practice, this is a significant omission: the algorithm actually implemented is not the algorithm actually analyzed. No quantitative bound is provided on how this approximation affects convergence, nor is there any empirical measurement of the approximation error along the training trajectory.

3. Adaptive Stepsizes Only Proven in the Deterministic Setting:



The adaptive stepsize formula from Theorem 4.1 only holds for full-batch gradients. The stochastic case (Theorem 4.3) reverts to a pre-specified schedule $t_i^k = t_i(k+1)^{-3/4}$, which still requires choosing the base stepsize $t_i$ and cannot adapt to local curvature during training. However, it is precisely the stochastic setting that matters for real LLM training.



4. Limited Scale of Empirical Validation:

The empirical analysis is conducted only on NanoGPT with 124M parameters trained for 5,000 iterations and a small CNN on CIFAR-10. While the paper repeatedly motivates its work by referencing large-scale LLM training, these scales are far from the regime where optimizers like Muon and Scion are most impactful. It remains unclear whether the key empirical findings, particularly $L_i^0 \approx 0$ and the large gap in $L_i^1$ across layer types, still hold at 1B+ parameters or with longer training runs.

---

> ### Author Rebuttal · Authors · 2026-03-31
>
> We thank the reviewer for the careful and constructive review. We especially appreciate that the reviewer recognizes the main strengths of the paper: the layer-wise LMO framework, the empirical evidence for strong cross-layer heterogeneity, and the anytime stochastic schedule in Theorem 4.3. We agree that the key remaining issue is how strongly the current experiments support the practical claim, and we address each point below.
>
> > Q1: Could the authors provide additional experiments at larger scales / longer runs?
>
> We agree that larger-scale evidence is important. We would like to clarify, however, that the current paper already includes a larger-scale study in Appendix F.3.5 beyond NanoGPT-124M: we report the same aggregate layer-wise smoothness analysis for GPT-2 Medium ($\sim 355$M) and GPT-2 Large ($\sim 774$M). Across all three scales, we consistently observe pronounced cross-layer heterogeneity together with the empirical trend $L_i^0 \approx 0$. So while we do not yet claim 1B+ evidence, the phenomenon is not limited to the 124M model, and we will make this much more explicit in the revision. We agree that longer runs and still larger scales would be valuable follow-up validation.
>
> > Q2: The smoothness constants are fitted from the same trajectory that was generated using tuned hyperparameters. Could the authors run an end-to-end experiment using an independent pilot?
>
> We agree that this is the most important practical caveat. To clarify our intended claim: the current paper should be read as establishing theory-guided relative layer-wise scaling and explaining the strong separation between layer types, rather than as a complete parameter-free end-to-end prescription. In fact, we have already softened the manuscript compared to earlier phrasing, and we are happy to soften this language further in the final version.
>
> That said, we do have evidence that the relevant relative scales can be estimated from an independent short pilot rather than only post hoc from the full trajectory. In an additional pilot-vs-full analysis, fitting only on iterations 500--1500 already recovers nearly identical $L_1$ values to fitting the full trajectory on representative layers, with very similar fit quality on the full trajectory (anonymous figure: https://imgur.com/a/yrwEvNt). This does not replace the end-to-end experiment suggested by the reviewer, but it is an encouraging first step toward exactly that use case. We also emphasize that the same qualitative separation appears under AdamW, where the embedding/output block still has fitted $L_1$ roughly one order of magnitude smaller than transformer blocks. This suggests that the cross-layer scale separation is not specific to unScion.
>
> > Q3: The analysis assumes exact LMOs, but practical implementations use Newton--Schulz to approximate the SVD. Could the authors comment on the approximation error and whether inexact-LMO analysis is feasible?
>
> Yes. We agree this is an important and natural extension, but we do not view it as undermining the core contribution of the present paper. Our main novelty is the layer-wise $(L_0,L_1)$-smoothness model and the faithful per-layer analysis; the exact-vs-approximate LMO question is largely orthogonal to that contribution. For some blocks (notably the $\|\cdot\|_{1\to\infty}$ block used for the embedding/output layer), the LMO is already exact. For spectral-norm blocks, practical implementations indeed use Newton--Schulz.
>
> The good news is that extending the theory to inexact LMOs appears feasible. In particular, the recent work "Beyond the Ideal: Analyzing the Inexact Muon Update" studies a $\delta$-accurate inexact surrogate and, in a simplified $p=1$ $(L_0,0)$-smooth setting, preserves the $O(1/\varepsilon^2)$ dependence with only controlled constant degradation depending on $\delta$. Complementarily, the recent work "Convergence of Muon with Newton--Schulz" shows that the gap to the exact SVD-based polar update shrinks extremely quickly with the number of Newton--Schulz steps, so that a small number of steps already behaves very close to the exact update. We agree that adding an empirical measurement of the exact-vs-approximate gap along our trajectories would be informative; we do not currently include such a measurement and will state this more clearly.
>
> > Adaptive stepsizes are only proved in the deterministic setting.
>
> The reviewer is correct that the fully adaptive formula in Theorem 4.1 is deterministic. In the stochastic setting, our contribution is an anytime current-iteration schedule under the more realistic layer-wise $(L_0,L_1)$-smoothness assumption, rather than a complete online curvature-adaptive rule. We will revise the wording to avoid overstating this point.
>
> ---
> We thank the reviewer again for the constructive suggestions. We hope these clarifications address the main concerns, and we would be grateful if the reviewer would reconsider the score in light of them.

---

> > ### Author Rebuttal · Reviewer_izv4 · 2026-04-02
> >
> > Thanks for the authors' response; I have updated my score accordingly. The authors have adequately addressed my core concerns. I believe this paper presents a clear contribution, though a few minor issues remain. Specifically, the following point should be revised in the final/camera-ready version:
> >
> > **The claim regarding the theory's ability to predict practical performance needs to be modified.** In fact, I found this counter-intuitive upon my initial reading. In optimization theory, the theoretical bounds derived for any class of problems inherently represent the worst-case scenarios within that class. I do not believe that the optimization of modern neural networks can be adequately encompassed by the worst-case problem under a non-convex and smooth setup. Neural network optimization problems undoubtedly possess more specific and benign properties than non-convex and smooth setup worst-case problem. While there may be some intersection, the latter absolutely cannot cover the former, particularly because most neural networks are fundamentally non-smooth.

---

> > > ### Author Response · Authors · 2026-04-07
> > >
> > > Thank you for the careful reading of our rebuttal and for updating your score. We also appreciate the helpful suggestion regarding the theoretical claims.
> > >
> > > We agree that worst-case optimization guarantees do not directly reflect the practical behavior of deep learning systems, and we do not claim that our bounds predict practical performance (e.g., iteration counts). Rather, our point in Section 2.2 is that prior analyses of Muon/Scion rely on classical global smoothness assumptions, which introduces several practical problems (e.g., they prescribe extremely small uniform step sizes across layers, which are inconsistent with the learning rates used in practice and fail to explain the empirical success of the family of optimizers we consider).
> > >
> > > Our contribution is to introduce a generalized layer-wise smoothness model, motivated by the heterogeneous geometry across layers in deep networks. While this model is still an abstraction, it enables theoretical guidance that better reflects practical implementations than prior analyses. We already discuss this point in the paper, and in the final version we will revise the wording to make it as clear as possible and avoid any potential misinterpretation.
> > >
> > > Thank you again for the constructive feedback.

---

### Official Review · Reviewer_aX7b · 2026-03-13

**Soundness:** 3
**Presentation:** 3
**Significance:** 3
**Originality:** 3
**Overall Recommendation:** 4
**Confidence:** 3

**Summary:**

This paper proposes Gluon, a unified layer-wise LMO-based optimization framework in which optimizers such as Muon and Scion arise as special cases.  They propose  a layer-wise generalized smoothness condition, namely layer-wise ((L_0,L_1))-smoothness, which captures cross-layer heterogeneity and enables the analysis of practical per-layer updates and stepsize choices.

**Compliance With Llm Reviewing Policy:**

Affirmed.

**Final Justification:**

Due to authors's detailed rebuttal. I increase my score to 4.

**Key Questions For Authors:**

1. This paper [1] also considers layer-wise LMO-based method, could authors discuss the connection between your work and their paper?

[1] Crawshaw M, Modi C, Liu M, et al. An exploration of non-euclidean gradient descent: Muon and its many variants. arXiv preprint 2025.

**Limitations:**

Yes

**Strengths And Weaknesses:**

# Strengths
1. The paper is well written. The proposed layer-wise LMO-based method is interesting
2. It is novel to give the  layer-wise smoothness  assumption, which also explain why we should consider different step size for different layers.

# Weaknesses
1. I am still not fully convinced that Gluon should be framed  as an adaptive optimizer in a practically meaningful sense. The deterministic adaptive stepsize in Theorem 4.1 depends on unknown layer-wise constants  $L_i^0$ and $L_i^1$ while the empirical experiments estimates these quantities from an observed training trajectory via post-hoc fitting. As a result, the current framework appears more like explanatory.
2. One  of limitations is that the theoretically adaptive stepsize is only derived in the deterministic setting. In the stochastic case, the main guarantee reverts to a pre-specified decaying schedule $t_i^k \propto (k+1)^{-3/4}$. This weakens the claim that the paper provides a practically complete theory of adaptive layer-wise optimization.
3. Although Gluon provides a layer-wise LMO-based framework that encompasses many optimizers such as Muon and Scion, the paper does not provide experiments or discussions on how to identify the optimal optimizer within this framework.

---

> ### Author Rebuttal · Authors · 2026-03-31
>
> We thank the reviewer for the thoughtful feedback and for recognizing the novelty of the layer-wise smoothness assumption and the Gluon framework.
>
> > I am not fully convinced that Gluon should be framed as an adaptive optimizer in a practically meaningful sense.
>
> We agree that the term "adaptive" can be read too strongly. Our intended meaning is that the stepsize depends on the current iterate through the layer-wise gradient norm, in the same spirit as normalized / AdaGrad-type methods; we do not mean that the method is parameter-free or that all quantities are estimated online without tuning. To avoid overstating the claim, we are happy to revise the framing toward theory-guided layer-wise stepsizes / geometry-aware scaling.
>
> At the same time, we do think the framework is practically meaningful and not merely explanatory. The key practical message is that layer-wise $(L_i^0,L_i^1)$ statistics provide a principled prior for relative per-layer scaling. These quantities need not be estimated post hoc from Gluon's own trajectory: they can be estimated from other optimizer trajectories as well (e.g., AdamW in Appendix F.3.3), because the phenomenon of strong cross-layer heterogeneity is not specific to unScion. In that sense, the role of the theory is to explain and guide the relative structure of the learning rates, even if one global scale still needs tuning.
>
> > The theoretically adaptive stepsize is only derived in the deterministic setting; the stochastic result reverts to a pre-specified schedule.
>
> The reviewer is correct that the fully adaptive formula of Theorem 4.1 is derived in the deterministic setting. In the stochastic case, our main theorem gives an anytime schedule $t_i^k=t_i (k+1)^{-3/4}$ together with $\beta_k = 1-(k+1)^{-1/2}$, which is still a substantial improvement over prior analyses that require constant stepsizes depending on the total horizon $K$. Moreover, Section E.1 studies an adaptive stochastic variant under a stronger assumption on the stochastic gradients. That said, we agree this is not yet a complete theory of stochastic adaptivity, and we will make that limitation even more explicit.
>
> > Gluon is broad, but the paper does not identify the optimal optimizer within the framework.
>
> Our goal here is to provide a faithful analytical framework for the class of layer-wise LMO methods already used in practice, rather than to solve the separate meta-problem of which norm / aggregation / momentum choice is best within the entire class. We view that optimizer-selection problem as important future work, but orthogonal to the present contribution. The main contribution of this paper is the new layer-wise $(L_0,L_1)$-smoothness model and the first convergence theory that is aligned with the actual per-layer implementation. In other words, our aim is to explain why this family works and how it should be analyzed faithfully, not yet to close the separate optimizer-search problem inside the family.
>
> We would also emphasize that the breadth of the framework is a feature rather than a weakness: it shows that Muon/unScion-style methods, layer-wise normalized updates, and related LMO-based rules fit into one common geometry-aware template. Establishing such a common template is a prerequisite for any principled future comparison of which special case is best.
>
> > Could the authors discuss the connection to Crawshaw et al. (2025)?
>
> Yes. We agree that this relation should be discussed more clearly. Crawshaw et al. appeared several months after our work, but their paper is clearly relevant and we will add a comparison in the final version. We view the two works as complementary. Crawshaw et al. explore a broad empirical family of non-Euclidean / Muon-style variants, different layer aggregations, and robustness questions. Our paper, in contrast, focuses on (i) a faithful layer-wise LMO formulation capturing practical Muon/Scion-style updates, (ii) a new layer-wise $(L_0,L_1)$-smoothness model tailored to this setting, and (iii) convergence guarantees under that model. In particular, the matrix updates considered in Crawshaw et al. can be interpreted as instances of the Gluon template under specific norm / aggregation choices. Thus their work is primarily about exploring algorithmic variants within this family, whereas ours is about the analytical framework and guarantees for the family.
>
> ---
>
> We appreciate the reviewer's comments and hope these clarifications address the main concerns.

---

> > ### Author Rebuttal · Reviewer_aX7b · 2026-04-02
> >
> > I appreciate the authors' detailed response. I will increase my score to 4.

---

> > > ### Author Response · Authors · 2026-04-07
> > >
> > > Thank you for taking the time to review our rebuttal and for the positive update. We appreciate your constructive feedback and are glad the clarifications addressed your concerns.

---

### Official Review · Reviewer_os8B · 2026-03-24

**Soundness:** 4
**Presentation:** 3
**Significance:** 4
**Originality:** 2
**Overall Recommendation:** 5
**Confidence:** 3

**Summary:**

This paper proposes a generalization of the Linear Minimization Oracle (LMO)-based optimizers while providing a theoretical understanding of why such optimizers work in practice. The authors identify two flaws in the current understanding of these optimizers: First, they do not capture layer-specific geometry and the L-smoothness assumption they are built on is incorrect. The $(L^0, L^1)$-smoothness introduced in previous research is generalized in this work and applied to LMO optimizers: The authors introduce a new layer-wise version of $(L^0, L^1)$-smoothness with arbitrary norms and apply it to derive layer-wise stepsizes in order to capture different layers' geometries. The main contribution is to provide a more accurate smoothness model that results in the layer-wise stepsizes mulitpliers and their schedule across training.. They provide empirical evidence that $(L^0, L^1)$-smoothness is the correct model and find $L^0_i \sim 0$ universally across all layers and different architectures, and optimizers, meaning smoothness is gradient-dependent rather than constant. The resulting framework, called Gluon, recovers existing optimizers including Muon and Scion as special cases under different norm choices, and derives the first convergence guarantees faithful to their actual per-layer implementation.

**Compliance With Llm Reviewing Policy:**

Affirmed.

**Key Questions For Authors:**

1. $\mu P$ Framework (Yang et al. 2022) is a popular framework used in practice which also predicts layer-wise scaling rules from a completely different theoretical angle. Given that Gluon also prescribe a layer-dependanet stepsize. How does Gluon compare to \mu P prescription?

2. The ratio $L^1_i/L^1_p$ should be optimizer-independent. However, the experiments show different factors between unScion (53x) and AdamW (13–21x). How do the you reconcile this quantitative discrepancy and the claim that smoothness statistics reliably guide stepsize initialization?

**Limitations:**

There are no potential negative societal impacts worth mentioning. Among the technical limitations, the authors mention (Sec 6) extending to nonlinear models and adaptive averaging as future work. For some of the technical limitations not discussed by the authors see weaknesses and questions.

**Strengths And Weaknesses:**

### Strengths

1. Revisiting the classical L-smoothness assumption is well-motivated initiative. Prior convergence analyses of Muon and Scion relied on a smoothness model that the paper convincingly invalidates. The authors generalize the existing $(L^0, L^1)$-smoothness of Zhang et al. (2020) to a layer-wise version with arbitrary norms, and empirically validate it across two optimizers (AdamW and unScion), two norm choices (Euclidean and specialized operator norms), multiple architectures (NanoGPT, CNN), and three model scales (124M, 355M, 774M parameters). Notably, the finding that $L^0_i \sim 0$ universally across all layers and settings is a strong empirical result. The experiments to validate Assumption 2.1 are clear and well designed.
2. The derivation of the layer-wise stepsize $t_i \approx 1/L^1_i$ directly from the smoothness constants is the paper's most practically impactful contribution. This allows to tune only the global stepsize multipliers, the layer-wise scales are provided by the derived quantity $1/L^1_i$. The papers predicted stepsizes that closely match the manually tuned values of Pethick et al. (2025).
3. Prior theory prescribed stepsizes proportional to $1/K^(3/4)$ which requires to know the total training steps in prior. Gluon paper predicts a power-law schedule $1/k^(3/4)$.
4. The paper provides an interesting convergence analysis that captures the per-layer LMO implementation used in practice by working directly in the product space S with separate per-layer LMOs, norms, and stepsizes. All prior analyses (Kovalev 2025, Li \& Hong 2025, Pethick et al. 2025) analyzed a simplified global update over a single norm ball Bk which is anyways computationally infeasible for large models and theoretically misaligned with practice.
5. Gluon provides a unifying framework that recovers Muon, unScion and other LMO-based optimizers through different norm choices, with convergence guarantees applying uniformly to all.

### Weaknesses

1. The authors derive a power law decaying schedule for the step size for the stochastic case in Theorem 4.3. However, the schedule is not used for the experiments which are sotchastic. Instead the authors use the constant schedule of the deterministic case derived in Theorem 4.2. without discussing/justifying their choice.

2. In practice to use Gluon derived step size, one need to estimate the $L^0_i$ and $L^1_i$ constants. The authors suggests in Appendix F.3.3 to use a run with AdamW optimizer and its tuned hyperparameters to estimate those constant. However, following this approach $L^0_i$ and $L^1_i$ Estimates will depend on the AdamW Learning Rate. The paper uses one specific AdamW configuration from (Pethick et al. 2025b, Table 7) and never tests whether the estimated $L^1_i$ values and consequently the prescribed Scion stepsizes change when a different AdamW learning rate is used. Since $L^0_i$ and $L^1_i$ are theoretically optimizer-independent properties of the loss landscape, the estimates should be robust to the AdamW learning rate but this is never verified. The paper even show that while the $(L^0, L^1)$-smoothness structure is consistent across optimizers, the estimated $L^1_i$ values ratio between layers is different between AdamW and unScion. This makes it not clear if, in practice, $L^1$ values can be reliably transferred across optimizers.

3. No Demonstration That the derived stepsizes actually improve training. Without this experiment the paper does not demonstrate that the derived stepsizes are useful in practice only that they are numerically close to tuned values. The paper establishes theoretical consistency but not practical superiority since the multiplying factor of the stepsize still needs to be tuned as in the experiment in Figure 3.

4. The Fitting Procedure for $L^0_i$ and $L^1_i$ is lacking some discussions. The hyperparameter \lambda controlling the underestimation penalty is never discussed. The fitting uses stochastic gradient estimates which introduces noise to the regression but there is no discussion of how many training steps are needed for reliable estimates.

5. The plots y axis and title are not clear for most of the plots.

---

> ### Author Rebuttal · Authors · 2026-03-31
>
> We thank the reviewer for the careful reading and constructive feedback. We are glad that the reviewer found the paper technically solid and appreciated both the layer-wise smoothness model and the faithful per-layer analysis of practical LMO-based optimizers. Below we clarify the main points.
>
> >The authors derive a power-law decaying schedule for the stochastic case, but do not use it in the stochastic experiments.
>
> This is correct. Our experiments were designed primarily to validate Assumption 2.1 along practical training trajectories, rather than to benchmark a particular schedule. For that purpose, we kept the learning rates constant to avoid introducing an additional confound into the trajectory-smoothness measurements. A decaying schedule would change the path geometry and make the interpretation of the measured $\hat L_i[k]$ less clean. We will make this motivation explicit.
>
> >In practice, one needs to estimate $L_i^0,L_i^1$, and the estimates may depend on the AdamW learning rate.
>
> We agree that the exact fitted values obtained from a finite stochastic trajectory are not expected to be numerically identical across optimizers, or even across runs of the same optimizer. The theorem concerns layer-wise smoothness of the objective, whereas the experiments estimate trajectory-level proxies along the path actually taken by an optimizer. Accordingly, our practical claim is not that the exact fitted numbers are optimizer-invariant, but that the coarse ordering and scale separation across layer types are stable enough to guide initialization of relative per-layer learning rates.
>
> This is the consistent pattern we observe: under both unScion and AdamW, the embedding/output layers have much smaller fitted $L_1$ than transformer blocks. In practice, we therefore view these statistics as a prior for relative scaling rather than an exact plug-in oracle. This reduces the tuning problem from $p$ unrelated layer-wise multipliers to essentially one global multiplier together with theory-guided relative proportions.
>
> A systematic AdamW-learning-rate sensitivity sweep would indeed be valuable. While we do not include such a sweep in the current version, the main transfer claim we make is qualitative rather than exact: whether the observed gap is roughly $53\times$ on unScion or $13$--$21\times$ on AdamW, both trajectories indicate that the embedding/output block should use a much larger step than transformer blocks.
>
> >No demonstration that the derived stepsizes actually improve training.
>
> We agree that the current paper does not yet establish a fully parameter-free end-to-end prescription. The supported practical claim is narrower: the theory explains why tuned scales should differ so strongly across layers, and why the embedding/output block should use a much larger step than transformer blocks. The empirical agreement with the tuned $(\rho_2,\rho_3)$ region should therefore be viewed as theory-guided relative scaling / initialization.
>
> >The fitting procedure for $L_i^0,L_i^1$ lacks discussion of $\lambda$ and estimate stability.
>
> We will specify the exact value of $\lambda$ used and explain its role more clearly: it penalizes underestimation because Assumption 2.1 is an inequality. Regarding noise, full gradients are infeasible in the LLM experiments, so those fits necessarily use stochastic gradients; however, on CIFAR-10 we also validate the same phenomenon in the full-batch setting, and the conclusions are consistent. We also agree that estimate stability should be discussed more explicitly. In an additional pilot-vs-full analysis, fitting only on iterations 500--1500 already recovers nearly identical $L_1$ values to fitting the full trajectory on representative layers (https://imgur.com/a/yrwEvNt), suggesting that the relative scales stabilize fairly early.
>
> >How does this compare to $\mu$P?
>
> We view $\mu$P and Gluon as complementary rather than competing. $\mu$P prescribes how hyperparameters should scale with width / parameterization across model sizes, while our work studies how stepsizes should differ across layers within a fixed model and training regime based on layer-wise smoothness geometry.
>
> >The ratio $L_i^1/L_p^1$ should be optimizer-independent, yet the measured factors differ.
>
> Because the experimental quantities are trajectory-level proxies, different optimizers can produce different numerical fits by visiting different regions of the loss landscape. The practically relevant point is that both trajectories reveal the same order-of-magnitude separation between embedding/output layers and transformer blocks. Our claim is therefore about robust heterogeneity and useful relative initialization, not exact optimizer-independent equality of the fitted ratios.
>
> >The plots are not clear.
>
> Thank you for pointing this out. We will improve axis labels, titles readability of the plots in the final version.
>
> ---
> We thank the reviewer again for the helpful comments, which we will incorporate in the final version.

---

> > ### Author Rebuttal · Reviewer_os8B · 2026-04-04
> >
> > I thank the authors for the detailed answers and maintain the positive evaluation of the work.
> >
> > Some of the questions still left for me:
> > 1. You say μP and Gluon are complementary since "μP prescribes how hyperparameters scale with width," but μP also prescribes layer-wise relative scaling within a fixed model (e.g. width-independent embeddings). How is Gluon's contribution distinct in practice?
> > 2. Also I would recommend to make the distinction between the theory explaining existing tuned hyperparameters post-hoc and actionable end-to-end prescription maximally clear in the revised version of the paper, so that the message to practitioners is clear.

---

> > > ### Author Response · Authors · 2026-04-07
> > >
> > > Thank you for taking the time to read our rebuttal and for the thoughtful follow-up. We provide additional clarification below.
> > >
> > > 1. Thank you for pointing this out. Our claim of complementarity refers to the different aspect of training that each framework addresses:
> > > - $\mu$P specifies a parameterization and initialization scheme that ensures training dynamics remain stable and width-invariant when scaling models. In particular, it determines how weights and learning rates should scale with width so that gradients and updates have consistent magnitudes across model sizes.
> > > - Gluon, in contrast, studies layer-wise step-size allocation during optimization for a fixed architecture and parameterization. Our analysis shows that different layers exhibit substantially different smoothness constants, implying that uniform learning rates across layers are suboptimal. Gluon therefore proposes layer-dependent stepsizes derived from layer-wise smoothness geometry, which improves optimization efficiency without changing the underlying parameterization.
> > >
> > > In practice, this means the two approaches operate at different levels of the training stack:
> > > - $\mu$P: how to parameterize and scale models across width. Intuitively, it ensures that each layer’s updates have a comparable effect on activations across different widths (in the large-width limit).
> > > - Gluon: how to set optimizer step sizes across layers within a given model under a fixed parameterization, based on the layer-wise smoothness structure of the optimization problem.
> > >
> > > Overall, the two frameworks arise from different principles: $\mu$P from parameterization and width scaling, and our approach from generalized smoothness and optimization geometry. Understanding their precise interaction is an interesting direction for future work but is beyond the scope of this paper.
> > >
> > > 2. Thank you for the suggestion. We already discuss this in the main paper (as much as permitted by the page limit) and in the appendix. We agree that additional discussion would be valuable and will expand it in the final version.
> > >
> > > Thank you again for the useful feedback. We appreciate the opportunity to clarify these points.

---

### Decision · Program_Chairs · 2026-04-30

**Decision:**

Accept (regular)

**Comment:**

The contributions behind Gluon should be explicitly stated, the distinction between the theory explaining existing tuned hyperparameters post-hoc and actionable end-to-end prescription should be clearly explained for practitioners, and the claim regarding the theory's ability to predict practical performance needs to be modified.